# "Lossless" Compression of Deep Neural Networks: A High-dimensional Neural Tangent Kernel Approach

**Lingyu Gu**[*1]    **Yongqi Du**[*1]    **Yuan Zhang**[2]    **Di Xie**[2]    **Shiliang Pu**[2]    **Robert C. Qiu**[1]

**Zhenyu Liao**[†1]

[1]EIC, Huazhong University of Science and Technology, Wuhan, China
[2]Hikvision Research Institute, Hangzhou, China

## Abstract

Modern deep neural networks (DNNs) are extremely powerful; however, this comes at the price of increased depth and having more parameters per layer, making their training and inference more computationally challenging. In an attempt to address this key limitation, efforts have been devoted to the compression (e.g., sparsification and/or quantization) of these large-scale machine learning models, so that they can be deployed on low-power IoT devices. In this paper, building upon recent advances in neural tangent kernel (NTK) and random matrix theory (RMT), we provide a novel compression approach to wide and fully-connected *deep* neural nets. Specifically, we demonstrate that in the high-dimensional regime where the number of data points $n$ and their dimension $p$ are both large, and under a Gaussian mixture model for the data, there exists *asymptotic spectral equivalence* between the NTK matrices for a large family of DNN models. This theoretical result enables "lossless" compression of a given DNN to be performed, in the sense that the compressed network yields asymptotically the same NTK as the original (dense and unquantized) network, with its weights and activations taking values *only* in $\{0, \pm 1\}$ up to a scaling. Experiments on both synthetic and real-world data are conducted to support the advantages of the proposed compression scheme, with code available at `https://github.com/Model-Compression/Lossless_Compression`.

## 1 Introduction

Modern deep neural networks (DNNs) are becoming increasingly over-parameterized, having more parameters than required to fit the also increasingly large, complex, and high-dimensional data. While the list of successful applications of these large-scale machine learning (ML) models is rapidly growing, the energy consumption of these models is also increasing, making them more challenging to deploy on close-to-user and low-power devices. To address this issue, compression techniques have been proposed that prune, sparsify, and/or quantize DNN models [14, 23], thereby yielding DNNs of a much smaller size that can still achieve satisfactory performance on a given ML task. As an illustrative example, it has been recently shown that at least $90\%$ of the weights in popular DNN models such as VGG19 and ResNet32 can be removed with virtually no performance loss [53].

Despite the remarkable progress achieved by various DNN model compression techniques, due to the nonlinear and highly non-convex nature of DNNs, our theoretical understanding of these large-scale ML models, as well as of their compression schemes, is progressing at a more modest pace. For

---

[*]Equal contribution, listed in a random order by rolling a dice on WeChat.
[†]Author to whom any correspondence should be addressed. Email: zhenyu_liao@hust.edu.cn

36th Conference on Neural Information Processing Systems (NeurIPS 2022).

example, it remains unclear how much a given DNN model can be compressed *without* severe performance degradation; perhaps more importantly, on the degree to which such a *performance and complexity trade-off* depends on the ML task and the data also remains unknown.

In this respect, neural tangent kernels (NTKs) [28], provide a powerful tool for use in assessing the convergence and generalization properties of very wide (sometimes unrealistically so) DNNs by studying their corresponding NTK eigenspectra, which are *solely* dependent on the input data, the network architecture and activation function, as well as the random weights distribution.[1]

In this paper, building upon recent advances in random matrix theory (RMT) and high-dimensional statistics, we demonstrate that for data $\mathbf{x}_1, \ldots, \mathbf{x}_n \in \mathbb{R}^p$ drawn from a $K$-class Gaussian mixture model (GMM), in a high-dimensional and non-trivial classification regime where the input data dimension $p$ and their size $n$ are both large and comparable, the *eigenspectra* of both the NTK and the closely related conjugate kernel (CK) matrices at *any* layer $\ell \in \{1, \ldots, L\}$ are *independent* of the distribution of the i.i.d. entries of the (random) weight matrix $\mathbf{W}_\ell$, provided that they are "normalized" to have zero mean and unit variance, and *only* depend on the activation function $\sigma_\ell(\cdot)$ via *four* scalar parameters. In a sense, we establish, at least for GMM data, the asymptotic *spectral equivalence* between the CK and NTK matrices of the corresponding network layers, and consequently of the whole network, for a large family of DNN models with possibly very different weights and activations, given that they have normalized weights and share the same few activation-related parameters.

Since the convergence and generalization properties of wide DNNs depend only on the eigenspectra (i.e., eigenvalue-eigenvector pairs; see also Remark 2 below) of the corresponding NTK matrices [28, 18], in the sense that, e.g., the time evolutions (when trained with gradient descent using a sufficiently small step size) of both the residual error and the in-sample prediction of the network can be expressed as *explicit* functions of NTK eigenvalues and eigenvectors [18, 1, 26], we further exploit the above theoretical results to propose a novel "lossless" compression approach for *fully-connected* DNN models, by designing a sparse and quantized DNN that (i) has asymptotically the *same* NTK eigenspectra as the original "dense and full precision" network, and (ii) has both weights and activations taking values in the set $\{-1, 0, +1\}$ before scaling, and can thus be stored and computed much more efficiently.

Despite being derived here for Gaussian mixture data, an unexpected close match is observed between our theory and the empirical results on real-world datasets, suggesting possibly wider applicability for the proposed "lossless" compression approach. Looking forward, we expect that our analysis will open the door to improved analysis of involved ML methods based on RMT and high-dimensional statistics, which will demystify the seemingly striking empirical observations in, say, modern DNNs.

## 1.1 Our contributions

Our main results can be summarized as follows:

1. We provide, in Theorems 1 and 2 respectively, for GMM data and in the high-dimensional regime of Assumption 1, *precise* eigenspectral characterizations of CK and NTK matrices of fully-connected DNNs; we particularly show that the CK and NTK eigenspectra do *not* depend on the distribution of i.i.d. network weights, and *depend* solely on the activation function of each layer via a few scalar parameters.

2. In Corollary 1 and Algorithm 1, we exploit these results to propose a novel DNN compression scheme, with *sparsified and ternarized* weights and activations, without affecting the NTK spectral behavior, and thus the convergence and generalization properties of the network.

3. We provide empirical evidence on (not so) wide DNNs trained on both synthetic Gaussian and real-world datasets such as MNIST [31] and CIFAR10 [30], and show a factor of $10^3$ less memory is needed with the proposed DNN compression approach, with virtually no performance loss.

---

[1]In the remainder of this article, what we refer to as the "NTK matrix" is essentially the limiting (and nonrandom) NTK matrix to which the random NTK converges under the infinite-wide limit.

## 1.2 Related work

**Neural network model compression.** The study of NN compression dates back to early 1990 [32], at which point, in the absence of the (possibly more than) sufficient computational power that we have today, compression techniques allowed neural networks to be empirically evaluated on computers with limited computational and/or storage resources [49]. Alongside the rapid growth of increasingly powerful computing devices, the development of more efficient NN architectures and training/inference protocols, and the need to implement NNs on mobile and low-power devices, (D)NN model compression has become an active research topic and many elegant and efficient compression approaches have been proposed over the years [22, 25, 27, 23]. However, due to the nonlinear and highly non-convex nature of DNNs, our theoretical understanding of these large-scale ML models, as well as of (e.g., the fundamental "performance and complexity" trade-off of) compressed DNNs, is somewhat limited [23].

**Neural tangent kernel.** Neural tangent kernel (NTK) theory proposed in [28], by considering the limit of infinitely wide DNNs, characterizes the convergence and generalization properties of very wide DNNs when trained using gradient descent with small steps. Initially proposed for fully-connected nets, the NTK framework has been subsequently extended to convolutional [5], graph [16], and recurrent [3] settings. The NTK theory, while having the advantage of being mathematically more tractable (via, e.g., the characterization of the associated reproducing kernel Hilbert space [8]), seems to diverge from the regime on which modern (and not so wide) DNNs operate, see [10, 38, 18].

**Random matrix theory and neural networks.** Random matrix theory (RMT), a powerful and flexible tool for assessing the behavior of large-scale systems with a large "degree of freedom", is gaining popularity in the field of NN analysis [44, 45], in both shallow [48, 36, 37] and deep [7, 18, 47] settings, and for both homogeneous (e.g., standard normal) [48, 46] and mixture-type data [36, 4]. From a technical perspective, the most relevant paper is [4] , in which the authors proposed a RMT-inspired NN compression scheme, albeit only in the single-hidden-layer setting. This paper extends the analysis in [4] to multi-layer fully-connected DNNs, by focusing on both the CK and NTK matrices, and proposes a novel sparsification and quantization scheme for fully-connected DNNs (which is in spirit similar to, although formally different from, that in [4]).

## 1.3 Notations and organization of the paper

We denote scalars by lowercase letters, vectors by bold lowercase, and matrices by bold uppercase. We denote the transpose operator by $(\cdot)^\mathsf{T}$, and use $\|\cdot\|$ to denote the Euclidean norm for vectors and the spectral/operator norm for matrices. For a random variable $z$, $\mathbb{E}[z]$ denotes the expectation of $z$. We use $\mathbf{1}_p$ and $\mathbf{I}_p$ to represent an all-ones vector of dimension $p$ and the identity matrix of size $p \times p$.

The remainder of this article is structured as follows. In Section 2, we present the fully-connected DNN model under study, together with our working assumptions. Section 3 contains our main technical results on the eigenspectra of the conjugate kernel $\mathbf{K}_{\mathrm{CK}}$ and NTK matrix $\mathbf{K}_{\mathrm{NTK}}$, along with an account of how they apply to the compression of fully-connected DNNs with the proposed "lossless" compression scheme. Empirical evidence is provided in Section 4 to demonstrate the significant computation and storage savings, with minimal performance degradation, that can be obtained using the proposed approach. Conclusion and future perspectives are placed in Section 5.

## 2  Preliminaries

Let $\mathbf{x}_1, \ldots, \mathbf{x}_n \in \mathbb{R}^p$ be $n$ data vectors independently drawn from one of the $K$-class Gaussian mixtures $\mathcal{C}_1, \ldots, \mathcal{C}_K$, and let $\mathbf{X} = [\mathbf{x}_1, \ldots, \mathbf{x}_n] \in \mathbb{R}^{p \times n}$, with class $\mathcal{C}_a$ having cardinality $n_a$; that is,

$$\mathbf{x}_i \in \mathcal{C}_a \Leftrightarrow \mathbf{x}_i \sim \mathcal{N}(\boldsymbol{\mu}_a/\sqrt{p}, \mathbf{C}_a/p), \tag{1}$$

for mean vector $\boldsymbol{\mu}_a \in \mathbb{R}^p$ and covariance matrix $\mathbf{C}_a \in \mathbb{R}^{p \times p}$ associated with class $\mathcal{C}_a$.

In the high-dimensional scenario where $n, p$ are both large and comparable, we position ourselves in the following non-trivial classification setting, so that the classification of the $K$-class mixture is neither trivially easy nor impossible; see also [13] and [9, Section 2].

**Assumption 1** (High-dimensional asymptotics). *As $n \to \infty$, we have, for $a \in \{1, \ldots, K\}$ that (i) $p/n \to c \in (0, \infty)$ and $n_a/n \to c_a \in [0, 1)$; (ii) $\|\boldsymbol{\mu}_a\| = O(1)$; (iii) for $\mathbf{C}^\circ \equiv \sum_{a=1}^K \frac{n_a}{n} \mathbf{C}_a$*

and $\mathbf{C}_a^\circ \equiv \mathbf{C}_a - \mathbf{C}^\circ$, *we have* $\|\mathbf{C}_a\| = O(1)$, $\operatorname{tr}\mathbf{C}_a^\circ = O(\sqrt{p})$ *and* $\operatorname{tr}(\mathbf{C}_a\mathbf{C}_b) = O(p)$ *for* $a,b \in \{1,\ldots,K\}$*; and (iv)* $\tau_0 \equiv \sqrt{\operatorname{tr}\mathbf{C}^\circ/p}$ *converges in* $(0,\infty)$.

We consider using a fully-connected DNN model of depth $L$ for the classification of the above $K$-class Gaussian mixture. Such a network can be parameterized by a sequence of weight matrices $\mathbf{W}_1 \in \mathbb{R}^{d_1 \times d_0}, \ldots, \mathbf{W}_L \in \mathbb{R}^{d_L \times d_{L-1}}$ (with the convention $d_0 = p$), and nonlinear activation functions $\sigma_1, \ldots, \sigma_L$ that apply entry-wise, so that the network output $f(\mathbf{x}) \in \mathbb{R}$ is given by:

$$f(\mathbf{x}) = \frac{1}{\sqrt{d_L}}\mathbf{w}^\mathsf{T}\sigma_L\left(\frac{1}{\sqrt{d_{L-1}}}\mathbf{W}_L\sigma_{L-1}\left(\ldots\frac{1}{\sqrt{d_2}}\sigma_2\left(\frac{1}{\sqrt{d_1}}\mathbf{W}_2\sigma_1(\mathbf{W}_1\mathbf{x})\right)\right)\right), \quad (2)$$

for an input data vector $\mathbf{x} \in \mathbb{R}^p$ and output vector $\mathbf{w} \in \mathbb{R}^{d_L}$. We denote $\boldsymbol{\Sigma}_\ell \in \mathbb{R}^{d_\ell \times n}$ the representations of the data matrix $\mathbf{X} \in \mathbb{R}^{p \times n}$ at layer $\ell \in \{1,\ldots,L\}$ defined as

$$\boldsymbol{\Sigma}_\ell = \frac{1}{\sqrt{d_\ell}}\sigma_\ell\left(\frac{1}{\sqrt{d_{\ell-1}}}\mathbf{W}_\ell\sigma_{\ell-1}\left(\ldots\frac{1}{\sqrt{d_2}}\sigma_2\left(\frac{1}{\sqrt{d_1}}\mathbf{W}_2\sigma_1(\mathbf{W}_1\mathbf{X})\right)\right)\right). \quad (3)$$

The normalization by $1/\sqrt{d_\ell}$ follows from the NTK literature and ensures the consistent asymptotic behavior of the network in the high-dimensional setting in Assumption 1 and 2; see also [28, 8, 18].

The training and generalization performance of the NN model defined in (2) are closely related to two types of kernel matrices: the Conjugate Kernel (CK) matrix and Neural Tangent Kernel (NTK) matrix, defined respectively for $\ell \in \{1,\ldots,L\}$ as follows:

$$\mathbf{K}_{\mathrm{CK},\ell} = \mathbb{E}[\boldsymbol{\Sigma}_\ell^\mathsf{T}\boldsymbol{\Sigma}_\ell] \in \mathbb{R}^{n \times n}, \quad (4)$$

with expectation taken with respect to the random weights $\mathbf{W}_1,\ldots,\mathbf{W}_\ell$ and $\boldsymbol{\Sigma}_\ell \in \mathbb{R}^{d_\ell \times n}$ the data representation at the output of layer $\ell$ defined in (3). In particular, CK matrices are known to satisfy the following recursive relation [28, 8]

$$[\mathbf{K}_{\mathrm{CK},\ell}]_{ij} = \mathbb{E}_{u,v}[\sigma_\ell(u)\sigma_\ell(v)], \text{ with } u,v \sim \mathcal{N}\left(\mathbf{0}, \begin{bmatrix} [\mathbf{K}_{\mathrm{CK},\ell-1}]_{ii} & [\mathbf{K}_{\mathrm{CK},\ell-1}]_{ij} \\ [\mathbf{K}_{\mathrm{CK},\ell-1}]_{ij} & [\mathbf{K}_{\mathrm{CK},\ell-1}]_{jj} \end{bmatrix}\right), \quad (5)$$

while for the NTK matrix $\mathbf{K}_{\mathrm{NTK},\ell} \in \mathbb{R}^{n \times n}$ of layer $\ell$, we have:

$$\mathbf{K}_{\mathrm{NTK},\ell} = \mathbf{K}_{\mathrm{CK},\ell} + \mathbf{K}_{\mathrm{NTK},\ell-1} \circ \mathbf{K}'_{\mathrm{CK},\ell}, \quad \mathbf{K}_{\mathrm{NTK},0} = \mathbf{K}_{\mathrm{CK},0} = \mathbf{X}^\mathsf{T}\mathbf{X}, \quad (6)$$

where '$\mathbf{A} \circ \mathbf{B}$' denotes the Hadamard product between two matrices $\mathbf{A},\mathbf{B}$ of the same size, and $\mathbf{K}'_{\mathrm{CK},\ell}$ denotes the CK matrix with nonlinear function $\sigma'_\ell$ instead of $\sigma_\ell$ as for $\mathbf{K}_{\mathrm{CK},\ell}$ defined in (5); that is, $[\mathbf{K}'_{\mathrm{CK},\ell}]_{ij} = \mathbb{E}_{u,v}[\sigma'_\ell(u)\sigma'_\ell(v)]$. Note in particular that for a given DNN model, the corresponding CK and NTK matrices depend *only* on the network structure (i.e., the number of layers and the activation function in each layer), the *distribution* of the random (initializations of the) weights to be integrated over (e.g., in the expectation in equation (4)), and the input data.

It has been shown in a series of previous efforts [28, 18, 26] that for very (and sometimes unrealistically) wide DNNs trained using gradient descent with a small step size, the time evolution of the residual errors and in-sample predictions of a given DNN are *explicit* functionals of the corresponding $\mathbf{K}_{\mathrm{NTK}}$ involving its eigenvalues and eigenvectors. In this respect, the NTK theory provides, via the eigenspectral behavior of $\mathbf{K}_{\mathrm{NTK}}$, precise characterizations of the convergence and generalization properties of DNNs [28, 8], by focusing on the impact of the network structure, the input data, and the weight initialization schemes.

In this paper, we focus on fully-connected nets under the following assumption regarding the weights.

**Assumption 2** (On weight initializations). *The random weights* $\mathbf{W}_1 \in \mathbb{R}^{d_1 \times p}, \ldots, \mathbf{W}_L \in \mathbb{R}^{d_L \times d_{L-1}}$ *are independent and have i.i.d. entries of zero mean, unit variance, and finite fourth-order moment.*

Assumption 2, together with the $1/\sqrt{d_\ell}$ normalization, is compatible with fully-connected DNNs in (2), which are admittedly less interesting, from a practical perspective, compared to their convolutional counterparts. The proposed framework is envisioned to be extendable to a convolutional [5, 8] and more involved setting (e.g., graph NNs [16]) by considering (e.g., Toeplitz-type) structures on $\mathbf{W}$s.

Unlike most existing NTK literature [28, 8, 17], we do not assume the Gaussianity of the entries of $\mathbf{W}_\ell$s, but only that they are i.i.d. and "normalized" to have zero mean and unit variance. As it turns

out, this assumption together with a (Lyapunov-type) central limit theorem argument, is sufficient to establish most existing results on the convergence and generalization of DNNs; see for example [34].

We also need the following assumption on the activation functions in each layer.

**Assumption 3** (On activation functions)**.** *The activations $\sigma_1, \ldots, \sigma_L$ are at least four-times differentiable with respect to standard normal measure, in the sense that $\max_{k \in \{0,1,2,3,4\}}\{|\mathbb{E}[\sigma_\ell^{(k)}(\xi)]|\} < C$ for some universal constant $C > 0$, $\xi \sim \mathcal{N}(0,1)$, and $\ell \in \{1, \ldots, L\}$.*

Using the Gaussian integration by parts formula, one has $\mathbb{E}[\sigma'(\xi)] = \mathbb{E}[\xi\sigma(\xi)]$ for $\xi \sim \mathcal{N}(0,1)$, as long as the right-hand side expectation exists. As a result, for non-differentiable functions, it suffices to have $|\sigma_\ell|$ upper-bounded by some (high-degree) polynomial function for Assumption 3 to hold.

With these preliminaries, we are now ready to present our main technical results on the eigenspectral behavior of the CK and NTK matrices for a large family of fully-connected DNN models.

## 3 Main results

For a fully-connected DNN defined in (2), our first result is on the eigenspectral behavior of the corresponding CK matrices $\mathbf{K}_{\mathrm{CK}}$ defined in (4). More specifically, we show for Gaussian mixture data in (1) and in the high-dimensional setting of Assumption 1, that the $\mathbf{K}_{\mathrm{CK},\ell}$ of layer $\ell \in \{1, \ldots, L\}$ is asymptotically *spectrally equivalent* to another random matrix $\tilde{\mathbf{K}}_{\mathrm{CK},\ell}$, in the sense that their spectral norm difference $\|\mathbf{K}_{\mathrm{CK},\ell} - \tilde{\mathbf{K}}_{\mathrm{CK},\ell}\|$ vanishes as $n, p \to \infty$. This result is stated as follows, the proof of which is based on an induction on $\ell$ and is given in Section A.1 of the appendix.

**Theorem 1** (Asymptotic spectral equivalents for CK matrices)**.** *Let Assumptions 1–3 hold, and let $\tau_0, \tau_1, \ldots, \tau_L \geq 0$ be a sequence of non-negative numbers satisfying the following recursion:*

$$\tau_\ell = \sqrt{\mathbb{E}[\sigma_\ell^2(\tau_{\ell-1}\xi)]}, \quad \xi \sim \mathcal{N}(0,1), \quad \ell \in \{1, \ldots, L\}. \tag{7}$$

*Further assume that the activation functions $\sigma_\ell(\cdot)$s are "centered," such that $\mathbb{E}[\sigma_\ell(\tau_{\ell-1}\xi)] = 0$. Then, for the CK matrix $\mathbf{K}_{\mathrm{CK},\ell}$ of layer $\ell \in \{0, 1, \ldots, L\}$ defined in (4), as $n, p \to \infty$, one has that:*

$$\|\mathbf{K}_{\mathrm{CK},\ell} - \tilde{\mathbf{K}}_{\mathrm{CK},\ell}\| \to 0, \quad \tilde{\mathbf{K}}_{\mathrm{CK},\ell} \equiv \alpha_{\ell,1}\mathbf{X}^\mathsf{T}\mathbf{X} + \mathbf{V}\mathbf{A}_\ell\mathbf{V}^\mathsf{T} + (\tau_\ell^2 - \tau_0^2\alpha_{\ell,1} - \tau_0^4\alpha_{\ell,3})\mathbf{I}_n, \tag{8}$$

*almost surely, with*

$$\mathbf{V} = [\mathbf{J}/\sqrt{p},\ \psi] \in \mathbb{R}^{n \times (K+1)}, \quad \mathbf{A}_\ell = \begin{bmatrix} \alpha_{\ell,2}\mathbf{t}\mathbf{t}^\mathsf{T} + \alpha_{\ell,3}\mathbf{T} & \alpha_{\ell,2}\mathbf{t} \\ \alpha_{\ell,2}\mathbf{t}^\mathsf{T} & \alpha_{\ell,2} \end{bmatrix} \in \mathbb{R}^{(K+1) \times (K+1)}, \tag{9}$$

*for class label vectors $\mathbf{J} = [\mathbf{j}_1, \ldots, \mathbf{j}_K] \in \mathbb{R}^{n \times K}$ with $[\mathbf{j}_a] = \delta_{\mathbf{x}_i \in \mathcal{C}_a}$, second-order data fluctuation vector $\psi = \{\|\mathbf{x}_i - \mathbb{E}[\mathbf{x}_i]\|^2 - \mathbb{E}[\|\mathbf{x}_i - \mathbb{E}[\mathbf{x}_i]\|^2]\}_{i=1}^n \in \mathbb{R}^n$, second-order discriminative statistics $\mathbf{t} = \{\mathrm{tr}\,\mathbf{C}_a^\circ/\sqrt{p}\}_{a=1}^K \in \mathbb{R}^K$ and $\mathbf{T} = \{\mathrm{tr}\,\mathbf{C}_a\mathbf{C}_b/p\}_{a,b=1}^K \in \mathbb{R}^{K \times K}$ of the Gaussian mixture in (1), as well as non-negative scalars $\alpha_{\ell,1}, \alpha_{\ell,2}, \alpha_{\ell,3} \geq 0$ satisfying the following recursions:*

$$\alpha_{\ell,1} = \mathbb{E}[\sigma_\ell'(\tau_{\ell-1}\xi)]^2\alpha_{\ell-1,1}, \quad \alpha_{\ell,2} = \mathbb{E}[\sigma_\ell'(\tau_{\ell-1}\xi)]^2\alpha_{\ell-1,2} + \frac{1}{4}\mathbb{E}[\sigma_\ell''(\tau_{\ell-1}\xi)]^2\alpha_{\ell-1,4}^2, \tag{10}$$

$$\alpha_{\ell,3} = \mathbb{E}[\sigma_\ell'(\tau_{\ell-1}\xi)]^2\alpha_{\ell-1,3} + \frac{1}{2}\mathbb{E}[\sigma_\ell''(\tau_{\ell-1}\xi)]^2\alpha_{\ell-1,1}^2, \tag{11}$$

*with $\alpha_{\ell,4} = \alpha_{\ell-1,4}\mathbb{E}\left[(\sigma_\ell'(\tau_{\ell-1}\xi))^2 + \sigma_\ell(\tau_{\ell-1}\xi)\sigma_\ell''(\tau_{\ell-1}\xi)\right]$ for $\xi \sim \mathcal{N}(0,1)$.*

A few remarks on Theorem 1 are in order. The first remark is on the assumption $\mathbb{E}[\sigma_\ell(\tau_{\ell-1}\xi)] = 0$.

**Remark 1** (On activation centering)**.** *The condition $\mathbb{E}[\sigma_\ell(\tau_{\ell-1}\xi)] = 0$, seemingly restrictive at first sight, in fact only subtracts an* identical constant *from all entries of the data representation $\Sigma_\ell$ at layer $\ell$, and should therefore* not *restrict the expressive power of the network, nor its performance on downstream ML tasks. For a given DNN model of interest, it suffices to "center" the output of each layer by subtracting a constant to satisfy Assumption 3, and to further apply our Theorem 1.*

Theorem 1 unveils the (possibly surprising) fact that, for the high-dimensional and non-trivial Gaussian mixture classification in (1), the spectral behavior of $\tilde{\mathbf{K}}_{\mathrm{CK},\ell}$, and thus that of the CK matrix $\mathbf{K}_{\mathrm{CK},\ell}$, is (i) *independent* of the distribution of the (entries of the) weights $\mathbf{W}_\ell$ when they are

"normalized" to have zero mean and unit variance, as demanded in Assumption 2, and (ii) depends on the activation function $\sigma_\ell$ *only* via four[2] scalar parameters $\alpha_{\ell,1}, \alpha_{\ell,2}, \alpha_{\ell,3}$ and $\tau_\ell$: such universal results have been previously observed in random matrix theory and high-dimensional statistics literature (see for example [12, 6, 56]) and indicate some kind of universality of DNN models.

On closer inspection of Theorem 1, we further observe that:

(i) for a given DNN, Theorem 1 characterizes, via the form of $\tilde{\mathbf{K}}_{\mathrm{CK},\ell}$ in (8) and the recursions in (10) and (11), how the linear (via $\alpha_{\ell,1}$, which is multiplied by $\mathbf{X}^\mathsf{T}\mathbf{X}$) and nonlinear (via $\alpha_{\ell,2}$ and $\alpha_{\ell,3}$ in $\mathbf{A}_\ell$, which respectively weight the second-order data statistics $\mathbf{t}$ and $\mathbf{T}$) data features "propagate" in a DNN, in a layer-by-layer fashion, as $\ell$ increases, as *quantitatively* measured by the corresponding $\alpha_\ell$s; and

(ii) for two DNNs with the same number of layers, but possibly different weights and activations, given the same input data $\mathbf{X}$ (so that the two nets have the same $\mathbf{K}_{\mathrm{CK},0}$), if they have asymptotically equivalent CK matrices $\mathbf{K}_{\mathrm{CK},\ell-1}$ at layer $\ell-1$ with the *same* $\alpha_{\ell-1,1}, \alpha_{\ell-1,2}$ and $\alpha_{\ell-1,3}$, then its follows from Equation (10) and (11) that having the *same* $\mathbb{E}[\sigma'_\ell(\tau_{\ell-1}\xi)]^2$, $\mathbb{E}[\sigma''_\ell(\tau_{\ell-1}\xi)]^2$, and $\mathbb{E}[(\sigma_\ell^2(\tau_{\ell-1}\xi))'']$ (which *only* depends on the activation $\sigma_\ell$ of layer $\ell$ and $\tau_{\ell-1}$) suffices for the two nets to have asymptotically equivalent $\mathbf{K}_{\mathrm{CK},\ell}$ at layer $\ell$.

It follows from the above item (ii) that for a given DNN of depth $L$, it is possible to design a novel DNN model that "matches" the original one – in the sense that both models will have asymptotically equivalent CK matrices *at each layer*, by using the following layer-by-layer matching strategy: Starting from the same $\mathbf{K}_{\mathrm{CK},0} = \mathbf{X}^\mathsf{T}\mathbf{X}$, one chooses the first-layer weights $\mathbf{W}_1$ of the novel DNN according to Assumption 2, and then select the first-layer activation $\sigma_1$ in such a way that the novel net has the same parameters $\alpha_{1,1}, \alpha_{1,2}$ and $\alpha_{1,3}$ as the original one, so that the first-layer CK matrices $\mathbf{K}_{\mathrm{CK},1}$ of the two nets are *spectrally* matched as per Theorem 1; one then proceeds similarly to match the second, the third, etc., and eventually the $L$th layer of the two nets. As we shall see below, this layer-by-layer matching strategy facilitates the "lossless" compression of a given DNN.

Using the relation in (6), a similar result (as in Theorem 1 for CKs) can be established for NTK matrices, as shown in the following theorem. The proof is also based on an induction on $\ell$ and is provided in Section A.2 of the appendix.

**Theorem 2** (Asymptotic spectral equivalent for NTK matrices). *Let Assumptions 1–3 hold, let the activations $\sigma_\ell(\cdot)$s be centered so that $\mathbb{E}[\sigma_\ell(\tau_{\ell-1}\xi)] = 0$ for $\tau_\ell$s defined in (7), let $\dot{\tau}_0, \dot{\tau}_1, \ldots, \dot{\tau}_L \geq 0$ be a sequence of non-negative numbers satisfying the following recursion:*

$$\dot{\tau}_\ell = \sqrt{\mathbb{E}\left[(\sigma'_\ell(\dot{\tau}_{\ell-1}\xi))^2\right]}, \quad \xi \sim \mathcal{N}(0,1), \quad \ell \in \{1, \ldots, L\}, \tag{12}$$

*with $\dot{\tau}_0 = \tau_0$ (which is similar to the $\tau_\ell$s defined in (7), but on the derivative $\sigma'_\ell$ instead of $\sigma_\ell$ itself), and let $\kappa_1, \ldots, \kappa_L \geq 0$ be a sequence of non-negative numbers satisfying*

$$\kappa_\ell = \sqrt{\tau_\ell^2 + \kappa_{\ell-1}^2(\dot{\tau}_\ell)^2}, \quad \xi \sim \mathcal{N}(0,1), \quad \ell \in \{1, \ldots, L\}, \tag{13}$$

*with $\kappa_0 = \tau_0$. Then, for the NTK matrix $\mathbf{K}_{\mathrm{NTK},\ell}$ of layer $\ell$ defined in (6), as $n, p \to \infty$, one has that*

$$\|\mathbf{K}_{\mathrm{NTK},\ell} - \tilde{\mathbf{K}}_{\mathrm{NTK},\ell}\| \to 0, \quad \tilde{\mathbf{K}}_{\mathrm{NTK},\ell} \equiv \beta_{\ell,1}\mathbf{X}^\mathsf{T}\mathbf{X} + \mathbf{V}\mathbf{B}_\ell\mathbf{V}^\mathsf{T} + (\kappa_\ell^2 - \tau_0^2\beta_{\ell,1} - \tau_0^4\beta_{\ell,3})\mathbf{I}_n, \tag{14}$$

*almost surely, with $\mathbf{V} \in \mathbb{R}^{n \times (K+1)}, \mathbf{t} \in \mathbb{R}^K, \mathbf{T} \in \mathbb{R}^{K \times K}$ as defined in Theorem 1, and*

$$\mathbf{B}_\ell \equiv \begin{bmatrix} \beta_{\ell,2}\mathbf{t}\mathbf{t}^\mathsf{T} + \beta_{\ell,3}\mathbf{T} & \beta_{\ell,2}\mathbf{t} \\ \beta_{\ell,2}\mathbf{t}^\mathsf{T} & \beta_{\ell,2} \end{bmatrix} \in \mathbb{R}^{(K+1) \times (K+1)}, \tag{15}$$

*as well as scalars $\beta_{\ell,1}, \beta_{\ell,2}, \beta_{\ell,3} \geq 0$, and $\dot{d}_\ell, \dot{\alpha}_{\ell,1}$ such that*

$$\beta_{\ell,1} = \alpha_{\ell,1} + \beta_{\ell-1,1}\dot{d}_\ell, \quad \beta_{\ell,2} = \alpha_{\ell,2} + \beta_{\ell-1,2}\dot{d}_\ell, \quad \beta_{\ell,3} = \alpha_{\ell,3} + \beta_{\ell-1,3}\dot{d}_\ell + \beta_{\ell-1,1}\dot{\alpha}_{\ell,1}, \tag{16}$$

---

[2]It is worth noting that the parameter $\tau_\ell$ appears in the CK eigenspectrum *only* by shifting all its eigenvalues (by $\tau_\ell^2$), thereby acting as an (implicit) ridge-type regularization in DNN models, see also [29, 15, 39].

*and*

$$\dot{d}_\ell = \mathbb{E}_{\xi_1,\xi_2}\left[\sigma'_\ell(\dot{\tau}_{\ell-1}\xi_1)\sigma'_\ell\left(\frac{\dot{d}_{\ell-1}}{\dot{\tau}_{\ell-1}}\xi_1 + \sqrt{\dot{\lambda}_{\ell-1}}\xi_2\right)\right],$$

$$\dot{\alpha}_{\ell,1} = \mathbb{E}_{\xi_1,\xi_2}\left[\sigma'_\ell(\dot{\tau}_{\ell-1}\xi_1)\sigma''_\ell\left(\frac{\dot{d}_{\ell-1}}{\dot{\tau}_{\ell-1}}\xi_1 + \sqrt{\dot{\lambda}_{\ell-1}}\xi_2\right)\left(\frac{\dot{\alpha}_{\ell-1,1}}{\dot{\tau}_{\ell-1}}\xi_1 - \frac{1}{\sqrt{\dot{\lambda}_{\ell-1}}}\frac{\dot{d}_{\ell-1}\dot{\alpha}_{\ell-1,1}}{\dot{\tau}_{\ell-1}^2}\xi_2\right)\right],$$

*with* $\dot{\lambda}_{\ell-1} = \dot{\tau}_{\ell-1}^2 - \frac{\dot{d}_{\ell-1}^2}{\dot{\tau}_{\ell-1}^2}$ *for independent* $\xi_1, \xi_2 \sim \mathcal{N}(0,1)$.

Roughly speaking, Theorem 2 shows that the eigenspectral behavior established in Theorem 1 for CK matrices also holds for NTK matrices, up to a change of the associated coefficients $\alpha_\ell$s to $\beta_\ell$s. The remarks after Theorem 1 thus remain valid, at least in spirit, for NTK matrices.

**Remark 2** (On spectral norm characterization). *Note that the characterizations in Theorem 1 and 2 for CK and NTK matrices are provided in a spectral norm sense. It then follows from Weyl's inequality [24] and the Davis–Kahan theorem [61] that the difference between the eigenvalues (e.g., when listed in a decreasing order) and the associated eigenvectors (when the eigenvalues under study are "isolated") of* $\mathbf{K}_{\mathrm{NTK}}$ *and* $\tilde{\mathbf{K}}_{\mathrm{NTK}}$ *vanish asymptotically as* $n, p \to \infty$. *As such, the spectral norm guarantees in Theorem 1 and 2 provide more tractable access to the convergence and generalization properties of wide DNNs, at least for GMM data, via the spectral study of* $\tilde{\mathbf{K}}_{\mathrm{NTK}}$ *[28, 18].*

Despite being derived here for the Gaussian mixture model in (1), we conjecture that the results in Theorem 1 and 2 hold beyond the Gaussian setting and extend, for example, to the family of concentrated random vectors [50, 33]. As discussed after Assumption 2 for the distribution of $\mathbf{W}$, such universality commonly arises in random matrix theory and high-dimensional statistics [12, 6, 56]; we refer interested readers to Remark 4 in Appendix A for further discussions.

From a technical perspective, the results in Theorem 1 and 2 extend the single-hidden-layer CK analysis in [4, 36] to both CK and NTK matrices of fully-connected DNNs with an arbitrary number of layers. In particular, taking $\ell = 1$ in Theorem 1, one obtains [4, Theorem 1] as a special case.[3]

**Remark 3** (On CK and NTK matrices). *It follows from Theorem 1 that for a given DNN model, it suffices to match the coefficients* $\alpha_{\ell,1}, \alpha_{\ell,2}$ *and* $\alpha_{\ell,3}$ *in a layer-by-layer manner to propose a novel DNN with asymptotically equivalent CK matrices. Similarly, matching the key coefficients* $\beta_{\ell,1}, \beta_{\ell,2}$, *and* $\beta_{\ell,3}$ *leads to a DNN model with asymptotically equivalent NTK matrices. Moreover, for two nets with the same* $\alpha_{\ell,1}, \alpha_{\ell,2}, \alpha_{\ell,3}$, *and therefore the same CKs, it suffices to match the additional* $\dot{d}_\ell, \dot{\alpha}_\ell$ *to render the two nets asymptotically equivalent in both CK and NTK senses.*

In the following corollary, we present a concrete example of how to apply the results in Theorem 1 and 2 in the design of a novel computationally and storage efficient DNN, that shares the same CK and NTK eigenspectra with any given fully-connected neural net having centered activation.

**Corollary 1** (Sparse and quantized DNNs). *For a given fully-connected DNN (referred to as* DNN1*) of depth* $L$ *with centered activation such that* $\mathbb{E}[\sigma_\ell(\tau_{\ell-1}\xi)] = 0$ *for* $\xi \sim \mathcal{N}(0,1)$, *one is able to construct, say in a layer-by-layer manner, a sparse and quantized "equivalent" DNN model, of depth* $L$ *and referred to as* DNN2*, such that the two nets have asymptotically the same eigenspectra for their CK (and NTK similarly, as per Remark 3) matrices, by using the following ternary weights:*

$$[\mathbf{W}]_{ij} = 0 \text{ with proba } \varepsilon \in [0,1), \quad [\mathbf{W}]_{ij} = \pm(1-\varepsilon)^{-1/2} \text{ each with proba } 1/2 - \varepsilon/2, \quad (17)$$

*as well as quantized activations (as visually displayed in Figure 1):*

$$\sigma_T(t) = a \cdot (1_{t<s_1} + 1_{t>s_2}), \quad \sigma_Q(t) = b_1 \cdot (1_{t<r_1} + 1_{t>r_4}) + b_2 \cdot 1_{r_2 \le t \le r_3}. \quad (18)$$

We refer readers to Appendix A.4 for the proof and discussions of Corollary 1, as well as the detailed expressions of $\mathbb{E}[\sigma'(\tau\xi)]$, $\mathbb{E}[\sigma''(\tau\xi)]$, and $\mathbb{E}[(\sigma^2(\tau\xi))'']$, of direct algorithmic use for both $\sigma_T$ and $\sigma_Q$ as functions of the parameters $a, s_1, s_2$ and $b_1, b_2, r_1, r_2, r_3, r_4$. Built upon Corollary 1, we propose a DNN "lossless" compression scheme with equivalent CKs, as summarized in Algorithm 1 below.

---

[3]Note that in [4, Theorem 1], the authors do *not* assume $\mathbb{E}[\sigma(\tau_0\xi)] = 0$ as in our Theorem 1, but instead "center" the CK matrices by pre- and post-multiplying $\mathbf{K}_{\mathrm{CK}}$ with $\mathbf{P} = \mathbf{I}_n - \mathbf{1}_n\mathbf{1}_n^\mathsf{T}/n$. This can be shown equivalent to taking $\mathbb{E}[\sigma(\tau_0\xi)] = 0$ in the single-hidden-layer setting; see Appendix A.3 for more details.

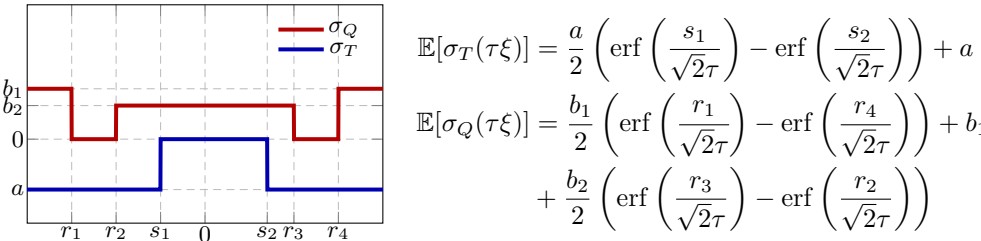

$$\mathbb{E}[\sigma_T(\tau\xi)] = \frac{a}{2}\left(\mathrm{erf}\left(\frac{s_1}{\sqrt{2}\tau}\right) - \mathrm{erf}\left(\frac{s_2}{\sqrt{2}\tau}\right)\right) + a$$

$$\mathbb{E}[\sigma_Q(\tau\xi)] = \frac{b_1}{2}\left(\mathrm{erf}\left(\frac{r_1}{\sqrt{2}\tau}\right) - \mathrm{erf}\left(\frac{r_4}{\sqrt{2}\tau}\right)\right) + b_1$$

$$+ \frac{b_2}{2}\left(\mathrm{erf}\left(\frac{r_3}{\sqrt{2}\tau}\right) - \mathrm{erf}\left(\frac{r_2}{\sqrt{2}\tau}\right)\right)$$

Figure 1: Visual representations of activations $\sigma_T$ and $\sigma_Q$ in (18) **(left)** and the expressions of $\mathbb{E}[\sigma_T(\tau\xi)]$ and $\mathbb{E}[\sigma_Q(\tau\xi)]$**(right)**, with $r_1 - r_2 = r_3 - r_4$ here and $\mathrm{erf}(\cdot)$ the Gaussian error function.

---

**Algorithm 1** "Lossless" compression scheme for fully-connected DNNs
---
1: **Input:** Input data $\mathbf{x}_1, \ldots, \mathbf{x}_n$, sparsity level $\varepsilon \in [0, 1)$, and DNN1 with activations $\sigma_1, \ldots, \sigma_L$.
2: **Output:** Sparse and quantized model DNN2 with weights $\mathbf{W}_\ell$ and activations $\tilde{\sigma}_\ell, \ell \in \{1, \ldots, L\}$.

3: Estimate $\tau_0$ from data as $\tau_0 = \sqrt{\frac{1}{n}\sum_{i=1}^n \|\mathbf{x}_i\|^2}$. Set $\tau = \tau_0$ for DNN1, and $\tilde{\tau} = \tau_0$ for DNN2.
4: **for** $\ell = 1, \ldots, L-1$ **do**
5:     Compute $\alpha_{\ell,1}, \alpha_{\ell,2}, \alpha_{\ell,3}$ of DNN1 using $\tau_{\ell-1}$, and derive the expressions of $\tilde{\alpha}_{\ell,1}, \tilde{\alpha}_{\ell,2}, \tilde{\alpha}_{\ell,3}$ of DNN2 using $\tilde{\tau}_{\ell-1}$ as per (10) and (11).
6:     Solve, with Corollary 1 and the detailed expressions in Appendix A.4, the system of equations $(\alpha_{\ell,1}, \alpha_{\ell,2}, \alpha_{\ell,3}) = (\tilde{\alpha}_{\ell,1}, \tilde{\alpha}_{\ell,2}, \tilde{\alpha}_{\ell,3})$ for the parameters $a, s_1, s_2$, to get the activation $\tilde{\sigma}_\ell$ of DNN2 at layer $\ell$.
7:     Update $\tau, \tilde{\tau}$ as $\tau = \sqrt{\mathbb{E}[\sigma_\ell^2(\tau\xi)]}$, and $\tilde{\tau} = \sqrt{\mathbb{E}[\tilde{\sigma}_\ell^2(\tilde{\tau}\xi)]}$.
8: **end for**
9: For the layer $\ell = L$, compute $\alpha_{L,1}, \alpha_{L,2}, \alpha_{L,3}, \tau_L$ and $\tilde{\alpha}_{L,1}, \tilde{\alpha}_{L,2}, \tilde{\alpha}_{L,3}, \tilde{\tau}_L$. Use them to solve for the parameters $b_1, b_2, r_1, r_2, r_3, r_4$ to obtain the activation $\tilde{\sigma}_L$ of DNN2 at layer $\ell$.
10: Draw independently the i.i.d. entries of $\mathbf{W}_1, \ldots, \mathbf{W}_L$ according to (17) with sparsity level $\varepsilon$.
11: **return** DNN2 model with weights $\mathbf{W}_\ell$ and activations $\tilde{\sigma}_\ell, \ell = 1, \ldots, L$.

---

As a side remark, note that the "sign" of activations does not matter in Corollary 1 or Algorithm 1, in the sense that the key parameters $\alpha_{\ell,1}, \alpha_{\ell,2},$ and $\alpha_{\ell,3}$ for CKs, as well as $\beta_{\ell,1}, \beta_{\ell,2},$ and $\beta_{\ell,3}$ for NTKs, remain unchanged when $-\sigma_\ell(t)$ is used instead of $\sigma_\ell(t)$.

Before embarking on the detailed numerical experiments in Section 4, we would like to bring the readers' attention to the recent line of works [35, 53, 57, 54, 19] showing that for wide and deep NN models, very efficient sparse sub-networks can be found that almost match the performance of the original dense nets *with little or even no training*, for instance by uniformly pruning the network weights [53]. To develop a theoretical grasp of these (extremely counterintuitive) empirical successes, a few attempts have been made, for example, to carefully prune the network weights to retain the same (limiting) NTK [40], or to show that randomly pruned sparse nets have the same (limiting) NTK as the original net up to a scaling factor [60]. Instead, our work, by considering the *statistical structure* of the input data, leverages tools from RMT to "compress" both the weights and activations (per Theorem 2 and Corollary 1) without affecting the NTK eigenstructure.

## 4 Numerical experiments

In this section, we provide numerical experiments to (i) validate the asymptotic characterizations in Theorem 1 and 2, on both synthetic GMM and real-world data (such as MNIST and CIFAR10) of (in fact not so) large sizes and dimensions; and to (ii) show how these results can be used to sparsify and quantize fully-connected DNNs, leading to huge savings in computational and storage resources (up to a factor of $10^3$ in memory and a level of sparsity $\varepsilon = 90\%$) without significant performance degradation. We refer readers to Section B in the appendix for further experiments and discussions.

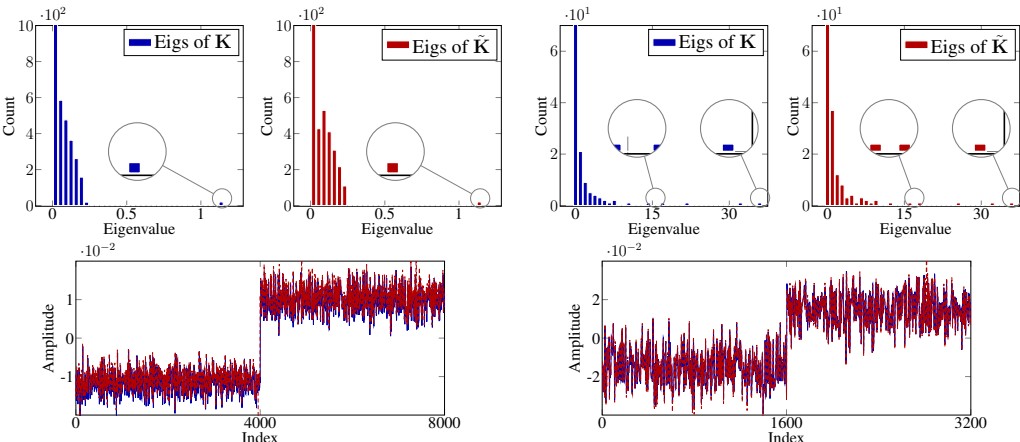

Figure 2: Eigenvalue histograms (**top**) and dominant eigenvectors (**bottom**) of last-layer CK matrices $\mathbf{K}_{CK}$ (**blue**) defined in (4) (with expectation estimated from $1\,000$ independent realizations of $\mathbf{W}$s) and the asymptotic equivalent $\tilde{\mathbf{K}}_{CK}$ (**red**) matrices. (**Left**) Gaussian $\mathbf{W}$ on two-class GMM data, with $p = 2\,000$, $n = 8\,000$, $\boldsymbol{\mu}_a = [\mathbf{0}_{8(a-1)};\ 8;\ \mathbf{0}_{p-8a+7}]$, $\mathbf{C}_a = (1 + 8(a-1)/\sqrt{p})\mathbf{I}_p$, $a \in \{1, 2\}$ using [ReLU, ReLU, ReLU] activations, here $\|\mathbf{K}_{CK} - \tilde{\mathbf{K}}_{CK}\| = 0.15$; and (**right**) symmetric Bernoulli $\mathbf{W}$ on MNIST data (number 6 versus 8) [31], with $p = 784$, $n = 3\,200$, using [poly, ReLU, ReLU] activations, $\|\mathbf{K}_{CK} - \tilde{\mathbf{K}}_{CK}\| = 6.86$. $\mathbf{x}_1, \ldots, \mathbf{x}_{n/2} \in \mathcal{C}_1$ and $\mathbf{x}_{n/2+1}, \ldots, \mathbf{x}_n \in \mathcal{C}_2$ in both cases.

Figure 2 compares the eigenvalues and dominant eigenvectors of the CK matrices $\mathbf{K}_{CK}$ defined in (4) versus those of their asymptotic approximations $\tilde{\mathbf{K}}_{CK}$ given in Theorem 1, in the case of fully-connected DNNs having three hidden layers (of width $d_1 = 2\,000$, $d_2 = 2\,000$, $d_3 = 1\,000$). For different types of activations: $\mathrm{poly}(t) = 0.2t^2 + t$ and $\mathrm{ReLU}(t) = \max(t, 0)$, different weight distributions (Gaussian and symmetric Bernoulli), and on synthetic GMM as well as MNIST data, we consistently observe a close match between the eigenvalues and dominant eigenvectors of $\mathbf{K}_{CK}$ and $\tilde{\mathbf{K}}_{CK}$, as a consequence of the spectral norm convergence in Theorem 1 (and Remark 2), suggesting a possibly wider applicability of the proposed results beyond GMM data.[4]

Following the idea of CK matching in Figure 2, Figure 3 depicts the test accuracies of (i) the original dense and unquantized network with three fully-connected layers, (ii) the proposed "lossless" compression scheme described in Corollary 1 and Algorithm 1 via CK matching, as well as its variant having ternarized weights but *dense and unquantized* activations, (iii) the popular magnitude-based pruning approach as in [20], together with (iv) two "heuristic" compression approaches: (iv-i) sparsification by uniformly zeroing out $80\%$ of the weights (we *cannot* do more, as the resultant performance is too poor to be visually compared with other curves), and (iv-ii) binarization using $\sigma(t) = 1_{t<-1} + 1_{t>1}$, for different choices of width per layer, and the ten-class classification problems of MNIST and CIFAR10. Specifically, neural networks before and after compression have three fully-connected layers, and the original network uses ReLU activation in each layer. Classification is performed on a concatenated and trainable fully-connected layer. For MNIST datasets, raw data are taken as the network input; for CIFAR10 dataset, flattened output of the 16th convolutional layer of VGG19 [52] are taken as the network input.

We observe from Figure 3 that the proposed "lossless" compression scheme produces significantly sparser networks (up to $90\%$ of weights set to zero) with minimal performance loss, while occupying (up to) a factor of $10^3$ less memory, when compared to the original or the heuristically compressed nets. Also, higher accuracies are obtained with the proposed approach than, e.g., the popular magnitude-based pruning under the same memory budget. In addition, the ternary weights variant (with *quantized* weights *only*) of the proposed scheme can achieve even better performance (with virtually no performance loss compared to the original dense net), however at the price of not conducive to inference accelerating, since the activations are *unquantized*. We also see that the sparsity level $\varepsilon$ has limited impact on the classification accuracy, as in line with our theory.

---

[4]Small eigenvalues of $\mathbf{K}_{CK}, \tilde{\mathbf{K}}_{CK}$ close to zero are removed from Figure 2 for better visualization.

These experimental results show that the proposed approach achieves a better *performance-complexity trade-off* than commonly used heuristic DNN compression methods.

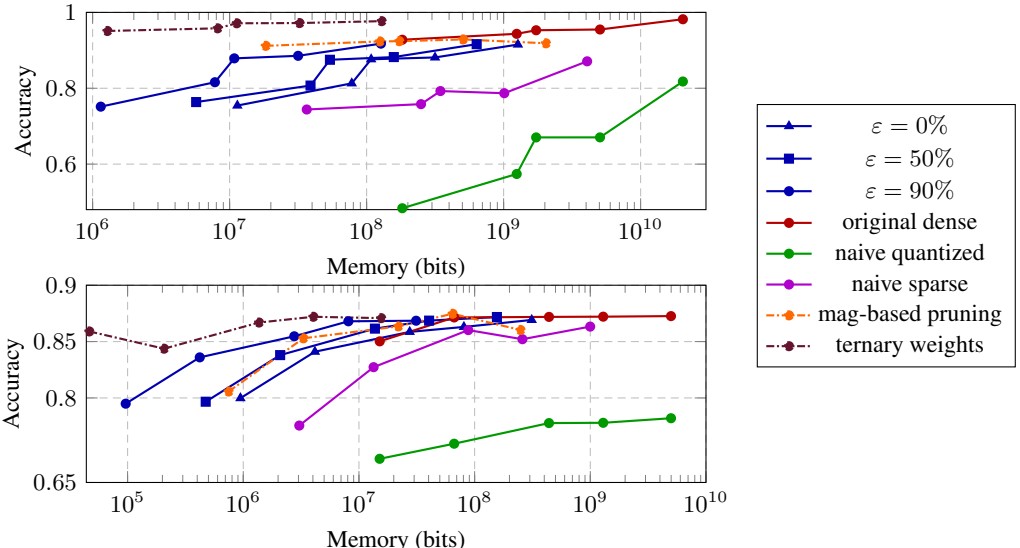

Figure 3: Classification accuracies of different compressed fully-connected nets on MNIST [31] (**top**) and CIFAR10 [30] (**bottom**) datasets. **Blue** curves represent the proposed compression approach with different levels of sparsity $\varepsilon \in \{0\%, 50\%, 90\%\}$, **purple** curves represent the heuristic sparsification approach by uniformly zeroing out $80\%$ of the weights, **green** curves represent the heuristic quantization approach using the binary activation $\sigma(t) = 1_{t<-1} + 1_{t>1}$, **red** curves represent the original network, **brown** curves represent the proposed compression approach *without* activation quantization, with $\varepsilon = 90\%$ for MNIST (**top**) and $\varepsilon = 95\%$ for CIFAR10 (**bottom**), and **orange** curves represent magnitude-based pruning [20] with the same sparsity level $\varepsilon$ as **brown**. Memory varies due to the **change of layer width** of the network.

## 5  Conclusion and perspectives

In this paper, built upon recent advances in random matrix theory and high-dimensional statistics, we provide *precise* characterizations of the eigenspectra of both conjugate kernel and neural tangent kernel matrices, for high-dimensional Gaussian mixture data and fully-connected multi-layer neural nets. These results further allows us to sparsify and quantize fully-connected deep nets, resulting in a factor of $10^3$ less memory consumption with virtually no performance degradation.

Extending the present theoretical framework to more involved settings such as convolutional nets requires refined analysis on the block Toeplitz weights $\mathbf{W}$ and on their connection to the corresponding CK and NTK matrices [5, 59]. Also, since the NTK eigenspectra determine the gradient descent dynamics of ultra-wide deep nets, the asymptotic characterizations in Theorem 2 can be applied to assess the learning dynamics [37, 2, 2] of fully-connected DNN models, in and possibly beyond the infinitely wide NTK regime.

## Acknowledgments and Disclosure of Funding

ZL would like to acknowledge the CCF-Hikvision Open Fund (20210008), the National Natural Science Foundation of China (via fund NSFC-62206101 and NSFC-12141107), the Fundamental Research Funds for the Central Universities of China (2021XXJS110), the Key Research and Development Program of Hubei (2021BAA037), and the Key Research and Development Program of Guangxi (GuiKe-AB21196034) for providing partial support.

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
