# Supplementary Material

### "Lossless" Compression of Deep Neural Networks:
### A High-dimensional Neural Tangent Kernel Approach

## A   Proofs of theorems and auxiliary results

### A.1   Proof of Theorem 1

In this section, we provide the detailed proof of Theorem 1. Before going into details of the proof, we first recall our system model and working assumptions as follows.

We consider $n$ data vectors $\mathbf{x}_1, \ldots, \mathbf{x}_n \in \mathbb{R}^p$ independently drawn from one of the $K$-class Gaussian mixtures $\mathcal{C}_1, \ldots, \mathcal{C}_K$ and denote $\mathbf{X} = [\mathbf{x}_1, \ldots, \mathbf{x}_n] \in \mathbb{R}^{p \times n}$, with class $\mathcal{C}_a$ having cardinality $n_a$; that is, for $\mathbf{x}_i \in \mathcal{C}_a$ we have

$$\mathbf{x}_i \sim \mathcal{N}(\boldsymbol{\mu}_a/\sqrt{p}, \mathbf{C}_a/p), \tag{19}$$

for mean vector $\boldsymbol{\mu}_a \in \mathbb{R}^p$ and non-negative definite covariance $\mathbf{C}_a \in \mathbb{R}^{p \times p}$ associated with class $\mathcal{C}_a$.

We position ourselves in the high-dimensional and non-trivial classification regime as stated in Assumption 1, that is: As $n \to \infty$, we have, for $a \in \{1, \ldots, K\}$ that,

(i)  $p/n \to c \in (0, \infty)$ and $n_a/n \to c_a \in (0, 1)$; and

(ii)  $\|\boldsymbol{\mu}_a\| = O(1)$; and

(iii)  for $\mathbf{C}^\circ \equiv \sum_{a=1}^K \frac{n_a}{n} \mathbf{C}_a$ and $\mathbf{C}_a^\circ \equiv \mathbf{C}_a - \mathbf{C}^\circ$, we have $\|\mathbf{C}_a\| = O(1)$, $\operatorname{tr} \mathbf{C}_a^\circ = O(\sqrt{p})$ and $\operatorname{tr}(\mathbf{C}_a \mathbf{C}_b) = O(p)$ for $a, b \in \{1, \ldots, K\}$; and

(iv)  $\tau_0 \equiv \sqrt{\operatorname{tr} \mathbf{C}^\circ / p}$ converges in $(0, \infty)$.

**Remark 4** (Beyond Gaussian mixture data). *Despite derived here for Gaussian mixture data, we conjecture that our results hold more generally beyond the Gaussian setting. As concrete examples, many results in random matrix theory and high dimensional statistics such as the popular Marčenko-Pastur law [43], the semicircular law [58], as well as the circular laws [55], have all been shown* universal *in the sense that they do* not *depend on the distribution of the (independent entries of the) data, as long as they are normalized to have zero mean and unit variance. In a machine learning (ML) context, such* universal *behavior are observed to hold beyond the above models, and extends to* nonlinear *models such as kernel matrices [51] and neural nets [50], in that, say, for data drawn from the family of concentrated random vectors [33, 41] (so* not *necessarily Gaussian), the performance on those ML models are the* same*, in the larger $n, p$ setting, as if they were mere Gaussian mixtures with the same means and covariances. We refer the interested readers to [12, Chapter 8] for more discussions on this point.*

We consider the fully-connected neural network model of depth $L$ and of successive widths $d_1, \ldots, d_L$ as defined in (2), and denote $\mathbf{W}_\ell \in \mathbb{R}^{d_\ell \times d_{\ell-1}}$ as well as $\sigma_\ell(\cdot)$ the weight matrix and activation at layer $\ell \in \{1, \ldots, L\}$, respectively.

We assume that the following conditions hold for the random weight matrices $\mathbf{W}_\ell$s and the activation function $\sigma_\ell$s for $\ell \in \{1, \ldots, L\}$, as demanded in Assumption 2 and 3:

(i)  The weight matrices $\mathbf{W}_\ell$s are independent and have i.i.d. entries of zero mean, unit variance, and finite fourth-order moment.

(ii)  The activations $\sigma_\ell$s are at least four-times differentiable with respect to standard normal distribution, in the sense that $\max_{k \in \{0,1,2,3,4\}} \{|\mathbb{E}[\sigma_\ell^{(k)}(\xi)]|\} < C$ for some universal constant $C > 0$ and $\xi \sim \mathcal{N}(0, 1)$.

In this section, we focus on the Conjugate Kernel (CK) matrix defined via the following recursive relation [28, 8]

$$[\mathbf{K}_{\mathrm{CK},\ell}]_{ij} = \mathbb{E}_{u,v}[\sigma_\ell(u)\sigma_\ell(v)], \quad \mathbf{K}_{\mathrm{CK},0} = \mathbf{X}^\mathsf{T}\mathbf{X}, \tag{20}$$

with

$$u, v \sim \mathcal{N}\left(\mathbf{0}, \begin{bmatrix} [\mathbf{K}_{\mathrm{CK},\ell-1}]_{ii} & [\mathbf{K}_{\mathrm{CK},\ell-1}]_{ij} \\ [\mathbf{K}_{\mathrm{CK},\ell-1}]_{ij} & [\mathbf{K}_{\mathrm{CK},\ell-1}]_{jj} \end{bmatrix}\right). \tag{21}$$

The derivation and discussion of the closely related neural tangent kernel (NTK) matrix is given in Section A.2.

Let Assumptions 1–3 hold, and let $\tau_0, \tau_1, \ldots, \tau_L \geq 0$ be a sequence of non-negative numbers recursively defined via

$$\tau_\ell = \sqrt{\mathbb{E}[\sigma_\ell^2(\tau_{\ell-1}\xi)]}, \tag{22}$$

as in (7). Further assume that the activation functions $\sigma_\ell(\cdot)$s are "centered" such that $\mathbb{E}[\sigma_\ell(\tau_{\ell-1}\xi)] = 0$. (This assumption, as we shall see, plays a central role in our proof.)

Then, to prove Theorem 1 it suffices to show that,

(i) the CK matrix $\mathbf{K}_{\mathrm{CK},\ell}$ of layer $\ell \in \{0, 1, \ldots, L\}$ defined in (4) satisfies

$$\|\mathbf{K}_{\mathrm{CK},\ell} - \tilde{\mathbf{K}}_{\mathrm{CK},\ell}\| \to 0, \tag{23}$$

almost surely as $n, p \to \infty$, with $\tilde{\mathbf{K}}_{\mathrm{CK},\ell}$ taking the "universal" form

$$\tilde{\mathbf{K}}_{\mathrm{CK},\ell} \equiv \alpha_{\ell,1}\mathbf{X}^\mathsf{T}\mathbf{X} + \mathbf{V}\mathbf{A}_\ell\mathbf{V}^\mathsf{T} + (\tau_\ell^2 - \tau_0^2\alpha_{\ell,1} - \tau_0^4\alpha_{\ell,3})\mathbf{I}_n, \tag{24}$$

for all $\ell \in \{1 \ldots, L\}$, $\mathbf{J} = [\mathbf{j}_1, \ldots, \mathbf{j}_K] \in \mathbb{R}^{n \times K}$, random vector $\boldsymbol{\psi} = \{\|\mathbf{x}_i - \mathbb{E}[\mathbf{x}_i]\|^2 - \mathbb{E}[\|\mathbf{x}_i - \mathbb{E}[\mathbf{x}_i]\|^2]\}_{i=1}^n \in \mathbb{R}^n$, $\mathbf{t} = \{\operatorname{tr}\mathbf{C}_a^\circ/\sqrt{p}\}_{a=1}^K \in \mathbb{R}^K$, $\mathbf{T} = \{\operatorname{tr}\mathbf{C}_a\mathbf{C}_b/p\}_{a,b=1}^K \in \mathbb{R}^{K \times K}$, and

$$\mathbf{V} = [\mathbf{J}/\sqrt{p}, \ \boldsymbol{\psi}] \in \mathbb{R}^{n \times (K+1)}, \quad \mathbf{A}_\ell = \begin{bmatrix} \alpha_{\ell,2}\mathbf{t}\mathbf{t}^\mathsf{T} + \alpha_{\ell,3}\mathbf{T} & \alpha_{\ell,2}\mathbf{t} \\ \alpha_{\ell,2}\mathbf{t}^\mathsf{T} & \alpha_{\ell,2} \end{bmatrix} \in \mathbb{R}^{(K+1) \times (K+1)}; \tag{25}$$

(ii) the coefficients $\alpha_{\ell,1}, \alpha_{\ell,2}, \alpha_{\ell,3}$ are non-negative and satisfy the following recursive relations

$$\alpha_{\ell,1} = \mathbb{E}[\sigma_\ell'(\tau_{\ell-1}\xi)]^2\alpha_{\ell-1,1}, \quad \alpha_{\ell,2} = \mathbb{E}[\sigma_\ell'(\tau_{\ell-1}\xi)]^2\alpha_{\ell-1,2} + \frac{1}{4}\mathbb{E}[\sigma_\ell''(\tau_{\ell-1}\xi)]^2\alpha_{\ell-1,4}^2,$$

$$\alpha_{\ell,3} = \mathbb{E}[\sigma_\ell'(\tau_{\ell-1}\xi)]^2\alpha_{\ell-1,3} + \frac{1}{2}\mathbb{E}[\sigma_\ell''(\tau_{\ell-1}\xi)]^2\alpha_{\ell-1,1}^2.$$

with $\alpha_{\ell,4} = \alpha_{\ell-1,4}\mathbb{E}\left[(\sigma_\ell'(\tau_{\ell-1}\xi))^2 + \sigma_\ell(\tau_{\ell-1}\xi)\sigma_\ell''(\tau_{\ell-1}\xi)\right]$ for $\xi \sim \mathcal{N}(0,1)$.

We will prove the above results by induction on $\ell \in \{0, 1, \ldots, L\}$.

We first introduce the following notations that will be consistently used in the proof: for $\mathbf{x}_i, \mathbf{x}_j \in \mathbb{R}^p$ with $i \neq j$, let

$$\mathbf{x}_i = \boldsymbol{\mu}_i/\sqrt{p} + \mathbf{z}_i/\sqrt{p}, \quad \mathbf{x}_j = \boldsymbol{\mu}_j/\sqrt{p} + \mathbf{z}_j/\sqrt{p}, \tag{26}$$

so that $\mathbf{z}_i \sim \mathcal{N}(\mathbf{0}, \mathbf{C}_i)$, $\mathbf{z}_j \sim \mathcal{N}(\mathbf{0}, \mathbf{C}_j)$, and

$$A_{ij} \equiv \underbrace{\frac{1}{p}\mathbf{z}_i^\mathsf{T}\mathbf{z}_j}_{O(p^{-1/2})} + \underbrace{\frac{1}{p}\boldsymbol{\mu}_i^\mathsf{T}\boldsymbol{\mu}_j + \frac{1}{p}(\boldsymbol{\mu}_i^\mathsf{T}\mathbf{z}_j + \boldsymbol{\mu}_j^\mathsf{T}\mathbf{z}_i)}_{O(p^{-1})},$$

$$t_i \equiv \frac{1}{p}\operatorname{tr}\mathbf{C}_i^\circ = O(p^{-1/2}), \quad \psi_i = \frac{1}{p}\|\mathbf{z}_i\|^2 - \frac{1}{p}\operatorname{tr}\mathbf{C}_i = O(p^{-1/2}),$$

$$\tau_0 \equiv \sqrt{\frac{1}{p}\operatorname{tr}\mathbf{C}^\circ} = O(1),$$

$$\chi_i \equiv \underbrace{t_i + \psi_i}_{O(p^{-1/2})} + \underbrace{\|\boldsymbol{\mu}_i\|^2/p + 2\boldsymbol{\mu}_i^\mathsf{T}\mathbf{z}_i/p}_{O(p^{-1})} = \|\mathbf{x}_i\|^2 - \tau_0^2,$$

where we note that the notations $\tau_0, \psi_i$ and $t_i$ (with a slight abuse of notation to denote $\mathbf{C}_i = \mathbf{C}_a$ for $\mathbf{x}_i \in \mathcal{C}_a$) are in line with those defined in Assumption 1 and Theorem 1, and we denote $S_{ij}$ terms of the form

$$S_{ij} \equiv S_{ij}(\gamma_1, \gamma_2) = \frac{1}{p}\mathbf{z}_i^\mathsf{T}\mathbf{z}_j\left(\gamma_1(t_i + \psi_i) + \gamma_2(t_j + \psi_j)\right), \tag{27}$$

for random or deterministic scalars $\gamma_1, \gamma_2 = O(1)$ (with high probability when being random). We have $S_{ij} = O(p^{-1})$ and more importantly, it leads to, in matrix form, a matrix of spectral norm order $O(p^{-1/2})$, see [11]. This spectral norm result will be exploited in the remainder of the proof.

Then we give the process of induction as follows.

For $\ell = 0$, we have $\mathbf{K}_{\mathrm{CK},0} = \mathbf{X}^{\mathsf{T}}\mathbf{X}$ so that

$$\mathbf{K}_{\mathrm{CK},0} = \tilde{\mathbf{K}}_{\mathrm{CK},0} = \mathbf{X}^{\mathsf{T}}\mathbf{X}, \tag{28}$$

with $\alpha_{0,1} = 1$, $\alpha_{0,2} = 0$, and $\alpha_{0,3} = 0$.

We then assume $\|\mathbf{K}_{\mathrm{CK},\ell-1} - \tilde{\mathbf{K}}_{\mathrm{CK},\ell-1}\| \to 0$ holds at layer $\ell - 1$ with

$$\tilde{\mathbf{K}}_{\mathrm{CK},\ell-1} \equiv \alpha_{\ell-1,1}\mathbf{X}^{\mathsf{T}}\mathbf{X} + \mathbf{V}\mathbf{A}_{\ell-1}\mathbf{V}^{\mathsf{T}} + (\tau_{\ell-1}^2 - \tau_0^2\alpha_{\ell-1,1} - \tau_0^4\alpha_{\ell-1,3})\mathbf{I}_n,$$

for $\mathbf{A}_{\ell-1} = \begin{bmatrix} \alpha_{\ell-1,2}\mathbf{t}\mathbf{t}^{\mathsf{T}} + \alpha_{\ell-1,3}\mathbf{T} & \alpha_{\ell-1,2}\mathbf{t} \\ \alpha_{\ell-1,2}\mathbf{t}^{\mathsf{T}} & \alpha_{\ell-1,2} \end{bmatrix}$, and work on the CK matrix $\mathbf{K}_{\mathrm{CK},\ell}$ at layer $\ell$.

By definition in (5), using the Gram-Schmidt orthogonalization procedure for standard Gaussian random variable as in [18, 4], we write

$$u = \sqrt{[\mathbf{K}_{\mathrm{CK},\ell-1}]_{ii}} \cdot \xi_i, \quad v = \frac{[\mathbf{K}_{\mathrm{CK},\ell-1}]_{ij}}{\sqrt{[\mathbf{K}_{\mathrm{CK},\ell-1}]_{ii}}} \cdot \xi_i + \sqrt{[\mathbf{K}_{\mathrm{CK},\ell-1}]_{jj} - \frac{[\mathbf{K}_{\mathrm{CK},\ell-1}]_{ij}^2}{[\mathbf{K}_{\mathrm{CK},\ell-1}]_{ii}}} \cdot \xi_j, \tag{29}$$

for *independent* $\xi_i, \xi_j \sim \mathcal{N}(0,1)$. As such, we have, for layer $\ell$ that

$$[\mathbf{K}_{\mathrm{CK},\ell}]_{ii} = \mathbb{E}\left[\sigma_\ell^2\left(\sqrt{[\mathbf{K}_{\mathrm{CK},\ell-1}]_{ii}} \cdot \xi_i\right)\right] \tag{30}$$

$$[\mathbf{K}_{\mathrm{CK},\ell}]_{ij} = \mathbb{E}\left[\sigma_\ell\left(\sqrt{[\mathbf{K}_{\mathrm{CK},\ell-1}]_{ii}} \cdot \xi_i\right)\right.$$

$$\left.\times \sigma_\ell\left(\frac{[\mathbf{K}_{\mathrm{CK},\ell-1}]_{ij}}{\sqrt{[\mathbf{K}_{\mathrm{CK},\ell-1}]_{ii}}} \cdot \xi_i + \sqrt{[\mathbf{K}_{\mathrm{CK},\ell-1}]_{jj} - \frac{[\mathbf{K}_{\mathrm{CK},\ell-1}]_{ij}^2}{[\mathbf{K}_{\mathrm{CK},\ell-1}]_{ii}}} \cdot \xi_j\right)\right], \tag{31}$$

where the expectations are now taken with respect to the *independent* random variables $\xi_i$ and $\xi_j$, and conditioned on the random vectors $\mathbf{x}_i$ and $\mathbf{x}_j$.

Based on the induction hypothesis on the layer $\ell - 1$, we have

$$[\mathbf{K}_{\mathrm{CK},\ell-1}]_{ij} = \alpha_{\ell-1,1}A_{ij} + \alpha_{\ell-1,2}(t_i+\psi_i)(t_j+\psi_j) + \alpha_{\ell-1,3}\left(\frac{1}{p}\mathbf{z}_i^{\mathsf{T}}\mathbf{z}_j\right)^2 + S_{ij} + O(p^{-3/2}), \tag{32}$$

for $i \neq j$, and

$$[\mathbf{K}_{\mathrm{CK},\ell-1}]_{ii} = \tau_{\ell-1}^2 + \alpha_{\ell-1,4}\chi_i + \alpha_{\ell-1,5}(t_i+\psi_i)^2 + O(p^{-3/2}). \tag{33}$$

(Note that the introduction of the term $S_{ij}$ does *not* alter the form of $\mathbf{K}$ or $\tilde{\mathbf{K}}$ in a spectral norm sense.)

The objective is then to derive the approximation of $[\mathbf{K}_{\mathrm{CK},\ell}]_{ij}$ and $[\mathbf{K}_{\mathrm{CK},\ell}]_{ii}$ at layer $\ell$, both to terms of order $O(p^{-3/2})$, and to subsequently derive the recursive relations between the key coefficients of layer $\ell - 1$:

$$\{\alpha_{\ell-1,1}, \alpha_{\ell-1,2}, \alpha_{\ell-1,3}, \alpha_{\ell-1,4}, \alpha_{\ell-1,5}\}, \tag{34}$$

and those of layer $\ell$

$$\{\alpha_{\ell,1}, \alpha_{\ell,2}, \alpha_{\ell,3}, \alpha_{\ell,4}, \alpha_{\ell,5}\}. \tag{35}$$

To this end, we first focus on the diagonal entries and evaluate $[\mathbf{K}_{\mathrm{CK},\ell}]_{ii}$, then on the off-diagonal terms $[\mathbf{K}_{\mathrm{CK},\ell}]_{ij}$ for $i \neq j$, and finally we conclude the proof by putting everything in matrix form.

**On the diagonal.** We start with the diagonal entries of $\mathbf{K}_{\mathrm{CK},\ell}$, which, as per its expression in (30), depends on the diagonal entries $[\mathbf{K}_{\mathrm{CK},\ell-1}]_{ii}$ at layer $\ell - 1$ as defined in (33). By Taylor-expanding $\sqrt{t}$ around the leading order term $t \simeq \tau_{\ell-1}^2 = O(1)$, one gets

$$\sqrt{[\mathbf{K}_{\mathrm{CK},\ell-1}]_{ii}} = \sqrt{\tau_{\ell-1}^2 + \alpha_{\ell-1,4}\chi_i + \alpha_{\ell-1,5}(t_i+\psi_i)^2 + O(p^{-3/2})}$$

$$= \tau_{\ell-1} + \frac{1}{2\tau_{\ell-1}}\left(\alpha_{\ell-1,4}\chi_i + \alpha_{\ell-1,5}(t_i+\psi_i)^2\right) - \frac{\alpha_{\ell-1,4}^2}{8\tau_{\ell-1}^3}(t_i+\psi_i)^2 + O(p^{-3/2})$$

$$= \tau_{\ell-1} + \frac{1}{2\tau_{\ell-1}}\alpha_{\ell-1,4}\chi_i + \frac{4\tau_{\ell-1}^2\alpha_{\ell-1,5} - \alpha_{\ell-1,4}^2}{8\tau_{\ell-1}^3}(t_i+\psi_i)^2 + O(p^{-3/2}).$$

Further Taylor-expand $\sigma_\ell^2(\sqrt{[\mathbf{K}_{\text{CK},\ell-1}]_{ii}} \cdot \xi_i) = f(\sqrt{[\mathbf{K}_{\text{CK},\ell-1}]_{ii}} \cdot \xi_i)$ around $\tau_{\ell-1}\xi_i$, we get

$$[\mathbf{K}_{\text{CK},\ell}]_{ii} = \mathbb{E}\left[\sigma_\ell^2\left(\sqrt{[\mathbf{K}_{\text{CK},\ell-1}]_{ii}} \cdot \xi\right)\right] = \mathbb{E}\left[f\left(\sqrt{[\mathbf{K}_{\text{CK},\ell-1}]_{ii}} \cdot \xi\right)\right]$$

$$= \mathbb{E}\left[f(\tau_{\ell-1}\xi) + f'(\tau_{\ell-1}\xi)\xi\left(\frac{1}{2\tau_{\ell-1}}\alpha_{\ell-1,4}\chi_i + \frac{4\tau_{\ell-1}^2\alpha_{\ell-1,5} - \alpha_{\ell-1,4}^2}{8\tau_{\ell-1}^3}(t_i + \psi_i)^2\right)\right]$$

$$+ \mathbb{E}\left[\frac{1}{2}f''(\tau_{\ell-1}\xi)\xi^2\right]\frac{\alpha_{\ell-1,4}^2}{4\tau_{\ell-1}^2}(t_i + \psi_i)^2 + O(p^{-3/2})$$

$$= \mathbb{E}[f(\tau_{\ell-1}\xi)] + \mathbb{E}[f''(\tau_{\ell-1}\xi)]\left(\frac{1}{2}\alpha_{\ell-1,4}\chi_i + \frac{4\tau_{\ell-1}^2\alpha_{\ell-1,5} - \alpha_{\ell-1,4}^2}{8\tau_{\ell-1}^2}(t_i + \psi_i)^2\right)$$

$$+ \mathbb{E}\left[\frac{1}{2}f''(\tau_{\ell-1}\xi)\xi^2\right]\frac{\alpha_{\ell-1,4}^2}{4\tau_{\ell-1}^2}(t_i + \psi_i)^2 + O(p^{-3/2})$$

$$= \mathbb{E}[f(\tau_{\ell-1}\xi)] + \frac{\alpha_{\ell-1,4}}{2}\mathbb{E}[f''(\tau_{\ell-1}\xi)]\chi_i$$

$$+ \frac{4\alpha_{\ell-1,5}\mathbb{E}[f''(\tau_{\ell-1}\xi)] + \alpha_{\ell-1,4}^2\mathbb{E}[f''''(\tau_{\ell-1}\xi)]}{8}(t_i + \psi_i)^2 + O(p^{-3/2}),$$

where we denote the shortcut $f(x) = \sigma_\ell^2(x)$ and use the facts that

$$\mathbb{E}[f'(\tau_{\ell-1}\xi)\xi] = \tau_{\ell-1}\mathbb{E}[f''(\tau_{\ell-1}\xi)], \quad \mathbb{E}[f''(\tau_{\ell-1}\xi)(\xi^2 - 1)] = \tau_{\ell-1}^2\mathbb{E}[f''''(\tau_{\ell-1}\xi)], \qquad (36)$$

for $\xi \sim \mathcal{N}(0,1)$, as a consequence of the Gaussian integration by parts formula.

As a consequence, we obtain the following relation

$$\tau_\ell = \sqrt{\mathbb{E}[\sigma_\ell^2(\tau_{\ell-1}\xi)]}, \quad \alpha_{\ell,4} = \alpha_{\ell-1,4}\mathbb{E}\left[(\sigma_\ell'(\tau_{\ell-1}\xi))^2 + \sigma_\ell(\tau_{\ell-1}\xi)\sigma_\ell''(\tau_{\ell-1}\xi)\right],$$

$$\alpha_{\ell,5} = \alpha_{\ell-1,5}\mathbb{E}\left[(\sigma_\ell'(\tau_{\ell-1}\xi))^2 + \sigma_\ell(\tau_{\ell-1}\xi)\sigma_\ell''(\tau_{\ell-1}\xi)\right]$$

$$+ \frac{\alpha_{\ell-1,4}^2}{4}\mathbb{E}\left[\sigma_\ell(\tau_{\ell-1}\xi)\sigma_\ell''''(\tau_{\ell-1}\xi) + 4\sigma_\ell'(\tau_{\ell-1}\xi)\sigma_\ell'''(\tau_{\ell-1}\xi) + 3(\sigma_\ell''(\tau_{\ell-1}\xi))^2\right].$$

**Off the diagonal.** We now move on to the more involved non-diagonal entries of $\mathbf{K}_{\text{CK},\ell}$. First note, for $i \neq j$, that

$$\frac{[\mathbf{K}_{\text{CK},\ell-1}]_{ij}}{\sqrt{[\mathbf{K}_{\text{CK},\ell-1}]_{ii}}} = \frac{[\mathbf{K}_{\text{CK},\ell-1}]_{ij}}{\tau_{\ell-1} + \frac{1}{2\tau_{\ell-1}}\alpha_{\ell-1,4}\chi_i + \frac{4\tau_{\ell-1}^2\alpha_{\ell-1,5} - \alpha_{\ell-1,4}^2}{8\tau_{\ell-1}^3}(t_i + \psi_i)^2 + O(p^{-3/2})}$$

$$= [\mathbf{K}_{\text{CK},\ell-1}]_{ij}\left(\frac{1}{\tau_{\ell-1}} - \frac{1}{\tau_{\ell-1}^2}\left(\frac{\alpha_{\ell-1,4}}{2\tau_{\ell-1}}(t_i + \psi_i) + O(p^{-1})\right)\right) + O(p^{-3/2})$$

$$= \frac{1}{\tau_{\ell-1}}[\mathbf{K}_{\text{CK},\ell-1}]_{ij} - \frac{\alpha_{\ell-1,4}\alpha_{\ell-1,1}}{2\tau_{\ell-1}^3}(t_i + \psi_i)\frac{1}{p}\mathbf{z}_i^\mathsf{T}\mathbf{z}_j + O(p^{-3/2})$$

$$= \frac{1}{\tau_{\ell-1}}[\mathbf{K}_{\text{CK},\ell-1}]_{ij} + S_{ij} + O(p^{-3/2}) = O(p^{-1/2}),$$

with

$$[\mathbf{K}_{\text{CK},\ell-1}]_{ij} = \alpha_{\ell-1,1}A_{ij} + \alpha_{\ell-1,2}(t_i + \psi_i)(t_j + \psi_j) + \alpha_{\ell-1,3}\left(\frac{1}{p}\mathbf{z}_i^\mathsf{T}\mathbf{z}_j\right)^2 + S_{ij} + O(p^{-3/2})$$

$$= O(p^{-1/2}),$$

as in Equation (32), where we recall that $S_{ij} = O(p^{-1})$ denotes a matrix of the form $\frac{1}{p}\mathbf{z}_i^\mathsf{T}\mathbf{z}_j\left(\gamma_1(t_i + \psi_i) + \gamma_2(t_j + \psi_j)\right)$ and of vanishing spectral norm as defined in (27).

Then,

$$\sqrt{[\mathbf{K}_{\mathrm{CK},\ell-1}]_{jj} - \frac{[\mathbf{K}_{\mathrm{CK},\ell-1}]_{ij}^2}{[\mathbf{K}_{\mathrm{CK},\ell-1}]_{ii}}}$$

$$= \sqrt{\tau_{\ell-1}^2 + \alpha_{\ell-1,4}\chi_j + \alpha_{\ell-1,5}(t_j + \psi_j)^2 - \frac{\alpha_{\ell-1,1}^2}{\tau_{\ell-1}^2}\left(\frac{1}{p}\mathbf{z}_i^{\mathsf{T}}\mathbf{z}_j\right)^2 + O(p^{-3/2})}$$

$$= \tau_{\ell-1} + \frac{1}{2\tau_{\ell-1}}\left(\alpha_{\ell-1,4}\chi_j + \alpha_{\ell-1,5}(t_j + \psi_j)^2 - \frac{\alpha_{\ell-1,1}^2}{\tau_{\ell-1}^2}\left(\frac{1}{p}\mathbf{z}_i^{\mathsf{T}}\mathbf{z}_j\right)^2\right) - \frac{\alpha_{\ell-1,4}^2(t_j + \psi_j)^2}{8\tau_{\ell-1}^3}$$

$$\quad + O(p^{-3/2})$$

$$= \tau_{\ell-1} + \frac{1}{2\tau_{\ell-1}}\left(\alpha_{\ell-1,4}\chi_j - \frac{\alpha_{\ell-1,1}^2}{\tau_{\ell-1}^2}\left(\frac{1}{p}\mathbf{z}_i^{\mathsf{T}}\mathbf{z}_j\right)^2\right) + \frac{4\tau_{\ell-1}^2\alpha_{\ell-1,5} - \alpha_{\ell-1,4}^2}{8\tau_{\ell-1}^3}(t_j + \psi_j)^2$$

$$\quad + O(p^{-3/2}).$$

As a consequence, we get, again by Taylor expansion that

$$\sigma_\ell\left(\sqrt{[\mathbf{K}_{\mathrm{CK},\ell-1}]_{ii}} \cdot \xi_i\right)\sigma_\ell\left(\frac{[\mathbf{K}_{\mathrm{CK},\ell-1}]_{ij}}{\sqrt{[\mathbf{K}_{\mathrm{CK},\ell-1}]_{ii}}} \cdot \xi_i + \sqrt{[\mathbf{K}_{\mathrm{CK},\ell-1}]_{jj} - \frac{[\mathbf{K}_{\mathrm{CK},\ell-1}]_{ij}^2}{[\mathbf{K}_{\mathrm{CK},\ell-1}]_{ii}}} \cdot \xi_j\right)$$

$$= \sigma_\ell\left(\tau_{\ell-1}\xi_i + \frac{1}{2\tau_{\ell-1}}\alpha_{\ell-1,4}\chi_i\xi_i + \frac{4\tau_{\ell-1}^2\alpha_{\ell-1,5} - \alpha_{\ell-1,4}^2}{8\tau_{\ell-1}^3}(t_i + \psi_i)^2\xi_i + O(p^{-3/2})\right)$$

$$\times \sigma_\ell\left(\frac{1}{\tau_{\ell-1}}[\mathbf{K}_{\mathrm{CK},\ell-1}]_{ij}\xi_i + \tau_{\ell-1}\xi_j + \frac{1}{2\tau_{\ell-1}}\left(\alpha_{\ell-1,4}\chi_j - \frac{\alpha_{\ell-1,1}^2}{\tau_{\ell-1}^2}\left(\frac{1}{p}\mathbf{z}_i^{\mathsf{T}}\mathbf{z}_j\right)^2\right)\xi_j\right.$$

$$\left. + \frac{4\tau_{\ell-1}^2\alpha_{\ell-1,5} - \alpha_{\ell-1,4}^2}{8\tau_{\ell-1}^3}(t_j + \psi_j)^2\xi_j + O(p^{-3/2})\right)$$

$$= \left(\sigma_\ell(\tau_{\ell-1}\xi_i) + \sigma_\ell'(\tau_{\ell-1}\xi_i)\xi_i\left(\frac{1}{2\tau_{\ell-1}}\alpha_{\ell-1,4}\chi_i + \frac{4\tau_{\ell-1}^2\alpha_{\ell-1,5} - \alpha_{\ell-1,4}^2}{8\tau_{\ell-1}^3}(t_i + \psi_i)^2\right)\right.$$

$$\left. + \sigma_\ell''(\tau_{\ell-1}\xi_i)\xi_i^2\frac{\alpha_{\ell-1,4}^2}{8\tau_{\ell-1}^2}(t_i + \psi_i)^2\right)$$

$$\times \left(\sigma_\ell(\tau_{\ell-1}\xi_j) + \sigma_\ell'(\tau_{\ell-1}\xi_j)\left(\frac{[\mathbf{K}_{\mathrm{CK},\ell-1}]_{ij}}{\tau_{\ell-1}}\xi_i + \frac{1}{2\tau_{\ell-1}}\left(\alpha_{\ell-1,4}\chi_j - \frac{\alpha_{\ell-1,1}^2}{\tau_{\ell-1}^2}\left(\frac{1}{p}\mathbf{z}_i^{\mathsf{T}}\mathbf{z}_j\right)^2\right)\xi_j\right.\right.$$

$$\left.\left. + \frac{4\tau_{\ell-1}^2\alpha_{\ell-1,5} - \alpha_{\ell-1,4}^2}{8\tau_{\ell-1}^3}(t_j + \psi_j)^2\xi_j\right)\right.$$

$$\left. + \frac{1}{2}\sigma_\ell''(\tau_{\ell-1}\xi_j)\left(\frac{\alpha_{\ell-1,1}\frac{1}{p}\mathbf{z}_i^{\mathsf{T}}\mathbf{z}_j}{\tau_{\ell-1}}\xi_i + \frac{\alpha_{\ell-1,4}(t_j + \psi_j)}{2\tau_{\ell-1}}\xi_j\right)^2\right) + O(p^{-3/2})$$

$$\equiv \left(\sigma_\ell(\tau_{\ell-1}\xi_i) + T_{1,i} + T_{2,i}\right)\left(\sigma_\ell(\tau_{\ell-1}\xi_j) + T_{3,ij} + T_{3,j} + T_{4,ij} + T_{4,j} + S_{ij}\right) + O(p^{-3/2}),$$

where we denote the shortcuts:

$$T_{1,i} = \sigma_\ell'(\tau_{\ell-1}\xi_i)\xi_i \cdot \frac{\alpha_{\ell-1,4}}{2\tau_{\ell-1}}\chi_i = O(p^{-1/2}),$$

$$T_{2,i} = \left(\frac{\alpha_{\ell-1,5}\sigma_\ell'(\tau_{\ell-1}\xi_i)\xi_i}{2\tau_{\ell-1}} + \alpha_{\ell-1,4}^2\frac{\sigma_\ell''(\tau_{\ell-1}\xi_i)\xi_i^2\tau_{\ell-1} - \sigma_\ell'(\tau_{\ell-1}\xi_i)\xi_i}{8\tau_{\ell-1}^3}\right)(t_i + \psi_i)^2 = O(p^{-1}),$$

that *only* depend on $\xi_i$; and

$$T_{3,ij} = \sigma'_\ell(\tau_{\ell-1}\xi_j)\xi_i \cdot \frac{[\mathbf{K}_{\mathrm{CK},\ell-1}]_{ij}}{\tau_{\ell-1}} = O(p^{-1/2}),$$

$$T_{4,ij} = \frac{1}{2}\sigma''_\ell(\tau_{\ell-1}\xi_j)\xi_i^2 \cdot \frac{\alpha_{\ell-1,1}^2}{\tau_{\ell-1}^2}\left(\frac{1}{p}\mathbf{z}_i^\mathsf{T}\mathbf{z}_j\right)^2 = O(p^{-1}),$$

that depend on both $\xi_i$ and $\xi_j$; and

$$T_{3,j} = \sigma'_\ell(\tau_{\ell-1}\xi_j)\xi_j \cdot \left(\frac{\alpha_{\ell-1,4}}{2\tau_{\ell-1}}\chi_j - \frac{\alpha_{\ell-1,1}^2}{2\tau_{\ell-1}^3}\left(\frac{1}{p}\mathbf{z}_i^\mathsf{T}\mathbf{z}_j\right)^2 + \frac{4\tau_{\ell-1}^2\alpha_{\ell-1,5} - \alpha_{\ell-1,4}^2}{8\tau_{\ell-1}^3}(t_j + \psi_j)^2\right)$$

$$= O(p^{-1/2}),$$

$$T_{4,j} = \frac{1}{2}\sigma''_\ell(\tau_{\ell-1}\xi_j)\xi_j^2 \cdot \frac{\alpha_{\ell-1,4}^2}{4\tau_{\ell-1}^2}(t_j + \psi_j)^2 = O(p^{-1}),$$

that *only* depend on $\xi_j$, where we particularly note that the cross terms are of the form $S_{ij}$.

As such, we have

$$\sigma_\ell\left(\sqrt{[\mathbf{K}_{\mathrm{CK},\ell-1}]_{ii}} \cdot \xi_i\right)\sigma_\ell\left(\frac{[\mathbf{K}_{\mathrm{CK},\ell-1}]_{ij}}{\sqrt{[\mathbf{K}_{\mathrm{CK},\ell-1}]_{ii}}} \cdot \xi_i + \sqrt{[\mathbf{K}_{\mathrm{CK},\ell-1}]_{jj} - \frac{[\mathbf{K}_{\mathrm{CK},\ell-1}]_{ij}^2}{[\mathbf{K}_{\mathrm{CK},\ell-1}]_{ii}}} \cdot \xi_j\right)$$

$$= \sigma_\ell(\tau_{\ell-1}\xi_i)\sigma_\ell(\tau_{\ell-1}\xi_j) + \sigma_\ell(\tau_{\ell-1}\xi_i)(T_{3,ij} + T_{4,ij}) + \sigma_\ell(\tau_{\ell-1}\xi_i)(T_{3,j} + T_{4,j})$$

$$+ \sigma_\ell(\tau_{\ell-1}\xi_j)(T_{1,i} + T_{2,i}) + T_{1,i}(T_{3,ij} + T_{3,j}) + S_{ij} + O(p^{-3/2}),$$

with in particular

$$T_{1,i}(T_{3,ij} + T_{3,j}) = \sigma'_\ell(\tau_{\ell-1}\xi_i)\xi_i \cdot \frac{\alpha_{\ell-1,4}}{2\tau_{\ell-1}}(t_i + \psi_i) \cdot \sigma'_\ell(\tau_{\ell-1}\xi_j)\xi_i\frac{\alpha_{\ell-1,1}\frac{1}{p}\mathbf{z}_i^\mathsf{T}\mathbf{z}_j}{\tau_{\ell-1}}$$

$$+ \sigma'_\ell(\tau_{\ell-1}\xi_i)\xi_i \cdot \frac{\alpha_{\ell-1,4}}{2\tau_{\ell-1}}(t_i + \psi_i) \cdot \sigma'_\ell(\tau_{\ell-1}\xi_j)\xi_j\frac{\alpha_{\ell-1,4}}{2\tau_{\ell-1}}(t_j + \psi_j)$$

$$+ O(p^{-3/2})$$

$$= \sigma'_\ell(\tau_{\ell-1}\xi_i)\xi_i\sigma'_\ell(\tau_{\ell-1}\xi_j)\xi_j\frac{\alpha_{\ell-1,4}^2}{4\tau_{\ell-1}^2}(t_i + \psi_i)(t_j + \psi_j) + S_{ij} + O(p^{-3/2}).$$

We thus conclude, for $\mathbb{E}[\sigma_\ell(\tau_{\ell-1}\xi)] = 0$ with $\xi \sim \mathcal{N}(0,1)$, that

$$[\mathbf{K}_{\mathrm{CK},\ell}]_{ij}$$

$$= \mathbb{E}\left[\sigma_\ell\left(\sqrt{[\mathbf{K}_{\mathrm{CK},\ell-1}]_{ii}} \cdot \xi_i\right)\sigma_\ell\left(\frac{[\mathbf{K}_{\mathrm{CK},\ell-1}]_{ij}}{\sqrt{[\mathbf{K}_{\mathrm{CK},\ell-1}]_{ii}}} \cdot \xi_i + \sqrt{[\mathbf{K}_{\mathrm{CK},\ell-1}]_{jj} - \frac{[\mathbf{K}_{\mathrm{CK},\ell-1}]_{ij}^2}{[\mathbf{K}_{\mathrm{CK},\ell-1}]_{ii}}} \cdot \xi_j\right)\right]$$

$$= \mathbb{E}[\sigma_\ell(\tau_{\ell-1}\xi_i)(T_{3,ij} + T_{4,ij})] + \mathbb{E}\left[\sigma'_\ell(\tau_{\ell-1}\xi_i)\xi_i\sigma'_\ell(\tau_{\ell-1}\xi_j)\xi_j\right]\frac{\alpha_{\ell-1,4}^2}{4\tau_{\ell-1}^2}(t_i + \psi_i)(t_j + \psi_j)$$

$$+ S_{ij} + O(p^{-3/2})$$

$$= \mathbb{E}\left[\sigma_\ell(\tau_{\ell-1}\xi_i)\sigma'_\ell(\tau_{\ell-1}\xi_j)\xi_i\frac{[\mathbf{K}_{\mathrm{CK},\ell-1}]_{ij}}{\tau_{\ell-1}}\right] + \frac{1}{2}\mathbb{E}\left[\sigma_\ell(\tau_{\ell-1}\xi_i)\sigma''_\ell(\tau_{\ell-1}\xi_j)\xi_i^2\frac{\alpha_{\ell-1,1}^2}{\tau_{\ell-1}^2}\left(\frac{1}{p}\mathbf{z}_i^\mathsf{T}\mathbf{z}_j\right)^2\right]$$

$$+ \mathbb{E}[\sigma'_\ell(\tau_{\ell-1}\xi_i)\xi_i]\mathbb{E}[\sigma'_\ell(\tau_{\ell-1}\xi_j)\xi_j]\frac{\alpha_{\ell-1,4}^2}{4\tau_{\ell-1}^2}(t_i + \psi_i)(t_j + \psi_j) + S_{ij} + O(p^{-3/2})$$

$$= \mathbb{E}[\sigma'_\ell(\tau_{\ell-1}\xi)]^2[\mathbf{K}_{\mathrm{CK},\ell-1}]_{ij} + \frac{\alpha_{\ell-1,1}^2}{2}\mathbb{E}[\sigma''_\ell(\tau_{\ell-1}\xi)]^2\left(\frac{1}{p}\mathbf{z}_i^\mathsf{T}\mathbf{z}_j\right)^2$$

$$+ \frac{\alpha_{\ell-1,4}^2}{4}\mathbb{E}[\sigma''_\ell(\tau_{\ell-1}\xi)]^2(t_i + \psi_i)(t_j + \psi_j) + S_{ij} + O(p^{-3/2}),$$

where we used again the fact that

$$\mathbb{E}[\xi f(\tau\xi)] = \tau\mathbb{E}[f'(\tau\xi)], \quad \mathbb{E}[\xi^2 f(\tau\xi)] = \mathbb{E}[(\xi^2-1)f(\tau\xi)] = \tau^2\mathbb{E}[f''(\tau\xi)], \tag{37}$$

for $\mathbb{E}[f(\tau\xi)] = 0$.

This allows us to conclude that

$$\alpha_{\ell,1} = \mathbb{E}[\sigma'_\ell(\tau_{\ell-1}\xi)]^2\alpha_{\ell-1,1}, \quad \alpha_{\ell,2} = \mathbb{E}[\sigma'_\ell(\tau_{\ell-1}\xi)]^2\alpha_{\ell-1,2} + \frac{1}{4}\mathbb{E}[\sigma''_\ell(\tau_{\ell-1}\xi)]^2\alpha^2_{\ell-1,4}, \tag{38}$$

$$\alpha_{\ell,3} = \mathbb{E}[\sigma'_\ell(\tau_{\ell-1}\xi)]^2\alpha_{\ell-1,3} + \frac{1}{2}\mathbb{E}[\sigma''_\ell(\tau_{\ell-1}\xi)]^2\alpha^2_{\ell-1,1}. \tag{39}$$

**Assembling in matrix form.** Following the discussion above, we have, uniformly for all $i \neq j \in \{1,\ldots,n\}$ that,

$$[\mathbf{K}_{\mathrm{CK},\ell}]_{ij} = \alpha_{\ell,1}A_{ij} + \alpha_{\ell,2}(t_i+\psi_i)(t_j+\psi_j) + \alpha_{\ell,3}\left(\frac{1}{p}\mathbf{z}_i^\mathsf{T}\mathbf{z}_j\right)^2 + S_{ij} + O(p^{-3/2}), \tag{40}$$

and

$$[\mathbf{K}_{\mathrm{CK},\ell}]_{ii} = \tau_\ell^2 + O(p^{-1/2}), \tag{41}$$

so that in matrix form (by using the fact that $\|\mathbf{A}\| \leq n\|\mathbf{A}\|_\infty$ for $\mathbf{A} \in \mathbb{R}^{n\times n}$ with $\|\mathbf{A}\|_\infty = \max_{ij}|\mathbf{A}|_{ij}$ and $\{S_{ij}\}_{i,j} = O_{\|\cdot\|}(p^{-\frac{1}{2}})$, see [11]):

$$\mathbf{K}_{\mathrm{CK},\ell} = \alpha_{\ell,1}\mathbf{X}^\mathsf{T}\mathbf{X} + \mathbf{V}\mathbf{A}_\ell\mathbf{V}^\mathsf{T} + (\tau_\ell^2 - \tau_0^2\alpha_{\ell,1} - \tau_0^4\alpha_{\ell,3})\mathbf{I}_n + O_{\|\cdot\|}(p^{-\frac{1}{2}}), \tag{42}$$

where $O_{\|\cdot\|}(p^{-\frac{1}{2}})$ denotes matrices of spectral norm order $O(p^{-\frac{1}{2}})$, with

$$\mathbf{V} = [\mathbf{J}/\sqrt{p},\ \boldsymbol{\psi}] \in \mathbb{R}^{n\times(K+1)}, \quad \mathbf{A}_\ell = \begin{bmatrix} \alpha_{\ell,2}\mathbf{t}\mathbf{t}^\mathsf{T} + \alpha_{\ell,3}\mathbf{T} & \alpha_{\ell,2}\mathbf{t} \\ \alpha_{\ell,2}\mathbf{t}^\mathsf{T} & \alpha_{\ell,2} \end{bmatrix}, \tag{43}$$

and

$$\mathbf{T} = \left\{\frac{1}{p}\operatorname{tr}\mathbf{C}_a\mathbf{C}_b\right\}_{a,b=1}^K, \quad \mathbf{t} = \left\{\frac{1}{\sqrt{p}}\operatorname{tr}\mathbf{C}_a^\circ\right\}_{a=1}^K, \tag{44}$$

as in the statement of Theorem 1. This concludes the proof of Theorem 1.

**Lemma 1** (Consistent estimation of $\tau_0$). *Let Assumption 1 hold and let $\tau_0 \equiv \sqrt{\operatorname{tr}\mathbf{C}^\circ/p}$. Then, as $n,p \to \infty$ with $p/n \to c \in (0,\infty)$, we have,*

$$\frac{1}{n}\sum_{i=1}^n \|\mathbf{x}_i\|^2 - \tau_0^2 \to 0, \tag{45}$$

*almost surely.*

*Proof of Lemma 1.* It follows from (26) that

$$\frac{1}{n}\sum_{i=1}^n \|\mathbf{x}_i\|^2 = \frac{1}{n}\sum_{a=1}^K\sum_{i=1}^{n_a}\left(\frac{1}{p}\|\boldsymbol{\mu}_a\|^2 - \frac{2}{p}\boldsymbol{\mu}_a^\mathsf{T}\mathbf{z}_i + \frac{1}{p}\|\mathbf{z}_i\|^2\right). \tag{46}$$

From Assumption 1, we have $\|\boldsymbol{\mu}_a\| = O(1)$ so that $\frac{1}{n}\sum_{a=1}^K\sum_{i=1}^{n_a}\frac{1}{p}\|\boldsymbol{\mu}_a\|^2 = O(p^{-1})$. Since $\mathbb{E}[\mathbf{z}_i] = \mathbf{0}$, the second term $\frac{2}{p}\boldsymbol{\mu}_a^\mathsf{T}\mathbf{z}_i$ of (46) is a weighted sum of independent zero mean random variables and vanishes with probability one as $n,p \to \infty$ by a mere application of Chebyshev's inequality and the Borel–Cantelli lemma. Finally, using the strong law of large numbers on the third term of equation (46), we have almost surely,

$$\frac{1}{n}\frac{1}{p}\sum_{a=1}^K\sum_{i=1}^{n_a}\|\mathbf{z}_i\|^2 = \frac{1}{p}\sum_{a=1}^K\frac{n_a}{n}\operatorname{tr}\mathbf{C}_a + o(1) = \operatorname{tr}\mathbf{C}^\circ/p + o(1), \tag{47}$$

where in the last line we use $\operatorname{tr}\mathbf{C}_a^\circ = O(\sqrt{p})$ from Assumption 1, and thus $\frac{1}{n}\sum_{i=1}^n\|\mathbf{z}_i\|^2 - \tau_0^2 \to 0$ almost surely. This concludes the proof of Lemma 1. $\square$

## A.2  Proof of Theorem 2

In this section, we provide detailed proof of Theorem 2. We follows the same notations and working assumptions as in the proof of Theorem 1 in Appendix A.1.

As already mentioned in (6), the NTK matrices $\mathbf{K}_{\mathrm{NTK},\ell}$ of layer $\ell$ can be defined, again in an iterative manner, via the CK matrices $\mathbf{K}_{\mathrm{CK},\ell}$ and $\mathbf{K}'_{\mathrm{CK},\ell}$ as follows [28]:

$$\mathbf{K}_{\mathrm{NTK},0} = \mathbf{K}_{\mathrm{CK},0} = \mathbf{X}^{\mathsf{T}}\mathbf{X},$$
$$\mathbf{K}_{\mathrm{NTK},\ell} = \mathbf{K}_{\mathrm{CK},\ell} + \mathbf{K}_{\mathrm{NTK},\ell-1} \circ \mathbf{K}'_{\mathrm{CK},\ell}.$$

where '$\mathbf{A} \circ \mathbf{B}$' denotes the Hadamard product between two matrices $\mathbf{A}, \mathbf{B}$, and $\mathbf{K}'_{\mathrm{CK},\ell} \in \mathbb{R}^{n \times n}$ denotes a CK matrix with nonlinear activation $\sigma'_\ell(\cdot)$ instead of $\sigma_\ell(\cdot)$ as for $\mathbf{K}_{\mathrm{CK},\ell}$ in (4), that is

$$[\mathbf{K}'_{\mathrm{CK},\ell}]_{ij} = \mathbb{E}_{u,v}[\sigma'_\ell(u)\sigma'_\ell(v)], \quad u,v \sim \mathcal{N}\left(\mathbf{0}, \begin{bmatrix} [\mathbf{K}'_{\mathrm{CK},\ell-1}]_{ii} & [\mathbf{K}'_{\mathrm{CK},\ell-1}]_{ij} \\ [\mathbf{K}'_{\mathrm{CK},\ell-1}]_{ij} & [\mathbf{K}'_{\mathrm{CK},\ell-1}]_{jj} \end{bmatrix}\right). \tag{48}$$

To evaluate the eigenspectral behavior of the NTK matrix $\mathbf{K}_{\mathrm{NTK},\ell}$, it is crucial to assess the behavior of the CK matrix $\mathbf{K}'_{\mathrm{CK},\ell}$ with activation $\sigma'_\ell$. This is, however, far from trivial, since one *cannot* apply Theorem 1 directly (which relies on the key assumption $\mathbb{E}[\sigma_\ell(\tau_{\ell-1}\xi)] = 0$) by simply assuming that both $\mathbb{E}[\sigma_\ell(\tau_{\ell-1}\xi)]$ and $\mathbb{E}[\sigma'_\ell(\tau_{\ell-1}\xi)]$ are zero. In fact, as shown in the statement of Theorem 2, the evaluation of the CK matrices $\mathbf{K}'_{\mathrm{CK},\ell}$ leads to the *different* sequence $\dot\tau_\ell$ that needs to be carefully studied. In the sequel, we extend, in Section A.2.1, the result in Theorem 1 to the general case of $\mathbb{E}[\sigma_\ell(\tau_{\ell-1}\xi)] \neq 0$, which will play a key role in the proof of Theorem 2 presented in Section A.2.2.

### A.2.1  Assessment of CK matrix $\mathbf{K}_{\mathrm{CK},\ell}$ with $\mathbb{E}[\sigma_\ell(\tau_{\ell-1}\xi)] \neq 0$

In this subsection, we will extend Theorem 1 to the case of possibly non-zero $\mathbb{E}[\sigma_\ell(\tau_{\ell-1}\xi)]$, $\xi \sim \mathcal{N}(0,1)$, which will be the key ingredient in the proof of Theorem 2 in Section A.2.2.

In the following, we recall some notations introduced in Section A.1 and will be used in the remainder of the proof: For $\mathbf{x}_i, \mathbf{x}_j \in \mathbb{R}^p$ with $i \neq j$, we have

$$\mathbf{x}_i = \boldsymbol{\mu}_i/\sqrt{p} + \mathbf{z}_i/\sqrt{p}, \quad \mathbf{x}_j = \boldsymbol{\mu}_j/\sqrt{p} + \mathbf{z}_j/\sqrt{p}, \tag{49}$$

with $\mathbf{z}_i \sim \mathcal{N}(\mathbf{0}, \mathbf{C}_i)$, $\mathbf{z}_j \sim \mathcal{N}(\mathbf{0}, \mathbf{C}_j)$, and

$$A_{ij} \equiv \underbrace{\frac{1}{p}\mathbf{z}_i^{\mathsf{T}}\mathbf{z}_j}_{O(p^{-1/2})} + \underbrace{\frac{1}{p}\boldsymbol{\mu}_i^{\mathsf{T}}\boldsymbol{\mu}_j + \frac{1}{p}(\boldsymbol{\mu}_i^{\mathsf{T}}\mathbf{z}_j + \boldsymbol{\mu}_j^{\mathsf{T}}\mathbf{z}_i)}_{O(p^{-1})}, \tag{50}$$

$$t_i \equiv \frac{1}{p}\operatorname{tr}\mathbf{C}_i^\circ = O(p^{-1/2}), \quad \psi_i = \frac{1}{p}\|\mathbf{z}_i\|^2 - \frac{1}{p}\operatorname{tr}\mathbf{C}_i = O(p^{-1/2}),$$

$$\chi_i \equiv \underbrace{t_i + \psi_i}_{O(p^{-1/2})} + \underbrace{\|\boldsymbol{\mu}_i\|^2/p + 2\boldsymbol{\mu}_i^{\mathsf{T}}\mathbf{z}_i/p}_{O(p^{-1})} = \|\mathbf{x}_i\|^2 - \tau_0^2,$$

$$\tau_0 \equiv \sqrt{\frac{1}{p}\operatorname{tr}\mathbf{C}^\circ} = O(1).$$

Let

$$S_{ij}(\gamma_1, \gamma_2) = \frac{1}{p}\mathbf{z}_i^{\mathsf{T}}\mathbf{z}_j\left(\gamma_1(t_i + \psi_i) + \gamma_2(t_j + \psi_j)\right), \tag{51}$$

and

$$P_{ij}(\rho_1, \ldots, \rho_6) = \rho_1\chi_i + \rho_2\chi_j + \rho_3(t_i + \psi_i)^2 + \rho_4(t_j + \psi_j)^2 + \rho_5(t_i + \psi_i)(t_j + \psi_j)$$
$$+ \rho_6\left(\frac{1}{p}\mathbf{z}_i^{\mathsf{T}}\mathbf{z}_j\right)^2 = O(p^{-1/2}) \tag{52}$$

for random or deterministic scalars $\gamma_1, \gamma_2, \rho_1, \rho_2, \rho_3, \rho_4, \rho_5, \rho_6 = O(1)$ (with high probability when being random) with respect to $n, p$. We will simply denote them $S_{ij}$ and $P_{ij}$, respectively.

We have already known from [11] and have used in the proof of Theorem 1 in Section A.1 that $S_{ij} = O(p^{-1})$ and it leads to a matrix of vanishing spectral norm as $n, p \to \infty$.

In the following remark, we present a few results on the products of $P_{ij}$, $S_{ij}$, and $A_{ij}$.

**Lemma 2** (Products between $P_{ij}$, $S_{ij}$, and $A_{ij}$). *We have, for random or deterministic scalars* $\rho_1, \rho_2, \rho_3, \rho_4, \rho_5, \rho_6$ *of order* $O(1)$ *(with high probability when being random), that*

$$P_{ij}(\rho_1, \ldots, \rho_6) \cdot P_{ij}(\tilde{\rho}_1, \ldots, \tilde{\rho}_6) = P_{ij}(0, 0, \rho_1\tilde{\rho}_1, \rho_2\tilde{\rho}_2, \rho_1\tilde{\rho}_2 + \rho_2\tilde{\rho}_1, 0) + O(p^{-3/2}).$$

*We have similarly that*

$$P_{ij}(\rho_1, \rho_2, \rho_3, \rho_4, \rho_5, \rho_6) \cdot A_{ij} = S_{ij} + O(p^{-3/2}),$$

*as well as*

$$A_{ij} \cdot A_{ij} = P_{ij}(0, 0, 0, 0, 0, 1) + O(p^{-3/2}).$$

With Lemma 2, we are ready for the following result that generalizes the characterization in Theorem 1 to (possibly) non-centered activations.

**Theorem 3** (Asymptotic behavior of CK matrix: general case). *Let Assumptions 1–3 hold, let* $\tau_0, \tau_1, \ldots, \tau_L \geq 0$ *be the non-negative sequence defined in* (7). *Then, for the CK matrix* $\mathbf{K}_{\mathrm{CK},\ell}$ *of layer* $\ell \in \{0, 1, \ldots, L\}$ *defined in* (4), *we have, as* $n, p \to \infty$, *the following entry-wise result:*

    *(i) on the diagonal, for* $i \in \{1, \ldots, n\}$ *that,*

$$[\mathbf{K}_{\mathrm{CK},\ell}]_{ii} = \tau_\ell^2 + P_{ij} + O(p^{-3/2}); \tag{53}$$

    *(ii) off the diagonal, for* $i \neq j$ *that,*

$$[\mathbf{K}_{\mathrm{CK},\ell}]_{ij} = d_\ell + \alpha_{\ell,1} A_{ij} + S_{ij} + P_{ij} + O(p^{-3/2}), \tag{54}$$

*almost surely, for* $A_{ij}$, $S_{ij}$, *and* $P_{ij}$ *of the form in* (50), (51), *and* (52), *respectively, as well as* $d_\ell, \alpha_{\ell,1}$ *satisfying*

$$d_\ell = \mathbb{E}_{\xi_1, \xi_2} \left[ \sigma_\ell(\tau_{\ell-1}\xi_1) \sigma_\ell\left(\frac{d_{\ell-1}}{\tau_{\ell-1}}\xi_1 + \sqrt{\lambda_{\ell-1}}\xi_2\right) \right],$$

$$\alpha_{\ell,1} = \mathbb{E}_{\xi_1, \xi_2} \left[ \sigma_\ell(\tau_{\ell-1}\xi_1) \sigma_\ell'\left(\frac{d_{\ell-1}}{\tau_{\ell-1}}\xi_1 + \sqrt{\lambda_{\ell-1}}\xi_2\right) \left(\frac{\alpha_{\ell-1,1}}{\tau_{\ell-1}}\xi_1 - \frac{1}{\sqrt{\lambda_{\ell-1}}}\frac{d_{\ell-1}\alpha_{\ell-1,1}}{\tau_{\ell-1}^2}\xi_2\right) \right].$$

*with* $d_0 = 0$, $\alpha_{0,1} = 1$, *and* $\lambda_{\ell-1} = \tau_{\ell-1}^2 - d_{\ell-1}^2/\tau_{\ell-1}^2$ *for independent* $\xi_1, \xi_2 \sim \mathcal{N}(0, 1)$.

*Proof of Theorem 3.* We prove Theorem 3 by induction on $\ell \in \{0, 1, \ldots, L\}$ as follows.

For $\ell = 0$, we have $\mathbf{K}_{\mathrm{CK},0} = \mathbf{X}^\mathsf{T}\mathbf{X}$, so that

$$[\mathbf{K}_{\mathrm{CK},0}]_{ii} = \tau_0^2 + \chi_i + (t_i + \psi_i)^2 + O(p^{-3/2}) = \tau_0^2 + P_{ij} + O(p^{-3/2}),$$

and

$$[\mathbf{K}_{\mathrm{CK},0}]_{ij} = A_{ij} + O(p^{-3/2}),$$

with $d_0 = 0$, $\alpha_{0,1} = 1$.

We then assume it holds for layer $\ell - 1$ that

$$[\mathbf{K}_{\mathrm{CK},\ell-1}]_{ii} = \tau_{\ell-1}^2 + P_{ij} + O(p^{-3/2}), \tag{55}$$

and for $i \neq j$,

$$[\mathbf{K}_{\mathrm{CK},\ell-1}]_{ij} = d_{\ell-1} + \alpha_{\ell-1,1} A_{ij} + P_{ij} + S_{ij} + O(p^{-3/2}). \tag{56}$$

Then, it suffices to show that $[\mathbf{K}_{\mathrm{CK},\ell}]_{ii}$ and $[\mathbf{K}_{\mathrm{CK},\ell}]_{ij}$ for layer $\ell$ take the same form, with coefficient $d_\ell$ and $\alpha_{\ell,1}$ satisfying the recursive relation in Theorem 3.

**Diagonal entries $[\mathbf{K}_{\mathrm{CK},\ell}]_{ii}$.** Similar to the proof of Theorem 1, we have, by Taylor-expansion that

$$\sqrt{[\mathbf{K}_{\mathrm{CK},\ell-1}]_{ii}} = \sqrt{\tau_{\ell-1}^2 + P_{ij} + O(p^{-3/2})}$$

$$= \tau_{\ell-1} + \frac{1}{2\tau_{\ell-1}}P_{ij} - \frac{1}{8\tau_{\ell-1}^3}P_{ij}^2 + O(p^{-3/2})$$

$$= \tau_{\ell-1} + P_{ij} + O(p^{-3/2}),$$

where we used the property on the product $P_{ij} \times P_{ij}$ in Lemma 2 in the last line. We thus have

$$[\mathbf{K}_{\mathrm{CK},\ell}]_{ii} = \mathbb{E}\left[\sigma_\ell^2\left(\sqrt{[\mathbf{K}_{\mathrm{CK},\ell-1}]_{ii}} \cdot \xi\right)\right] = \mathbb{E}\left[f\left(\sqrt{[\mathbf{K}_{\mathrm{CK},\ell-1}]_{ii}} \cdot \xi\right)\right]$$

$$= \mathbb{E}\left[f\left(\tau_{\ell-1}\xi\right) + f'\left(\tau_{\ell-1}\xi\right)\xi P_{ij} + \frac{1}{2}f''\left(\tau_{\ell-1}\xi\right)\xi^2 P_{ij}^2\right] + O(p^{-3/2})$$

$$= \mathbb{E}\left[f\left(\tau_{\ell-1}\xi\right)\right] + P_{ij} + O(p^{-3/2})$$

$$= \tau_\ell^2 + P_{ij} + O(p^{-3/2}),$$

with $f(t) = \sigma_\ell^2(t)$ and $\xi \sim \mathcal{N}(0,1)$.

**Non-diagonal entries $[\mathbf{K}_{\mathrm{CK},\ell}]_{ij}$.** Since $\sqrt{[\mathbf{K}_{\mathrm{CK},\ell-1}]_{ii}} = \tau_{\ell-1} + P_{ij} + O(p^{-3/2})$, we have, similar to the proof of the diagonal entries $[\mathbf{K}_{\mathrm{CK},\ell}]_{ii}$ above that

$$\frac{[\mathbf{K}_{\mathrm{CK},\ell-1}]_{ij}}{\sqrt{[\mathbf{K}_{\mathrm{CK},\ell-1}]_{ii}}} = \frac{[\mathbf{K}_{\mathrm{CK},\ell-1}]_{ij}}{\tau_{\ell-1} + P_{ij} + O(p^{-3/2})}$$

$$= \frac{1}{\tau_{\ell-1}}[\mathbf{K}_{\mathrm{CK},\ell-1}]_{ij} - \frac{1}{\tau_{\ell-1}^2}P_{ij} \cdot [\mathbf{K}_{\mathrm{CK},\ell-1}]_{ij} + \frac{2}{\tau_{\ell-1}^3}P_{ij}^2 \cdot [\mathbf{K}_{\mathrm{CK},\ell-1}]_{ij} + O(p^{-3/2})$$

$$= \frac{1}{\tau_{\ell-1}}[\mathbf{K}_{\mathrm{CK},\ell-1}]_{ij} + S_{ij} + P_{ij} + O(p^{-3/2})$$

$$= \frac{d_{\ell-1}}{\tau_{\ell-1}} + \frac{\alpha_{\ell-1,1}}{\tau_{\ell-1}}A_{ij} + S_{ij} + P_{ij} + O(p^{-3/2}),$$

for $[\mathbf{K}_{\mathrm{CK},\ell-1}]_{ij}$ of the form as defined in (56) by induction hypothesis, and

$$\sqrt{[\mathbf{K}_{\mathrm{CK},\ell-1}]_{jj} - \frac{[\mathbf{K}_{\mathrm{CK},\ell-1}]_{ij}^2}{[\mathbf{K}_{\mathrm{CK},\ell-1}]_{ii}}}$$

$$= \sqrt{\tau_{\ell-1}^2 + P_{ij} - \left(\frac{d_{\ell-1}^2}{\tau_{\ell-1}^2} + \frac{2d_{\ell-1} \cdot \alpha_{\ell-1,1}}{\tau_{\ell-1}^2}A_{ij} + P_{ij} + S_{ij}\right) + O(p^{-3/2})}$$

$$= \sqrt{\left(\tau_{\ell-1}^2 - \frac{d_{\ell-1}^2}{\tau_{\ell-1}^2}\right) - \frac{2d_{\ell-1} \cdot \alpha_{\ell-1,1}}{\tau_{\ell-1}^2}A_{ij} + S_{ij} + P_{ij} + O(p^{-3/2})}$$

$$= \sqrt{\lambda_{\ell-1}} - \frac{1}{\sqrt{\lambda_{\ell-1}}}\frac{d_{\ell-1} \cdot \alpha_{\ell-1,1}}{\tau_{\ell-1}^2}A_{ij} + P_{ij} + S_{ij} + O(p^{-3/2}).$$

where we denote the shortcut $\lambda_{\ell-1} \equiv \tau_{\ell-1}^2 - \frac{d_{\ell-1}^2}{\tau_{\ell-1}^2} = O(1)$.

We thus get, for non-diagonal entries $[\mathbf{K}_{\mathrm{CK},\ell}]_{ij}$, $i \neq j$, that

$$[\mathbf{K}_{\mathrm{CK},\ell}]_{ij}$$

$$= \mathbb{E}\left[\sigma_\ell\left(\sqrt{[\mathbf{K}_{\mathrm{CK},\ell-1}]_{ii}} \cdot \xi_i\right)\sigma_\ell\left(\frac{[\mathbf{K}_{\mathrm{CK},\ell-1}]_{ij}}{\sqrt{[\mathbf{K}_{\mathrm{CK},\ell-1}]_{ii}}} \cdot \xi_i + \sqrt{[\mathbf{K}_{\mathrm{CK},\ell-1}]_{jj} - \frac{[\mathbf{K}_{\mathrm{CK},\ell-1}]_{ij}^2}{[\mathbf{K}_{\mathrm{CK},\ell-1}]_{ii}}} \cdot \xi_j\right)\right]$$

$$= \mathbb{E}\left[\sigma_\ell\left(\tau_{\ell-1}\xi_i + P_{ij}\xi_i\right) \times \sigma_\ell\left(\left(\frac{d_{\ell-1}}{\tau_{\ell-1}} + \frac{\alpha_{\ell-1,1}}{\tau_{\ell-1}}A_{ij} + S_{ij} + P_{ij}\right)\xi_i\right.\right.$$

$$\left.\left. + \left(\sqrt{\lambda_{\ell-1}} - \frac{1}{\sqrt{\lambda_{\ell-1}}}\frac{d_{\ell-1} \cdot \alpha_{\ell-1,1}}{\tau_{\ell-1}^2}A_{ij} + S_{ij} + P_{ij}\right)\xi_j\right)\right] + O(p^{-3/2})$$

$$
= \mathbb{E}\Bigg[\left(\sigma_\ell\left(\tau_{\ell-1}\xi_i\right) + \sigma_\ell'\left(\tau_{\ell-1}\xi_i\right)P_{ij}\xi_i + \frac{1}{2}\sigma_\ell''\left(\tau_{\ell-1}\xi_i\right)P_{ij}^2\xi_i^2\right)
$$

$$
\times \Bigg(\sigma_\ell\left(\frac{d_{\ell-1}}{\tau_{\ell-1}}\xi_i + \sqrt{\lambda_{\ell-1}}\xi_j\right)
$$

$$
+ \sigma_\ell'\left(\frac{d_{\ell-1}}{\tau_{\ell-1}}\xi_i + \sqrt{\lambda_{\ell-1}}\xi_j\right)\left(\left(\frac{\alpha_{\ell-1,1}}{\tau_{\ell-1}}\xi_i - \frac{1}{\sqrt{\lambda_{\ell-1}}}\frac{d_{\ell-1}\cdot\alpha_{\ell-1,1}}{\tau_{\ell-1}^2}\xi_j\right)\cdot A_{ij} + S_{ij} + P_{ij}\right)
$$

$$
+ \frac{1}{2}\sigma_\ell''\left(\frac{d_{\ell-1}}{\tau_{\ell-1}}\xi_i + \sqrt{\lambda_{\ell-1}}\xi_j\right)\left(S_{ij} + P_{ij}\right)\Bigg)\Bigg] + O(p^{-3/2})
$$

$$
= \mathbb{E}\Bigg[\left(\sigma_\ell\left(\tau_{\ell-1}\xi_i\right) + P_{ij}\right)
$$

$$
\times \Bigg(\sigma_\ell\left(\frac{d_{\ell-1}}{\tau_{\ell-1}}\xi_i + \sqrt{\lambda_{\ell-1}}\xi_j\right)
$$

$$
+ \sigma_\ell'\left(\frac{d_{\ell-1}}{\tau_{\ell-1}}\xi_i + \sqrt{\lambda_{\ell-1}}\xi_j\right)\left(\frac{\alpha_{\ell-1,1}}{\tau_{\ell-1}}\xi_i - \frac{1}{\sqrt{\lambda_{\ell-1}}}\frac{d_{\ell-1}\cdot\alpha_{\ell-1,1}}{\tau_{\ell-1}^2}\xi_j\right)\cdot A_{ij} + S_{ij} + P_{ij}\Bigg)\Bigg]
$$

$$
+ O(p^{-3/2})
$$

$$
= \mathbb{E}\Bigg[\sigma_\ell\left(\tau_{\ell-1}\xi_i\right)\times\sigma_\ell\left(\frac{d_{\ell-1}}{\tau_{\ell-1}}\xi_i + \sqrt{\lambda_{\ell-1}}\xi_j\right)
$$

$$
+ \sigma_\ell\left(\tau_{\ell-1}\xi_i\right)\times\sigma_\ell'\left(\frac{d_{\ell-1}}{\tau_{\ell-1}}\xi_i + \sqrt{\lambda_{\ell-1}}\xi_j\right)\left(\frac{\alpha_{\ell-1,1}}{\tau_{\ell-1}}\xi_i - \frac{1}{\sqrt{\lambda_{\ell-1}}}\frac{d_{\ell-1}\cdot\alpha_{\ell-1,1}}{\tau_{\ell-1}^2}\xi_j\right)\cdot A_{ij}
$$

$$
+ S_{ij} + P_{ij}\Bigg] + O(p^{-3/2})
$$

$$
= \mathbb{E}\left[\sigma_\ell\left(\tau_{\ell-1}\xi_i\right)\times\sigma_\ell\left(\frac{d_{\ell-1}}{\tau_{\ell-1}}\xi_i + \sqrt{\lambda_{\ell-1}}\xi_j\right)\right]
$$

$$
+ \mathbb{E}\left[\sigma_\ell\left(\tau_{\ell-1}\xi_i\right)\times\sigma_\ell'\left(\frac{d_{\ell-1}}{\tau_{\ell-1}}\xi_i + \sqrt{\lambda_{\ell-1}}\xi_j\right)\left(\frac{\alpha_{\ell-1,1}}{\tau_{\ell-1}}\xi_i - \frac{1}{\sqrt{\lambda_{\ell-1}}}\frac{d_{\ell-1}\cdot\alpha_{\ell-1,1}}{\tau_{\ell-1}^2}\xi_j\right)\right]\cdot A_{ij}
$$

$$
+ S_{ij} + P_{ij} + O(p^{-3/2})
$$

$$
\equiv d_\ell + \alpha_{\ell,1}A_{ij} + S_{ij} + P_{ij} + O(p^{-3/2}),
$$

where we recall the followings recursive relation

$$
d_\ell = \mathbb{E}\left[\sigma_\ell\left(\tau_{\ell-1}\xi_i\right)\sigma_\ell\left(\frac{d_{\ell-1}}{\tau_{\ell-1}}\xi_i + \sqrt{\lambda_{\ell-1}}\xi_j\right)\right],
$$

and

$$
\alpha_{\ell,1} = \mathbb{E}\left[\sigma_\ell\left(\tau_{\ell-1}\xi_i\right)\sigma_\ell'\left(\frac{d_{\ell-1}}{\tau_{\ell-1}}\xi_i + \sqrt{\lambda_{\ell-1}}\xi_j\right)\left(\frac{\alpha_{\ell-1,1}}{\tau_{\ell-1}}\xi_i - \frac{1}{\sqrt{\lambda_{\ell-1}}}\frac{d_{\ell-1}\cdot\alpha_{\ell-1,1}}{\tau_{\ell-1}^2}\xi_j\right)\right],
$$

with $\lambda_{\ell-1} = \tau_{\ell-1}^2 - d_{\ell-1}^2/\tau_{\ell-1}^2$ and for independent $\xi_i, \xi_j \sim \mathcal{N}(0,1)$. This concludes the proof of Theorem 3. $\qquad\square$

Note that in the defining equation of $d_\ell$, one needs to take the square root of $\lambda_\ell$. In the following, we show that $\tau_\ell^4 \geq d_\ell^2$ holds for all $\ell \in \{0, 1, \dots, L\}$ and the sequence $d_\ell$ above is well-defined.

**Remark 5** (On the existence of $d_\ell$). *Note in Theorem 3 that the sequence $d_\ell$, $\ell \in \{1, \dots, L\}$, is defined in a recursive manner, and needs to take the square root $\sqrt{\lambda_{\ell-1}} = \sqrt{\tau_\ell^2 - d_\ell^2/\tau_\ell^2}$ which may not be well defined. Here, we prove that $\tau_\ell^2 - d_\ell^2/\tau_\ell^2 \geq 0$ holds for all $\ell$, which further guarantees the existence of the sequence $d_\ell$.*

*By definition, we have*

$$d_\ell = \mathbb{E}_{\xi_1, \xi_2} \left[ \sigma_\ell \left( \tau_{\ell-1} \xi_1 \right) \sigma_\ell \left( \frac{d_{\ell-1}}{\tau_{\ell-1}} \xi_1 + \sqrt{\lambda_{\ell-1}} \xi_2 \right) \right], \tag{57}$$

*for independent $\xi_1, \xi_2 \sim \mathcal{N}(0, 1)$, which takes a form similar to the Gram–Schmidt orthogonalization performed in the proof of Theorem 1 in Section A.1. We thus have the following alternative definition*

$$d_\ell = \mathbb{E}_{u,v} \left[ \sigma_\ell(u) \sigma(v) \right], \tag{58}$$

*for*

$$(u, v) \sim \mathcal{N} \left( \mathbf{0}, \begin{bmatrix} \tau_{\ell-1}^2 & d_{\ell-1} \\ d_{\ell-1} & \tau_{\ell-1}^2 \end{bmatrix} \right), \tag{59}$$

*which is in essence the leading order of the non-diagonal term $[\mathbf{K}_{\mathrm{CK},\ell}]_{ij}$, $i \neq j$. It then follows from the Cauchy–Schwarz inequality that $\tau_{\ell-1}^4 \geq d_{\ell-1}^2$ and thus the conclusion.*

### A.2.2 Proof of Theorem 2

In the following proof, we follow the same notations and working assumptions as in the proof of Theorem 1 and Theorem 3 in Appendix A.1 and Appendix A.2.1, respectively.

We have obtained the eigenspectral behavior (in form of a layer-by-layer recurrence relation) of the CK matrices $\mathbf{K}_{\mathrm{CK},\ell}$, for $\mathbb{E}\left[ \sigma_\ell \left( \tau_{\ell-1} \xi \right) \right] = 0$ in Theorem 1 and more generally in Theorem 3.

As already mentioned in (6), the NTK matrices $\mathbf{K}_{\mathrm{NTK},\ell}$ of layer $\ell$ can be defined again, in an iterative manner, via the CK matrices $\mathbf{K}_{\mathrm{CK},\ell}$ and $\mathbf{K}'_{\mathrm{CK},\ell}$ as follows [28]

$$\mathbf{K}_{\mathrm{NTK},0} = \mathbf{K}_{\mathrm{CK},0} = \mathbf{X}^\mathsf{T} \mathbf{X},$$
$$\mathbf{K}_{\mathrm{NTK},\ell} = \mathbf{K}_{\mathrm{CK},\ell} + \mathbf{K}_{\mathrm{NTK},\ell-1} \circ \mathbf{K}'_{\mathrm{CK},\ell}.$$

where '$\mathbf{A} \circ \mathbf{B}$' denotes the Hadamard product between two matrices $\mathbf{A}, \mathbf{B}$ of the same size, and $\mathbf{K}'_{\mathrm{CK},\ell} \in \mathbb{R}^{n \times n}$ denotes the CK matrix with nonlinear function $\sigma'_\ell(\cdot)$ instead of $\sigma_\ell(\cdot)$ (as for $\mathbf{K}_{\mathrm{CK},\ell}$ in (4)), that is

$$[\mathbf{K}'_{\mathrm{CK},\ell}]_{ij} = \mathbb{E}_{u,v}[\sigma'_\ell(u)\sigma'_\ell(v)], \quad u, v \sim \mathcal{N} \left( \mathbf{0}, \begin{bmatrix} [\mathbf{K}'_{\mathrm{CK},\ell-1}]_{ii} & [\mathbf{K}'_{\mathrm{CK},\ell-1}]_{ij} \\ [\mathbf{K}'_{\mathrm{CK},\ell-1}]_{ij} & [\mathbf{K}'_{\mathrm{CK},\ell-1}]_{jj} \end{bmatrix} \right). \tag{60}$$

For $\mathbb{E}\left[ \sigma_\ell(\tau_{\ell-1}\xi) \right] = 0$, we have, by Theorem 1, that the CK matrices $\mathbf{K}_{\mathrm{CK},\ell} \simeq \tilde{\mathbf{K}}_{\mathrm{CK},\ell}$ in a spectral sense, with specifically

$$[\mathbf{K}_{\mathrm{CK},\ell}]_{ii} = \tau_\ell^2 + O(p^{-1/2}), \tag{61}$$

and for $i \neq j$,

$$[\mathbf{K}_{\mathrm{CK},\ell}]_{ij} = \alpha_{\ell,1} A_{ij} + \alpha_{\ell,2}(t_i + \psi_i)(t_j + \psi_j) + \alpha_{\ell,3} \left( \frac{1}{p} \mathbf{z}_i^\mathsf{T} \mathbf{z}_j \right)^2 + S_{ij} + O(p^{-3/2}). \tag{62}$$

For $\mathbf{K}'_{\mathrm{CK},\ell}$, one has in general $\mathbb{E}\left[ \sigma'_\ell(\dot{\tau}_{\ell-1}\xi) \right] \neq 0$ for the sequence $\dot{\tau}_\ell$ defined in (12) as follows:

$$\dot{\tau}_\ell = \sqrt{\mathbb{E}\left[ \left( \sigma'_\ell(\dot{\tau}_{\ell-1}\xi) \right)^2 \right]}, \quad \xi \sim \mathcal{N}(0, 1), \quad \ell \in \{1, \dots, L\}, \tag{63}$$

with $\dot{\tau}_0 = \tau_0$. We thus resort to Theorem 3 for the entries of $\mathbf{K}'_{\mathrm{CK},\ell}$. Specifically, we have that

$$[\mathbf{K}'_{\mathrm{CK},\ell}]_{ii} = \dot{\tau}_\ell^2 + O(p^{-1/2}), \tag{64}$$

and for $i \neq j$,

$$[\mathbf{K}'_{\mathrm{CK},\ell}]_{ij} = \dot{d}_\ell + \alpha_{\dot{\ell},1} A_{ij} + S_{ij} + P_{ij} + O(p^{-3/2}), \tag{65}$$

with

$$\dot{d}_\ell = \mathbb{E}_{\xi_1, \xi_2} \left[ \sigma'_\ell(\dot{\tau}_{\ell-1}\xi_1)\sigma'_\ell \left( \frac{\dot{d}_{\ell-1}}{\dot{\tau}_{\ell-1}} \xi_1 + \sqrt{\dot{\lambda}_{\ell-1}} \xi_2 \right) \right],$$

$$\dot{\alpha}_{\ell,1} = \mathbb{E}_{\xi_1, \xi_2} \left[ \sigma'_\ell(\dot{\tau}_{\ell-1}\xi_1)\sigma''_\ell \left( \frac{\dot{d}_{\ell-1}}{\dot{\tau}_{\ell-1}} \xi_1 + \sqrt{\dot{\lambda}_{\ell-1}} \xi_2 \right) \left( \frac{\dot{\alpha}_{\ell-1,1}}{\dot{\tau}_{\ell-1}} \xi_1 - \frac{1}{\sqrt{\dot{\lambda}_{\ell-1}}} \frac{\dot{d}_{\ell-1}\dot{\alpha}_{\ell-1,1}}{\dot{\tau}_{\ell-1}^2} \xi_2 \right) \right],$$

with $\dot{\lambda}_{\ell-1} = \dot{\tau}_{\ell-1}^2 - \dot{d}_{\ell-1}^2/\dot{\tau}_{\ell-1}^2$ for independent $\xi_1, \xi_2 \sim \mathcal{N}(0,1)$.

With the above results at hand, we now proceed to the proof of Theorem 2. As in the proof of Theorem 1 in Appendix A.1, we follow the three-step proof strategy to work on the non-diagonal, the diagonal, and eventually the matrix form of $\mathbf{K}_{\mathrm{NTK},\ell}$.

**Off the diagonal.** With (62) and (65), We first write the non-diagonal entries of $\mathbf{K}_{\mathrm{NTK},1}$ as

$$
\begin{aligned}
[\mathbf{K}_{\mathrm{NTK},1}]_{ij} &= [\mathbf{K}_{\mathrm{CK},1}]_{ij} + [\mathbf{K}_{\mathrm{NTK},0}]_{ij}[\mathbf{K}'_{\mathrm{CK},1}]_{ij} \\
&= \alpha_{1,1}A_{ij} + \alpha_{1,2}(t_i + \psi_i)(t_j + \psi_j) + \alpha_{1,3}\left(\frac{1}{p}\mathbf{z}_i^\mathsf{T}\mathbf{z}_j\right)^2 \\
&\quad + \left(\beta_{0,1}A_{ij} + \beta_{0,2}(t_i + \psi_i)(t_j + \psi_j) + \beta_{0,3}\left(\frac{1}{p}\mathbf{z}_i^\mathsf{T}\mathbf{z}_j\right)^2\right) \\
&\quad \times \left(\dot{d}_1 + \dot{\alpha}_{1,1}A_{ij} + P_{ij} + S_{ij}\right) \\
&\quad + O(p^{-3/2}) \\
&= (\alpha_{1,1} + \beta_{0,1} \cdot \dot{d}_1)A_{ij} + (\alpha_{1,2} + \beta_{0,2} \cdot \dot{d}_1)(t_i + \psi_i)(t_j + \psi_j) \\
&\quad + (\alpha_{1,3} + \beta_{0,3} \cdot \dot{d}_1 + \beta_{0,1}\dot{\alpha}_{1,1})\left(\frac{1}{p}\mathbf{z}_i^\mathsf{T}\mathbf{z}_j\right)^2 \\
&\quad + S_{ij} + O(p^{-3/2}),
\end{aligned}
$$

so that $[\mathbf{K}_{\mathrm{NTK},1}]_{ij}$ (and thus $[\mathbf{K}_{\mathrm{NTK},\ell}]_{ij}$ for $\ell \in \{1,\ldots,L\}$) *must* also take the form

$$
[\mathbf{K}_{\mathrm{NTK},\ell}]_{ij} = \beta_{\ell,1}A_{ij} + \beta_{\ell,2}(t_i + \psi_i)(t_j + \psi_j) + \beta_{\ell,3}\left(\frac{1}{p}\mathbf{z}_i^\mathsf{T}\mathbf{z}_j\right)^2 + S_{ij} + O(p^{-3/2}). \tag{66}
$$

Since

$$
\begin{aligned}
[\mathbf{K}_{\mathrm{NTK},\ell}]_{ij} &= [\mathbf{K}_{\mathrm{CK},\ell}]_{ij} + [\mathbf{K}_{\mathrm{NTK},\ell-1}]_{ij}[\mathbf{K}'_{\mathrm{CK},\ell}]_{ij} \\
&= \alpha_{\ell,1}A_{ij} + \alpha_{\ell,2}(t_i + \psi_i)(t_j + \psi_j) + \alpha_{\ell,3}\left(\frac{1}{p}\mathbf{z}_i^\mathsf{T}\mathbf{z}_j\right)^2 \\
&\quad + \left(\beta_{\ell-1,1}A_{ij} + \beta_{\ell-1,2}(t_i + \psi_i)(t_j + \psi_j) + \beta_{\ell-1,3}\left(\frac{1}{p}\mathbf{z}_i^\mathsf{T}\mathbf{z}_j\right)^2\right) \\
&\quad \times \left(\dot{d}_\ell + \dot{\alpha}_{\ell,1}A_{ij} + P_{ij} + S_{ij}\right) + O(p^{-3/2}) \\
&= (\alpha_{\ell,1} + \beta_{\ell-1,1} \cdot \dot{d}_\ell)A_{ij} + (\alpha_{\ell,2} + \beta_{\ell-1,2} \cdot \dot{d}_\ell)(t_i + \psi_i)(t_j + \psi_j) \\
&\quad + (\alpha_{\ell,3} + \beta_{\ell-1,3} \cdot \dot{d}_\ell + \beta_{\ell-1,1} \cdot \dot{\alpha}_{\ell,1})\left(\frac{1}{p}\mathbf{z}_i^\mathsf{T}\mathbf{z}_j\right)^2 \\
&\quad + S_{ij} + O(p^{-3/2}),
\end{aligned}
$$

so that

$$
\beta_{\ell,1} = \alpha_{\ell,1} + \beta_{\ell-1,1} \cdot \dot{d}_\ell, \tag{67}
$$

$$
\beta_{\ell,2} = \alpha_{\ell,2} + \beta_{\ell-1,2} \cdot \dot{d}_\ell, \tag{68}
$$

$$
\beta_{\ell,3} = \alpha_{\ell,3} + \beta_{\ell-1,3} \cdot \dot{d}_\ell + \beta_{\ell-1,1} \cdot \dot{\alpha}_{\ell,1}, \tag{69}
$$

with

$$
\dot{d}_\ell = \mathbb{E}_{\xi_1,\xi_2}\left[\sigma'_\ell(\dot{\tau}_{\ell-1}\xi_1)\sigma'_\ell\left(\frac{\dot{d}_{\ell-1}}{\dot{\tau}_{\ell-1}}\xi_1 + \sqrt{\dot{\lambda}_{\ell-1}}\xi_2\right)\right],
$$

$$
\dot{\alpha}_{\ell,1} = \mathbb{E}_{\xi_1,\xi_2}\left[\sigma'_\ell(\dot{\tau}_{\ell-1}\xi_1)\sigma''_\ell\left(\frac{\dot{d}_{\ell-1}}{\dot{\tau}_{\ell-1}}\xi_1 + \sqrt{\dot{\lambda}_{\ell-1}}\xi_2\right)\left(\frac{\dot{\alpha}_{\ell-1,1}}{\dot{\tau}_{\ell-1}}\xi_1 - \frac{1}{\sqrt{\dot{\lambda}_{\ell-1}}}\frac{\dot{d}_{\ell-1}\dot{\alpha}_{\ell-1,1}}{\dot{\tau}_{\ell-1}^2}\xi_2\right)\right],
$$

with $\dot{\lambda}_{\ell-1} = \dot{\tau}_{\ell-1}^2 - \frac{d_{\ell-1}^2}{\dot{\tau}_{\ell-1}^2}$ for independent $\xi_1, \xi_2 \sim \mathcal{N}(0,1)$.

**On the diagonal.** With (61) and (64), we next evaluate the diagonal entries of $\mathbf{K}_{\mathrm{NTK},\ell}$ as

$$[\mathbf{K}_{\mathrm{NTK},\ell}]_{ii} = [\mathbf{K}_{\mathrm{CK},\ell}]_{ii} + [\mathbf{K}_{\mathrm{NTK},\ell-1}]_{ii} \cdot [\mathbf{K}'_{\mathrm{CK},\ell}]_{ii} = \tau_\ell^2 + \kappa_{\ell-1}^2 \dot{\tau}_\ell^2 + O(p^{-1/2}),$$

so that we obtain the following relation

$$\kappa_\ell^2 = \tau_\ell^2 + \kappa_{\ell-1}^2 \dot{\tau}_\ell^2, \tag{70}$$

with $\kappa_0 = \tau_0 = \sqrt{\operatorname{tr} \mathbf{C}^\circ / p}$ defined in Assumption 1.

**Assembling in matrix form.** Following the discussion above, we have, uniformly for $i \neq j \in \{1, \ldots, n\}$ that, Putting everything together in matrix form, we obtain, as in the proof of Theorem 1 in Appendix A.1 that

$$[\mathbf{K}_{\mathrm{NTK},\ell}]_{ij} = \beta_{\ell,1} A_{ij} + \beta_{\ell,2}(t_i + \psi_i)(t_j + \psi_j) + \beta_{\ell,3}\left(\frac{1}{p}\mathbf{z}_i^\mathsf{T}\mathbf{z}_j\right)^2 + S_{ij} + O(p^{-3/2}),$$

and

$$[\mathbf{K}_{\mathrm{NTK},\ell}]_{ii} = [\mathbf{K}_{\mathrm{CK},\ell}]_{ii} + [\mathbf{K}_{\mathrm{NTK},\ell-1}]_{ii} \cdot [\mathbf{K}'_{\mathrm{CK},\ell}]_{ii} = \tau_\ell^2 + \kappa_{\ell-1}^2 \dot{\tau}_\ell^2 + O(p^{-1/2}) \equiv \kappa_\ell^2 + O(p^{-1/2}),$$

so that in matrix form:

$$\mathbf{K}_{\mathrm{NTK},\ell} = \beta_{\ell,1}\mathbf{X}^\mathsf{T}\mathbf{X} + \mathbf{V}\mathbf{B}_\ell\mathbf{V}^\mathsf{T} + (\kappa_\ell^2 - \tau_0^2\beta_{\ell,1} - \tau_0^4\beta_{\ell,3})\mathbf{I}_n + O_{\|\cdot\|}(p^{-\frac{1}{2}}), \tag{71}$$

where $O_{\|\cdot\|}(p^{-1/2})$ denotes matrices of spectral norm order $O(p^{-1/2})$ as $n, p \to \infty$, with

$$\mathbf{V} = [\mathbf{J}/\sqrt{p}, \, \boldsymbol{\psi}] \in \mathbb{R}^{n \times (K+1)}, \quad \mathbf{B}_\ell = \begin{bmatrix} \beta_{\ell,2}\mathbf{t}\mathbf{t}^\mathsf{T} + \beta_{\ell,3}\mathbf{T} & \beta_{\ell,2}\mathbf{t} \\ \beta_{\ell,2}\mathbf{t}^\mathsf{T} & \beta_{\ell,2} \end{bmatrix}, \tag{72}$$

which concludes the proof of Theorem 2.

## A.3 Two equivalent centering approaches in the single-hidden-layer case

In this section, we aim to show that "centering" the CK matrices $\mathbf{K}_{\mathrm{CK}}$ by pre- and post-multiplying $\mathbf{P} = \mathbf{I}_n - \mathbf{1}_n\mathbf{1}_n^\mathsf{T}/n$ performed in [4, Theorem 1] is *equivalent* to take $\mathbb{E}[\sigma(\tau_0\xi)] = 0$ as in our Theorem 1 in the single-hidden-layer $\ell = 1$ setting, in the sense that one has

$$\|\mathbf{P}(\mathbf{K}_{\mathrm{CK},1} - \tilde{\mathbf{K}}_{\mathrm{CK},1})\mathbf{P}\| \to 0, \tag{73}$$

almost surely as $n, p \to \infty$, for the *same* $\tilde{\mathbf{K}}_{\mathrm{CK},1}$ as defined in Theorem 1 and an *arbitrary* choice of $\mathbb{E}[\sigma(\tau_0\xi)]$ (so in particular, one may freely take $\mathbb{E}[\sigma(\tau_0\xi)] \neq 0$ which is different from the setting of our Theorem 1). The proof is as follows.

First note that the assumption $\mathbb{E}[\sigma(\tau_0\xi)] = 0$ is *only* used for the off-diagonal entries of the CK matrix $\mathbf{K}_{\mathrm{CK},1}$, so we focus, in the sequel, only on the off-diagonal terms, while the discussions on the on-diagonal entries are the same as in Appendix A.1.

By its definition in (5) and the fact that $\mathbf{K}_{\mathrm{CK},0} = \mathbf{X}^\mathsf{T}\mathbf{X}$, one has

$$[\mathbf{K}_{\mathrm{CK},1}]_{ij} = \mathbb{E}_{u,v}[\sigma_1(u)\sigma_1(v)], \text{ with } u, v \sim \mathcal{N}\left(\mathbf{0}, \begin{bmatrix} \|\mathbf{x}_i\|^2 & \mathbf{x}_i^\mathsf{T}\mathbf{x}_j \\ \mathbf{x}_i^\mathsf{T}\mathbf{x}_j & \|\mathbf{x}_j\|^2 \end{bmatrix}\right), \tag{74}$$

so by performing a Gram-Schmidt orthogonalization procedure as in the proof of Theorem 1 in Appendix A.1, one has

$$u = \|\mathbf{x}_i\| \cdot \xi_i, \quad v = \|\mathbf{x}_j\|\left(\angle_{ij} \cdot \xi_i + \sqrt{1 - \angle_{ij}^2} \cdot \xi_j\right), \tag{75}$$

for two *independent* standard Gaussian random variables $\xi_i$ and $\xi_j$, where we denote the shortcut $\angle_{ij} \equiv \frac{\mathbf{x}_i^\mathsf{T}\mathbf{x}_j}{\|\mathbf{x}_i\| \cdot \|\mathbf{x}_j\|}$ for the "angle" between data vectors $\mathbf{x}_i$ and $\mathbf{x}_j$.

It can be checked, for $\mathbf{x}_i = \boldsymbol{\mu}_i/\sqrt{p} + \mathbf{z}_i/\sqrt{p}$ with $\mathbb{E}[\mathbf{z}_i] = \mathbf{0}$ and $\mathbb{E}[\mathbf{z}_i \mathbf{z}_i^{\mathsf{T}}] = \mathbf{C}_i$ that

$$\|\mathbf{x}_i\|^2 = \frac{1}{p}(\boldsymbol{\mu}_i + \mathbf{z}_i)^{\mathsf{T}}(\boldsymbol{\mu}_i + \mathbf{z}_i) = \frac{1}{p}\|\boldsymbol{\mu}_i\|^2 + \frac{2}{p}\boldsymbol{\mu}_i^{\mathsf{T}}\mathbf{z}_i + \frac{1}{p}\mathbf{z}_i^{\mathsf{T}}\mathbf{z}_i$$

$$= \frac{1}{p}\|\boldsymbol{\mu}_i\|^2 + \underbrace{\frac{2}{p}\boldsymbol{\mu}_i^{\mathsf{T}}\mathbf{z}_i}_{O(p^{-1})} + \underbrace{\frac{1}{p}\operatorname{tr}\mathbf{C}^{\circ}}_{\equiv \tau_0^2 = O(1)} + \underbrace{\frac{1}{p}\operatorname{tr}\mathbf{C}_i^{\circ}}_{\equiv t_i = O(p^{-1/2})} + \underbrace{\psi_i}_{O(p^{-1/2})} \ ,$$

where we recall the definition $\psi_i \equiv \frac{1}{p}\|\mathbf{z}_i\|^2 - \frac{1}{p}\operatorname{tr}\mathbf{C}_i = O(p^{-1/2})$. As such, by Taylor-expanding $\sqrt{\|\mathbf{x}_i\|^2}$ around $\|\mathbf{x}_i\|^2 \simeq \tau_0^2 = O(1)$, we get

$$\|\mathbf{x}_i\| = \tau_0 + \frac{1}{2\tau_0}(\|\boldsymbol{\mu}_i\|^2/p + 2\boldsymbol{\mu}_i^{\mathsf{T}}\mathbf{z}_i/p + t_i + \psi_i) - \frac{1}{8\tau_0^3}(t_i + \psi_i)^2 + O(p^{-3/2})$$

$$\equiv \tau_0 + \theta_i + O(p^{-3/2}),$$

where we denote the shortcut

$$\theta_i \equiv \frac{1}{2\tau_0}(\|\boldsymbol{\mu}_i\|^2/p + 2\boldsymbol{\mu}_i^{\mathsf{T}}\mathbf{z}_i/p + t_i + \psi_i) - \frac{1}{8\tau_0^3}(t_i + \psi_i)^2 = O(p^{-1/2}), \tag{76}$$

so that

$$\|\mathbf{x}_j\|\angle_{ij} = \frac{\frac{1}{p}(\boldsymbol{\mu}_i + \mathbf{z}_i)^{\mathsf{T}}(\boldsymbol{\mu}_j + \mathbf{z}_j)}{\|\mathbf{x}_i\|}$$

$$= \frac{\frac{1}{p}\boldsymbol{\mu}_i^{\mathsf{T}}\boldsymbol{\mu}_j + \frac{1}{p}(\boldsymbol{\mu}_i^{\mathsf{T}}\mathbf{z}_j + \boldsymbol{\mu}_j^{\mathsf{T}}\mathbf{z}_i) + \frac{1}{p}\mathbf{z}_i^{\mathsf{T}}\mathbf{z}_j}{\tau_0 + \frac{1}{2\tau_0}(t_i + \psi_i) + O(p^{-1})} + O(p^{-3/2})$$

$$= \left(\frac{1}{\tau_0} - \frac{t_i + \psi_i}{2\tau_0^3} + O(p^{-1})\right)\left(\frac{1}{p}\boldsymbol{\mu}_i^{\mathsf{T}}\boldsymbol{\mu}_j + \frac{1}{p}(\boldsymbol{\mu}_i^{\mathsf{T}}\mathbf{z}_j + \boldsymbol{\mu}_j^{\mathsf{T}}\mathbf{z}_i) + \frac{1}{p}\mathbf{z}_i^{\mathsf{T}}\mathbf{z}_j\right) + O(p^{-3/2})$$

$$= \frac{1}{\tau_0}\left(\frac{1}{p}\mathbf{z}_i^{\mathsf{T}}\mathbf{z}_j + \frac{1}{p}\boldsymbol{\mu}_i^{\mathsf{T}}\boldsymbol{\mu}_j + \frac{1}{p}(\boldsymbol{\mu}_i^{\mathsf{T}}\mathbf{z}_j + \boldsymbol{\mu}_j^{\mathsf{T}}\mathbf{z}_i) + S_{ij}\right) + O(p^{-3/2})$$

$$= \frac{1}{\tau_0}A_{ij} + S_{ij} + O(p^{-3/2}).$$

Therefore, again by Taylor-expansion,

$$\sqrt{\|\mathbf{x}_j\|^2 - (\|\mathbf{x}_j\|\angle_{ij})^2} = \sqrt{(\|\boldsymbol{\mu}_j\|^2/p + 2\boldsymbol{\mu}_j^{\mathsf{T}}\mathbf{z}_j/p + \tau_0^2 + t_j + \psi_j) - (A_{ij}/\tau_0 + S_{ij})^2}$$

$$= \tau_0 + \frac{1}{2\tau_0}(\|\boldsymbol{\mu}_i\|^2/p + 2\boldsymbol{\mu}_i^{\mathsf{T}}\mathbf{z}_i/p + t_i + \psi_i) - \frac{1}{8\tau_0^3}(t_i + \psi_i)^2$$

$$- \frac{1}{2\tau_0^3}\left(\frac{1}{p}\mathbf{z}_i^{\mathsf{T}}\mathbf{z}_j\right)^2 + S_{ij} + O(p^{-3/2})$$

$$= \tau_0 + \theta_j - \frac{1}{2\tau_0^3}\left(\frac{1}{p}\mathbf{z}_i^{\mathsf{T}}\mathbf{z}_j\right)^2 + S_{ij} + O(p^{-3/2}).$$

Following the same idea, we again Taylor-expand $\sigma_1(\cdot)$ in the definition of $\mathbf{K}_{\mathrm{CK},1}$ as

$$\sigma_1(u) = \sigma_1(\tau_0\xi_i) + \sigma_1'(\tau_0\xi_i)\xi_i\theta_i + \frac{1}{8\tau_0^2}\sigma_1''(\tau_0\xi_i)\xi_i^2(t_i + \psi_i)^2 + O(p^{-3/2}),$$

and

$$\sigma_1(v) = \sigma_1\left(\|\mathbf{x}_j\|\angle_{ij}\xi_j + \|\mathbf{x}_j\|\sqrt{1 - \angle_{ij}^2}\xi_i\right)$$

$$= \sigma_1\left(\tau_0\xi_j + \xi_j\theta_j - \xi_j\frac{1}{2\tau_0^3}\left(\frac{1}{p}\mathbf{z}_i^{\mathsf{T}}\mathbf{z}_j\right)^2 + \xi_i\frac{1}{\tau_0}A_{ij} + S_{ij} + O(p^{-3/2})\right)$$

$$= \sigma_1(\tau_0\xi_j) + \sigma_1'(\tau_0\xi_j)\xi_j\theta_j + \frac{1}{8\tau_0^2}\sigma_1''(\tau_0\xi_j)\xi_j^2(t_j + \psi_j)^2 + X_{ij} + O(p^{-3/2}),$$

with

$$X_{ij} = \frac{1}{\tau_0}\xi_i\sigma_1'(\tau_0\xi_j)A_{ij} + \frac{1}{2\tau_0^2}\left(\frac{1}{p}\mathbf{z}_i^\mathsf{T}\mathbf{z}_j\right)^2\left(\xi_i^2\sigma_1''(\tau_0\xi_j) - \frac{1}{\tau_0}\xi_j\sigma_1'(\tau_0\xi_j)\right) + S_{ij} = O(p^{-1/2}),$$

where we recall the definition

$$A_{ij} = \underbrace{\frac{1}{p}\mathbf{z}_i^\mathsf{T}\mathbf{z}_j}_{O(p^{-1/2})} + \underbrace{\frac{1}{p}\boldsymbol{\mu}_i^\mathsf{T}\boldsymbol{\mu}_j + \frac{1}{p}(\boldsymbol{\mu}_i^\mathsf{T}\mathbf{z}_j + \boldsymbol{\mu}_j^\mathsf{T}\mathbf{z}_i)}_{O(p^{-1})} = \mathbf{x}_i^\mathsf{T}\mathbf{x}_j. \tag{77}$$

For independent $\xi_i$ and $\xi_j$, we denote the following coefficients

$$p_0 = \mathbb{E}[\sigma_1(\tau_0\xi)], \quad p_1 = \mathbb{E}[\sigma_1'(\tau_0\xi)], \quad p_2 = \mathbb{E}[\sigma_1''(\tau_0\xi)], \quad p_3 = \mathbb{E}[\sigma_1'''(\tau_0\xi)], \tag{78}$$

so that

$$\mathbb{E}[\xi\sigma_1(\tau_0\xi)] = \tau_0\mathbb{E}[\sigma_1'(\tau_0\xi)] = \tau_0 p_1, \quad \mathbb{E}[\xi\sigma_1'(\tau_0\xi)] = \tau_0 p_2, \tag{79}$$

as well as

$$\mathbb{E}[\xi^2\sigma_1''(\tau_0\xi)] = \mathbb{E}[(\xi^2-1)\sigma_1''(\tau_0\xi)] + p_2 = \tau_0^2\mathbb{E}[\sigma_1''''(\tau_0\xi)] + p_2 \equiv \tau_0^2 p_4 + p_2, \tag{80}$$

for $p_4 = \mathbb{E}[\sigma_1''''(\tau_0\xi)]$.

This further allows us to write, for $A_{ij} = O(p^{-1/2})$ and $\theta_j = O(p^{-1/2})$ that

$$\begin{aligned}
[\mathbf{K}_{\mathrm{CK},1}]_{ij} &= \mathbb{E}_{u,v}[\sigma_1(u)\sigma_1(v)]\\
&= \mathbb{E}\left[\sigma_1(\tau_0\xi_i) + \sigma_1'(\tau_0\xi_i)\xi_i\theta_i + \frac{1}{8\tau_0^2}\sigma_1''(\tau_0\xi_i)\xi_i^2(t_i+\psi_i)^2\right]\\
&\quad \times \mathbb{E}\left[\sigma_1(\tau_0\xi_j) + \sigma_1'(\tau_0\xi_j)\xi_j\theta_j + \frac{1}{8\tau_0^2}\sigma_1''(\tau_0\xi_j)\xi_j^2(t_j+\psi_j)^2\right]\\
&\quad + \mathbb{E}\left[\left(\sigma_1(\tau_0\xi_i) + \sigma_1'(\tau_0\xi_i)\xi_i\theta_i + \frac{1}{8\tau_0^2}\sigma_1''(\tau_0\xi_i)\xi_i^2(t_i+\psi_i)^2\right)X_{ij}\right] + O(p^{-3/2})\\
&= \left(p_0 + \tau_0 p_2\theta_i + \frac{\tau_0^2 p_4 + p_2}{8\tau_0^2}(t_i+\psi_i)^2\right)\left(p_0 + \tau_0 p_2\theta_j + \frac{\tau_0^2 p_4 + p_2}{8\tau_0^2}(t_j+\psi_j)^2\right)\\
&\quad + \mathbb{E}\left[\sigma_1(\tau_0\xi_i)X_{ij}\right] + S_{ij} + O(p^{-3/2}),
\end{aligned}$$

where the expectation is taken with respect to the *independent* $\xi_i$ and $\xi_j$ (so, in fact, conditioned on $\mathbf{x}_i, \mathbf{x}_j$), so that

$$\begin{aligned}
[\mathbf{K}_{\mathrm{CK},1}]_{ij} &= \mathbb{E}_{u,v}[\sigma_1(u)\sigma_1(v)]\\
&= \left(p_0 + \tau_0 p_2\theta_i + \frac{\tau_0^2 p_4 + p_2}{8\tau_0^2}(t_i+\psi_i)^2\right)\left(p_0 + \tau_0 p_2\theta_j + \frac{\tau_0^2 p_4 + p_2}{8\tau_0^2}(t_j+\psi_j)^2\right)\\
&\quad + \mathbb{E}\left[\frac{1}{\tau_0}\xi_i\sigma_1(\tau_0\xi_i)\sigma_1'(\tau_0\xi_j)A_{ij} + \frac{1}{2\tau_0^2}\left(\frac{1}{p}\mathbf{z}_i^\mathsf{T}\mathbf{z}_j\right)^2\right.\\
&\quad\quad \left.\left(\xi_i^2\sigma_1(\tau_0\xi_i)\sigma_1''(\tau_0\xi_j) - \frac{1}{\tau_0}\sigma_1(\tau_0\xi_i)\xi_j\sigma_1'(\tau_0\xi_j)\right)\right]\\
&\quad + S_{ij} + O(p^{-3/2})\\
&= \left(p_0 + \frac{p_2}{2}\chi_i + \frac{p_4}{8}(t_i+\psi_i)^2\right)\left(p_0 + \frac{p_2}{2}\chi_j + \frac{p_4}{8}(t_j+\psi_j)^2\right)\\
&\quad + p_1^2 A_{ij} + \frac{1}{2\tau_0^2}\left(\frac{1}{p}\mathbf{z}_i^\mathsf{T}\mathbf{z}_j\right)^2 \cdot p_2\left(\mathbb{E}[(\xi^2-1)\sigma_1(\tau_0\xi)]\right) + S_{ij} + O(p^{-3/2})\\
&= p_0^2 + \frac{p_0 p_2}{2}(\chi_i + \chi_j) + \frac{p_0 p_4}{8}\left((t_i+\psi_i)^2 + (t_j+\psi_j)^2\right) + \frac{p_2^2}{4}(t_i+\psi_i)(t_j+\psi_j)\\
&\quad + p_1^2 A_{ij} + \frac{p_2^2}{2}\left(\frac{1}{p}\mathbf{z}_i^\mathsf{T}\mathbf{z}_j\right)^2 + S_{ij} + O(p^{-3/2}),
\end{aligned}$$

where we recall the shortcut

$$\theta_i \equiv \frac{1}{2\tau_0}(\|\boldsymbol{\mu}_i\|^2/p + 2\boldsymbol{\mu}_i^\mathsf{T}\mathbf{z}_i/p + t_i + \psi_i) - \frac{1}{8\tau_0^2\tau_0}(t_i+\psi_i)^2 \equiv \frac{\chi_i}{2\tau_0} - \frac{(t_i+\psi_i)^2}{8\tau_0^2\tau_0} = O(p^{-1/2}), \quad (81)$$

with

$$\chi_i \equiv t_i + \psi_i + \|\boldsymbol{\mu}_i\|^2/p + 2\boldsymbol{\mu}_i^\mathsf{T}\mathbf{z}_i/p = \|\mathbf{x}_i\|^2 - \tau_0. \quad (82)$$

This gives, in matrix form,

$$\begin{aligned}
\mathbf{K}_{\mathrm{CK},1} = {}& p_0^2 \mathbf{1}_n \mathbf{1}_n^\mathsf{T} + p_1^2\left(\frac{1}{p}\mathbf{Z}^\mathsf{T}\mathbf{Z} + \frac{1}{p}\mathbf{J}\mathbf{M}^\mathsf{T}\mathbf{M}\mathbf{J}^\mathsf{T} + \frac{1}{p}(\mathbf{J}\mathbf{M}^\mathsf{T}\mathbf{Z} + \mathbf{Z}^\mathsf{T}\mathbf{M}\mathbf{J}^\mathsf{T})\right) \\
& + \frac{p_0 p_2}{2}(\boldsymbol{\chi}\mathbf{1}_n^\mathsf{T} + \mathbf{1}_n\boldsymbol{\chi}^\mathsf{T}) + \frac{p_0 p_4}{8}\left((\{t_a\mathbf{1}_{n_a}\}_{a=1}^K + \boldsymbol{\psi})^2\mathbf{1}_n^\mathsf{T} + \mathbf{1}_n[(\{t_a\mathbf{1}_{n_a}\}_{a=1}^K + \boldsymbol{\psi})^2]^\mathsf{T}\right) \\
& + \frac{p_2^2}{4}(\{t_a\mathbf{1}_{n_a}\}_{a=1}^K + \boldsymbol{\psi})(\{t_a\mathbf{1}_{n_a}\}_{a=1}^K + \boldsymbol{\psi})^\mathsf{T} + \frac{p_2^2}{2}\left(\frac{1}{p}\mathbf{Z}^\mathsf{T}\mathbf{Z}\right)^{\circ 2} \\
& + (\mathbb{E}[\sigma_1^2(\tau_0\xi)] - p_0^2 - \tau_0^2 p_1^2)\mathbf{I}_n + O_{\|\cdot\|}(p^{-1/2}),
\end{aligned}$$

where we denote $\boldsymbol{\chi} \equiv \{\chi_i\}_{i=1}^n \in \mathbb{R}^n$, $\mathbf{A}^{\circ 2}$ the *entry-wise* square of the matrix $\mathbf{A} \in \mathbb{R}^{n\times n}$, i.e., $[\mathbf{A}^{\circ 2}]_{ij} = [\mathbf{A}_{ij}]^2$, and use again the fact that $\|\mathbf{A}\| \le n\|\mathbf{A}\|_\infty$ for $\mathbf{A} \in \mathbb{R}^{n\times n}$ with $\|\mathbf{A}\|_\infty = \max_{ij}|\mathbf{A}|_{ij}$, $\{S_{ij}\}_{i,j} = O_{\|\cdot\|}(p^{-\frac{1}{2}})$ as well as $\left(\frac{1}{p}\mathbf{Z}^\mathsf{T}\mathbf{Z}\right)^{\circ 2} = \frac{1}{p}\mathbf{J}\mathbf{T}\mathbf{J}^\mathsf{T} + O_{\|\cdot\|}(p^{-1/2})$ according to [11].

Finally, using the fact that for $\mathbf{P} = \mathbf{I}_n - \mathbf{1}_n\mathbf{1}_n^\mathsf{T}/n$, we have $\mathbf{1}_n^\mathsf{T}\mathbf{P} = \mathbf{0}$, $\mathbf{P}\mathbf{1}_n = \mathbf{0}$, we conclude the proof of (73) with the same expression of $\tilde{\mathbf{K}}_{\mathrm{CK},1}$ as in the statement of our Theorem 1, *without* the assumption $\mathbb{E}[\sigma_1(\tau_0\xi)] = 0$. This, however, no longer holds in the multi-layer setting with a number of layers $L \ge 1$.

### A.4 Proof and discussions of Corollary 1

To prove Corollary 1, it can be easily checked that the i.i.d. entries of the weights $\mathbf{W}$ defined in (17) have zero mean and unit variance. So we focus on the design of the activations.

To ensure that the activation functions $\sigma_\ell(\cdot)$s are "centered" and satisfy $\mathbb{E}[\sigma_\ell(\tau_{\ell-1}\xi)] = 0$, we define, with a slight abuse of notation, for the non-negative sequence $\tau_1, \ldots, \tau_L$ defined in Theorem 1,

$$\sigma_T(t) = a \cdot (1_{t<s_1} + 1_{t>s_2}), \quad \sigma_Q(t) = b_1 \cdot (1_{t<r_1} + 1_{t>r_4}) + b_2 \cdot 1_{r_2 \le t \le r_3}, \quad (83)$$

and take $\alpha_{\ell,0} \equiv \mathbb{E}[\sigma_T(\tau_{\ell-1}\xi)]$, $\sigma_T(\tau_{\ell-1}\xi) \equiv \tilde{\sigma}_T(\tau_{\ell-1}\xi) = \sigma_T(\tau_{\ell-1}\xi) - \alpha_{\ell,0}$, which serves as the activation of the first $\ell = 1, \ldots, L-1$ layers, and $a$, $s_1$ and $s_2$ satisfying the following equations

$$\mathbb{E}[\sigma_T'(\tau_{\ell-1}\xi)] = \frac{a}{\sqrt{2\pi}\tau_{\ell-1}} \cdot \left(e^{-s_2^2/(2\tau_{\ell-1}^2)} - e^{-s_1^2/(2\tau_{\ell-1}^2)}\right), \quad (84)$$

$$\mathbb{E}[\sigma_T''(\tau_{\ell-1}\xi)] = \frac{a}{\sqrt{2\pi}\tau_{\ell-1}^3} \cdot \left(s_2 e^{-s_2^2/(2\tau_{\ell-1}^2)} - s_1 e^{-s_1^2/(2\tau_{\ell-1}^2)}\right), \quad (85)$$

$$\mathbb{E}[(\sigma_T^2(\tau_{\ell-1}\xi))''] = \frac{a^2 - 2a\cdot\alpha_{\ell,0}}{\sqrt{2\pi}\tau_{\ell-1}^3} \cdot \left(s_2 e^{-s_2^2/(2\tau_{\ell-1}^2)} - s_1 e^{-s_1^2/(2\tau_{\ell-1}^2)}\right), \quad (86)$$

$$\mathbb{E}[\sigma_T^2(\tau_{\ell-1}\xi)] = \frac{a^2}{2}\left(\mathrm{erf}\left(\frac{s_1}{\sqrt{2}\tau_{\ell-1}}\right) - \mathrm{erf}\left(\frac{s_2}{\sqrt{2}\tau_{\ell-1}}\right) + 2\right) - \alpha_{\ell,0}^2, \quad (87)$$

and $\alpha_{L,0} \equiv \mathbb{E}[\sigma_Q(\tau\xi)]$, $\sigma_T(\tau\xi) \equiv \tilde{\sigma}_T(\tau\xi) = \sigma_T(\tau\xi) - \alpha_{L,0}$, which serves as the activation of the last and $L$th layer, and $b_1$, $b_2$, $r_1$, $r_2$, $r_3$ and $r_4$ satisfying the following equations

$$\mathbb{E}[\sigma_Q'(\tau\xi)] = \frac{b_1\left(e^{-r_4^2/(2\tau^2)} - e^{-r_1^2/(2\tau^2)}\right)}{\sqrt{2\pi}\tau} + \frac{b_2\left(e^{-r_2^2/(2\tau^2)} - e^{-r_3^2/(2\tau^2)}\right)}{\sqrt{2\pi}\tau}, \quad (88)$$

$$\mathbb{E}[\sigma_Q''(\tau\xi)] = \frac{b_1\left(r_4 e^{-r_4^2/(2\tau^2)} - r_1 e^{-r_1^2/(2\tau^2)}\right)}{\sqrt{2\pi}\tau^3} + \frac{b_2\left(r_2 e^{-r_2^2/(2\tau^2)} - r_3 e^{-r_3^2/(2\tau^2)}\right)}{\sqrt{2\pi}\tau^3}, \quad (89)$$

$$\mathbb{E}[(\sigma_Q^2(\tau\xi))''] = \frac{b_1^2\left(r_4 e^{-r_4^2/(2\tau^2)} - r_1 e^{-r_1^2/(2\tau^2)}\right)}{\sqrt{2\pi}\tau^3} + \frac{b_2^2\left(r_2 e^{-r_2^2/(2\tau^2)} - r_3 e^{-r_2^3/(2\tau^2)}\right)}{\sqrt{2\pi}\tau^3}, \quad (90)$$
$$- 2\alpha_{L,0}\mathbb{E}[(\sigma_Q''(\tau\xi))],$$

$$\mathbb{E}[(\sigma_Q^2(\tau\xi))] = \frac{b_1^2}{2}\left(\mathrm{erf}\left(\frac{r_1}{\sqrt{2}\tau}\right) - \mathrm{erf}\left(\frac{r_4}{\sqrt{2}\tau}\right)\right) + b_1^2 + \frac{b_2^2}{2}\left(\mathrm{erf}\left(\frac{r_2}{\sqrt{2}\tau}\right) - \mathrm{erf}\left(\frac{r_3}{\sqrt{2}\tau}\right)\right)$$
$$- \alpha_{L,0}^2$$
$$(91)$$

with $\tau = \tau_{L-1}$.

A few remarks on Corollary 1 and Algorithm 1 are as follows.

**On the numerical determinations of $\sigma_T$ and $\sigma_Q$.** The above system of nonlinear equations does not admit explicit solutions, but can be solved efficiently using, for example, a (numerical) least squares method. Precisely, we use the numerical least squares method (the optimize.minimize function of SciPy library) to solve the above system of equations, and run for $1\,000$ times with random and independent initializations to get $1\,000$ solutions, among which we choose the optimal parameters to determine $\sigma_Q$ and $\sigma_T$.

**On the two activations.** Note that in Algorithm 1 we use the activation $\sigma_T$ and $\sigma_Q$ respectively for the first $\ell = 1, \ldots, L - 1$ and the final and $L$th layer, since we *only* need to match the key parameters $\alpha_{\ell,1}$, $\alpha_{\ell,2}$ and $\alpha_{\ell,3}$ for the first $\ell = 1, \ldots, L - 1$ layer, and the additional parameter $\tau_\ell$ for the last $L$th, so as to obtain spectrally equivalent CK and NTK matrices for the whole network of depth $L$. Also note that the proposed activation functions $\sigma_T$ and $\sigma_Q$ have respectively three and five (in fact six parameters with the symmetric constraint $r_1 - r_2 = r_3 - r_4$ as in Figure 1) parameters that are freely tunable. And we have respectively three and four (nonlinear) equations to determine these parameters in the system of equations above.

## B  Additional experiments

In this section, we provide additional experiments to demonstrate the advantageous performance of the proposed "lossless" compression approach. Figure 4 depicts the classification accuracies using three different neural networks: (i) the original "dense and unquantized" nets with three fully-connected layers of ReLU activations, (ii) the proposed sparse and quantized DNN model as per Algorithm 1, and (iii) the "heuristically" compressed networks by (iii-i) uniformly and randomly zeroing out $90\%$ of the weights, as well as (iii-ii) natively binarizing using $\sigma(t) = 1_{t<-1} + 1_{t>1}$, on two tasks of MNIST data classification [31] having five classes (digits $0, 1, 2, 3, 4$) and two classes (digits $6$ versus $8$). This allows us to have a more qualitative assessment of the impact of data and task on the performance of the proposed compression scheme. We see, as in Figure 3 for ten-class MNIST and ten-class CIFAR10, that the proposed compression approach significantly outperform the two "naive" compression approaches, and can achieve a memory compression rate of $10^3$ and a level of sparsity up to $90\%$, with virtually no performance loss. Also note that the experimental settings of Figure 4 is almost the *same* as those of Figure 3 in Section 4, except that the former networks have less neurons per layer and slightly higher level of sparsity ($90\%$ here instead of $80\%$ in the setting of Figure 3), to solve the simpler two-class or five-class classification problems.

In Table 1 and 2, we evaluate the impact of activation functions on the classification performance on data of *different nature*, on a set of fully-connected DNN models having three hidden layers (of width $d_1 = 3\,000, d_2 = 3\,000, d_3 = 1\,000$ in each layer) and use the *same* activation $\sigma(\cdot)$ for all layers.

More precisely, Table 1 depicts the classification accuracy and the values of the key parameters $\alpha_1$, $\alpha_2$, $\alpha_3$ and $\tau$ for different activations $\sigma(\cdot)$ in the asymptotic equivalent CK matrix $\tilde{\mathbf{K}}_{\mathrm{CK}}$ defined in Theorem 1 of the third and final layer of the network, on a binary classification of MNIST data (class $6$ versus $8$). Similarly, Table 2 compares the classification accuracy and $\alpha_1$, $\alpha_2$, $\alpha_3$, $\tau$ for different activations, on two-class GMM data with identical mean $\boldsymbol{\mu}_a = \mathbf{0}_p$ and different covariance

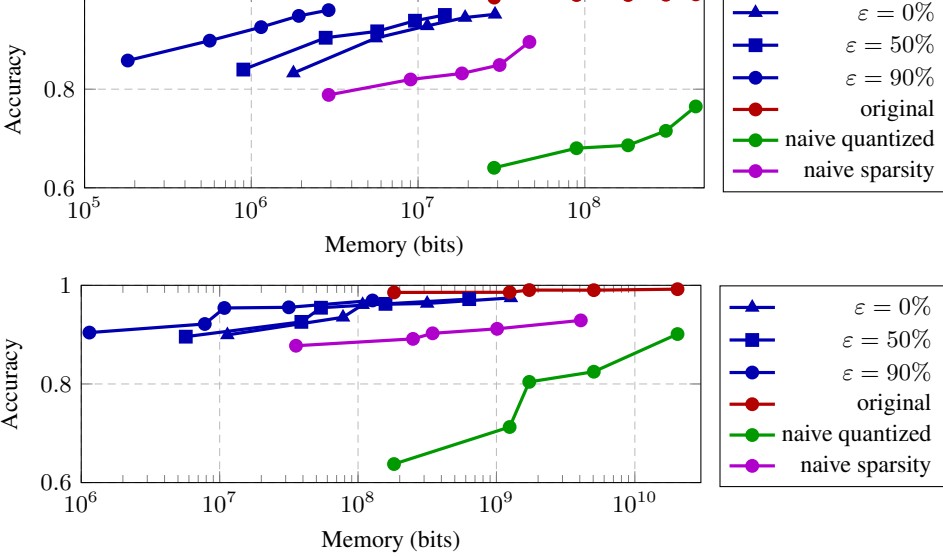

Figure 4: Test accuracy of classification on 2-class (**top**) MNIST dataset - digits 6 versus 8 and 5-class (**bottom**) MNIST dataset - digits $(0, 1, 2, 3, 4)$. **Blue** curves represent the proposed "lossless" compression scheme with different levels of sparsity $\varepsilon \in \{0\%, 50\%, 90\%\}$, **purple** curves represent the heuristic sparsification approach by uniformly zeroing out $90\%$ of the weights, **green** curves represent the heuristic quantization approach using the binary activation $\sigma(t) = 1_{t<-1} + 1_{t>1}$ (only applied on the first two layers, otherwise the performance is too poor to be compared to other curves), and **red** curves represent the original (dense and unquantized) network. All nets have three fully-connected layers, and the original network uses $\mathrm{ReLU}$ activations for all layers. Memory varies due to the **change of layer width** of the network.

$\mathbf{C}_a = (1 + 8(a - 1)/\sqrt{p})\mathbf{I}_p$, $a \in \{1, 2\}$, The numerical experiments are performed on a training set of size $12\,000$, a test set of $1\,800$, with $\mathbf{x}_1, \ldots, \mathbf{x}_{n/2} \in \mathcal{C}_1$ and $\mathbf{x}_{n/2+1}, \ldots, \mathbf{x}_n \in \mathcal{C}_2$, for standard Gaussian $\mathbf{W}$ on both MNIST and GMM data.

We observe from Table 1 and 2 that:

(i) while in theory, the key parameters $\alpha_1$, $\alpha_2$, $\alpha_3$ and $\tau$ (in Theorem 1) depend on both (the statistics of) the data and the activation, the impact of the activation $\sigma$ appears much more significant;

(ii) by using some $\sigma$ (with the corresponding $\alpha_1$, $\alpha_2$ and/or $\alpha_3$ being zero), one asymptotically "discards" either the first-order ($\boldsymbol{\mu}_a$) or the second-order ($\mathbf{t}, \mathbf{T}$) statistics of the data (which, per Theorem 1, are respectively weighted by the key parameter $\alpha_1$, $\alpha_2$ and $\alpha_3$), resulting in performance degradation;

(iii) precisely, we divide commonly used activations in Table 1 and 2 into the following three categories:

   1. covariance-oriented activations with $\alpha_1 = 0$: this includes $\cos(t)$ and $|t|$; and

   2. mean-oriented activations with $\alpha_2 = 0$ and $\alpha_3 = 0$: this includes $1_{t \geq 0}$, $\mathrm{sign}(t)$, $(1 + e^{-t})^{-1}$ [21], $\sin(t)$, linear function, and the Gaussian error function $\mathrm{erf}(t)$; and

   3. balanced activations with nonzero $\alpha_1, \alpha_2, \alpha_3$: this includes $\mathrm{ReLU}$ activation $\mathrm{ReLU}(t) = \max(t, 0)$ and Leaky ReLU activation [42].

The above classification of activation functions is reminiscent of that proposed in [36], which is, however, only valid in a single-hidden-layer setting. In line with the observations made in [36], we see in Table 1 that covariance-oriented activations behave poorly in the classification of MNIST data (that are known to have very different first-order statistics, see for example [36, Table 3]), while mean-oriented activations yield unsatisfactory performance on GMM data having different covariance structure in Table 2. In a sense, the parameter $\alpha_1$

characterizes the "ability" of a given net to extract first-order data statistics and $\alpha_2, \alpha_3$ the "ability" to extract second-order statistics from the input data, respectively.

Table 1: Classification accuracy and values of $\alpha_1$, $\alpha_2$, $\alpha_3$ and $\tau$ at the third and final layer, on MNIST data (digits 6 versus 8).

| $\sigma(t)$ | $\alpha_1$ | $\alpha_2$ | $\alpha_3$ | $\tau$ | Accuracy |
|---|---|---|---|---|---|
| $\max(0,t)$ | 0.0156 | 0.0105 | 0.0112 | 0.1994 | 0.971 |
| $0.1t \cdot 1_{t<0} + t \cdot 1_{t \geq 0}$ | 0.0083 | 0.0097 | 0.0081 | 0.1750 | 0.9654 |
| $1_{t \geq 0}$ | 0.0642 | 0 | 0 | 0.5 | 0.9665 |
| $\text{sign}(t)$ | 0.1779 | 0 | 0 | 0.4689 | 0.9715 |
| $1/(1+e^{-t})$ | 0.0002 | 0 | 0 | 0.0129 | 0.9637 |
| $\sin(t)$ | 0.1779 | 0 | 0 | 0.4689 | 0.9749 |
| $t$ | 1 | 0 | 0 | 1.0021 | 0.981 |
| $\text{erf}(t)$ | 0.2166 | 0 | 0 | 0.5053 | 0.9788 |
| $\cos(t)$ | 0 | 0.0003 | 0 | 0.0116 | 0.5257 |
| $|t|$ | 0 | 0.0209 | 0 | 0.2195 | 0.5709 |

Table 2: Classification accuracy and values of $\alpha_1$, $\alpha_2$, $\alpha_3$ and $\tau$ at the third and final layer, on GMM data.

| $\sigma(t)$ | $\alpha_1$ | $\alpha_2$ | $\alpha_3$ | $\tau$ | Accuracy |
|---|---|---|---|---|---|
| $\max(0,t)$ | 0.0156 | 0.0092 | 0.0099 | 0.2128 | 0.8945 |
| $0.1t \cdot 1_{t<0} + t \cdot 1_{t \geq 0}$ | 0.0083 | 0.0085 | 0.0071 | 0.1867 | 0.9079 |
| $1_{t \geq 0}$ | 0.0564 | 0 | 0 | 0.5 | 0.5028 |
| $\text{sign}(t)$ | 0.2256 | 0 | 0 | 1 | 0.4916 |
| $1/(1+e^{-t})$ | 0.0002 | 0 | 0 | 0.0135 | 0.5173 |
| $\sin(t)$ | 0.1512 | 0 | 0 | 0.4729 | 0.5025 |
| $t$ | 1 | 0 | 0 | 1.0693 | 0.5045 |
| $\text{erf}(t)$ | 0.1912 | 0 | 0 | 0.51 | 0.4989 |
| $\cos(t)$ | 0 | 0.0003 | 0 | 0.015 | 0.9598 |
| $|t|$ | 0 | 0.0184 | 0 | 0.2342 | 0.9302 |