# OpenReview forum: ""Lossless" Compression of Deep Neural Networks: A High-dimensional Neural Tangent Kernel Approach"
_NeurIPS.cc/2022/Conference — NeurIPS 2022 Accept_

### Official Review · Reviewer_7Ta5 · 2022-07-04

**Rating:** 7
**Confidence:** 3
**Soundness:** 4 excellent
**Presentation:** 4 excellent
**Contribution:** 3 good

**Summary:**

This paper characterizes the asymptotic spectral equivalence between NTKs of dense and quantized networks. It shows that under certain assumptions of data (high-dim, Gaussian mixture data) and network architectures (wide MLPs), quantized networks have the same NTK eigenspectra of unquantized ones. This finding allows the authors to perform model quantization with little performance degradation.

**Questions:**

* Quantized neural networks in the current theoretical and empirical results focus on ternary-valued weights, i.e., NNs with weights that take values in {-1, 0, 1}. Would it be possible to extend the results to binary networks, that is, {-1, 1}-valued networks? It seems that such binary-valued weights (up to a proper scaling) can still satisfy assumptions for Theorem 1 and 2.
* In Figure 3 in Section 4, the proposed lossless compression approach is numerically compared with two heuristic sparsification and quantization approaches.  The heuristic sparsification approaches uniformly zero out 80% of the weights. This approach is very brutal. Would it be possible to add a more realistic baseline of magnitude-based pruning -- remove a fraction of weights that have the lowest magnitude (absolute value) and keep other weights. This baseline is used more often as a baseline for model pruning and is sometimes surprisingly strong (see, e.g., https://arxiv.org/pdf/1902.09574.pdf)
* If I understand correctly, the current lossless compression approach uses layerwise pruning and quantization. That is, all layers share the same sparsity level in Algorithm 1. Would it be possible to use the same theoretical insights to perform global compression; that is, compress weights in MLP layers collectively without regard for the specific layer. This may lead to better compression results since it gives more freedom to the weight selection.

**Limitations:**

One limitation of the paper, as mentioned in my questions above, is its lack of natural baselines for model pruning in the experiment sections. I encourage the authors to consider incorporating them.

**Strengths And Weaknesses:**

The paper is very well written -- the authors crafted their paper with immense care and taste for mathematical detail. The main results of the paper (Theorem 1 and Theorem 2) are novel and subsume previous studies [2, 32] as special cases. Overall, I think this is a high-quality paper.

One weakness of this paper is in its numerical evaluation. As I detailed below, the baselines used for the model pruning (randomly removing weights) seem to be too brutal and too weak. It is beneficial to incorporate more realistic baselines such as magnitude-based pruning.

---

> ### Author Response · Authors · 2022-08-02
> **Reply to Reviewer 7Ta5**
>
> We thank the reviewer for his/her positive support and constructive comments.
> In the following, we provide point-by-point answers (**A**) to the comments raised by the reviewer.
>
>
> * "Quantized neural networks in the current theoretical and empirical results focus on ternary-valued weights, i.e., NNs with weights that take values in {-1, 0, 1}. Would it be possible to extend the results to binary networks, that is, {-1, 1}-valued networks? It seems that such binary-valued weights (up to a proper scaling) can still satisfy assumptions for Theorem 1 and 2."
>
> **A**: The binarized weights can be achieved by taking the sparsity level $\varepsilon = 0$ in the proposed NTK-LC approach, in which case the weights are distributed according to a symmetric Bernoulli, see also Equation (16) in the paper.
>
> * "In Figure 3 in Section 4, the proposed lossless compression approach is numerically compared with two heuristic sparsification and quantization approaches. The heuristic sparsification approaches uniformly zero out 80% of the weights. This approach is very brutal. Would it be possible to add a more realistic baseline of magnitude-based pruning -- remove a fraction of weights that have the lowest magnitude (absolute value) and keep other weights. This baseline is used more often as a baseline for model pruning and is sometimes surprisingly strong (see, e.g., https://arxiv.org/pdf/1902.09574.pdf)"
>
> **A**: We thank the reviewer for this constructive suggestion. We have added, in Figure 3 of the revised version of the paper, the comparison between the proposed NTK-LC approach and the widely used magnitude-based pruning method suggested by the reviewer. We observe a better "performance-complexity tradeoff" achieved by the proposed NTK-LC method.
> More specifically, we have added two groups of experiments using the NTK-LC ternary weight approach (that has ternarized weights but *unquantized* activations) and the popular magnitude-based pruning method.
> NTK-LC (with both weights and activation quantized) yields slightly inferior performance than its ternary-weights variant, but achieves higher accuracy under the same memory budget compared with the popular magnitude-based pruning.
>
> * "If I understand correctly, the current lossless compression approach uses layerwise pruning and quantization. That is, all layers share the same sparsity level in Algorithm 1. Would it be possible to use the same theoretical insights to perform global compression; that is, compress weights in MLP layers collectively without regard for the specific layer. This may lead to better compression results since it gives more freedom to the weight selection."
>
> **A**: We thank the reviewer for this interesting and insightful question. The proposed NTK-LC compression method can also be used with different sparsity for each layer, as long as Assumption 2 is satisfied. As for the trade-off between the sparsity of each layer, some previous efforts (e.g., [1, 2]) have already provided insightful results and discussions on this point, e.g., less pruning in the first layers and more in the last layers of the network. It could be of future interest to provide more theoretical insights on such "optimal sparsity schedule" (if there exists) within the proposed analysis framework. We leave that for future work.
>
> [1] Su  J, Chen Y, Cai T, et al. Sanity-checking pruning methods: Random tickets can win the jackpot[J]. Advances in Neural Information Processing Systems, 2020, 33: 20390-20401.
>
> [2] Han S, Pool J, Tran J, et al. Learning both weights and connections for efficient neural network[J]. Advances in neural information processing systems, 2015, 28.

---

> > ### Comment · Reviewer_7Ta5 · 2022-08-06
> > **Response to the rebuttal**
> >
> > I'd like to thank the authors for their response. It addressed all my questions. Especially, the performance-complexity tradeoff in Figure 3 is very helpful. I decided to keep my rating (7: accept).

---

### Official Review · Reviewer_ZUPo · 2022-07-12

**Rating:** 5
**Confidence:** 4
**Soundness:** 2 fair
**Presentation:** 3 good
**Contribution:** 3 good

**Summary:**

The authors showed asymptotic eigen-spectral equivalence conditions for fully-connected NTK given GMM data and certain assumptions, based thereon they proposed a net compression scheme with sparse and low-precision random weights, and demonstrated with examples.


**Questions:**

- The authors ran experiments on real not-so-wide DNNs that also break the assumptions, e.g. taking real image data that is not GMM and taking convolutional nets that are not fully-connected.  And they reported, in their own words "unexpected close match" to the theory.  This, in its own right, is an important observation that future theoretical relaxations might address.  But the gap is rather inadequately characterized by experiment here: e.g. any empirical characterization of the notion of "unexpectedness"?
- Another thing I do not appreciate is the trickiness of the scalar calibration due to nonlinearities.  This seems to stem from the requirement of unity Gaussianity.  Is there any empirical data supporting the near-Gaussianity under real data?  Does there exist any badly behaved nonlinearities that significantly deviated from theory?  Does this mean NNs trained with normalization layers are more conducive to such compression, or could the procedure be simpler in those cases?
- I find the naive sparse/quantized baselines presented in experiments inadequate.  Of course this is a post-training compression technique, but since it is derived from NTK, it is natural and important to ask how it generalizes compared to much stronger, non-post-training baselines, such as, say, a winning lottery ticket--does there exist any other compressed nets that generalizes better regardless of how expensive the compression procedure was?
- "Lossless" (used not in quotation marks) might be a false advertisement.
- Figure 2, missing axes labels.


**Limitations:**

See above.

**Strengths And Weaknesses:**

[+] Results linking NTK and random matrix theory with DNN compression is of timely interest to the field.

[+] Though I cannot say I followed all proofs, the main ideas and motivations are well presented.

[-] A lack of comprehensive experimental comparison with baseline approaches is limiting the practical significance of the findings.

---

> ### Author Response · Authors · 2022-08-02
> **Reply to Reviewer ZUPo**
>
> We thank the reviewer for his/her positive support and for the valuable insights shared in the constructive comments.
> In the following, we provide point-by-point answers (**A**) to the comments raised by the reviewer.
>
> * "The authors ran experiments on real not-so-wide DNNs that also break the assumptions, e.g. taking real image data that is not GMM and taking convolutional nets that are not fully-connected. And they reported, in their own words "unexpected close match" to the theory. This, in its own right, is an important observation that future theoretical relaxations might address. But the gap is rather inadequately characterized by experiment here: e.g. any empirical characterization of the notion of "unexpectedness"?"
>
> **A**: This is an interesting and important question. We would like to clarify that (i) in Figure 2, we compare the eigenspectra of CK matrices of fully-connected (FC) nets for GMM and real MNIST data of reasonable size ($n,p$ only in hundreds), for which we observe an "unexpected close match" between the theory and practice, since we clearly violate the asymptotic ($n,p\to\infty$) and GMM assumptions, but the theoretical results remain approximately valid: This may be due to a fast convergence rate as $n,p$ grow large or an underlying "universality" as discussed after Remark 2 and in Remark 4 of the Appendix of the paper; and (ii)
> in Figure 3, we compare the performance of uncompressed DNNs versus compressed nets using the proposed NTK-LC (which is based on the limiting NTK and does **not** hold, a priori, for the network of finite width under study) and other compression approaches, where we violate the infinite network size assumption of NTK and also the GMM data assumption (but not the fully-connected assumption, since we *only* compressed the FC layers, see the discussion above Figure 3) and obtain a sparse and quantized network with up to a factor of $10^3$ less memory and limited performance degradation.
> And we will conduct additional experiments to empirically characterize, in a quantitative manner, how the different types of violations above affect the final performance of the network (which is not available for the moment due to the short time slot).
>
> * "Another thing I do not appreciate is the trickiness of the scalar calibration due to nonlinearities. This seems to stem from the requirement of unity Gaussianity. Is there any empirical data supporting the near-Gaussianity under real data? Does there exist any badly behaved nonlinearities that significantly deviated from theory? Does this mean NNs trained with normalization layers are more conducive to such compression, or could the procedure be simpler in those cases?"
>
> **A**: We thank the reviewer for this interesting question. The (exact) Gaussian distribution assumption may not be necessary and is demanded here mainly for the simplicity of mathematical derivation, as has been discussed in the paragraph after Remark 2 and in Remark 4 in the Appendix. We agree with the reviewer's intuition and also conjecture that "NNs trained with normalization layers are more conducive to the proposed compression," but establishing such a proof in a rigorous manner seems out of the scope of this paper.
>
> * "I find the naive sparse/quantized baselines presented in experiments inadequate. Of course this is a post-training compression technique, but since it is derived from NTK, it is natural and important to ask how it generalizes compared to much stronger, non-post-training baselines, such as, say, a winning lottery ticket--does there exist any other compressed nets that generalizes better regardless of how expensive the compression procedure was?"
>
> **A**: We thank the reviewer for this constructive comment. We have added, in Figure 3 of the revised version of the manuscript, the comparison between the proposed NTK-LC approach and the magnitude-based pruning method (as demanded by Reviewer 7Ta5, among the most widely used NN compression schemes), which shows the advantageous performance of the proposed NTK-LC approach. More empirical results (such as those related to winning lottery ticket) will be made available in an updated version of the paper. The theoretical analysis, though, requires more effort and is out of the scope of this paper.
>
>
> *  "'Lossless' (used not in quotation marks) might be a false advertisement." and "Figure 2, missing axes labels".
>
> **A**: We thank the reviewer for pointing these out and they are all fixed in the revised version.

---

### Official Review · Reviewer_VcQf · 2022-07-18

**Rating:** 6
**Confidence:** 3
**Soundness:** 3 good
**Presentation:** 3 good
**Contribution:** 2 fair

**Summary:**

This paper studies the problem of neural network compression using analysis in the high-dimensional NTK regime. Their main results show that under this regime, the spectral properties of both NTK and CK matrices are independent of the distribution of the weights up to normalization and centering. Instead they depend on a number of parameters that define the activation functions at each layer. This finding informs a new compression technique where a new (compressed) network can match the activation parameters at each layer to enjoy the same spectral properties as the original net. This NTK-LC technique is evaluated on synthetic data by qualitatively comparing the distribution of eigenvalues and on real data by comparing test accuracy with naive baselines.

**Questions:**

Question: What is the novelty of the NTK-LC approach and how does it compare with state of the art methods?

Suggestion: As mentioned in Weakness 1, the paper would benefit from a discussion on how the convergence and generalization properties of ultra-wide DNNs can depend only on the eigenspectra.

Suggestion: As mentioned in Weakness 3, the paper would benefit from a quantitative metric of eigenvalue closeness.

**Limitations:**

The authors have addressed the limitations of using NTK theory to explain the behavior of modern neural networks.

**Strengths And Weaknesses:**

Strengths:

The paper has a clear motivation in the field of neural network compression, a relevant problem that is lacking theory. It is clearly written with thorough theoretical results and experiments on both synthetic and real-world data.

Weaknesses:
1. The claim in line 49 seems to be a central theme of the paper but has no follow-up discussion on its meaning and implications.
2. Theoretical claims are presented in the asymptotic regime of infinite n and p (Assumption 1).
3. A particular GMM distribution is chosen for the input data of the studied model without justification for why it is the relevant distribution to be analyzing.
4. The results in Figure 2 rely on a qualitative measure of "closeness" to evaluate the method instead of a metric that can be quantified and compared. Many of the markings in the top histograms are barely noticeable and require magnification to be seen.
5. The results in Figure 3 are compared with "naive" baselines instead of competitive state of the art methods.

---

> ### Author Response · Authors · 2022-08-02
> **Reply to Reviewer VcQf (Part-II)**
>
> We thank the reviewer for his/her positive support and constructive comments.
> In the following, we provide point-by-point answers (**A**) to the comments raised by the reviewer.
>
> * "The results in Figure 2 rely on a qualitative measure of "closeness" to evaluate the method instead of a metric that can be quantified and compared. Many of the markings in the top histograms are barely noticeable and require magnification to be seen." and "Suggestion: As mentioned in Weakness 4, the paper would benefit from a quantitative metric of eigenvalue closeness."
>
> **A**: To provide a quantitative measure for the NTK eigenvalue "closeness," we have added, in the revised version, the spectral norm errors between $K_{CK}$ and $\tilde K_{CK}$ (as has been established in Theorem 1 in the $n,p \to \infty$ limit). Specifically, in Figure 3 **top**, we have $\parallel K_{\rm CK} - \tilde K_{\rm CK} \parallel = 0.15$ (**left** for GMM data) and $\parallel K_{\rm CK} - \tilde K_{\rm CK} \parallel = 6.86$ (**right** for MNIST data).
> Besides, we have measured the similarity between the eigenvalues of $K_{\rm CK}$ and $\tilde K_{\rm CK}$ using three different (histogram similarity) metrics: the cosine similarity [7], the correlation and the intersection [8]. The similarity estimates based on these three approaches are all close to one (in fact all greater than 0.99), indicating an extremely close match between the two histograms.
> We have also redrawn Figure 3 to ensure the top histograms are easily readable.
>
> * "Question: What is the novelty of the NTK-LC approach and how does it compare with state-of-the-art methods?" and "The results in Figure 3 are compared with "naive" baselines instead of competitive state-of-the-art methods."
>
> **A**: The proposed NTK-LC approach is novel in that it has a novel and sound theoretical foundation that depends on the *precise* CK ad NTK eigenspectra of fully-connected DNN models, which, to the best of our knowledge, is derived for the first time under generic GMM data. In Figure 3 of the revised version, we compare the proposed NTK-LC approach to the magnitude-based pruning method (as also proposed by Reviewer 7Ta5, among the most widely used NN compression schemes), showing the advantageous performance of the proposed NTK-LC approach.
>
> [7] Alfirna Rizqi Lahitani, Adhistya Erna Permanasari, and Noor Akhmad Setiawan. Cosine similarity to determine similarity measure: Study case in online essay assessment. In 2016 4th International Conference on Cyber and IT Service Management, pages 1–6. IEEE, 2016
>
> [8] Lee S M, Xin J H, Westland S. Evaluation of image similarity by histogram intersection[J]. Color Research & Application: Endorsed by Inter‐Society Color Council, The Colour Group (Great Britain), Canadian Society for Color, Color Science Association of Japan, Dutch Society for the Study of Color, The Swedish Colour Centre Foundation, Colour Society of Australia, Centre Français de la Couleur, 2005, 30(4): 265-274.

---

> ### Author Response · Authors · 2022-08-02
> **Reply to Reviewer VcQf (Part-I)**
>
> We thank the reviewer for his/her positive support and constructive comments.
> In the following, we provide point-by-point answers (**A**) to the comments raised by the reviewer.
>
> * "The claim in line 49 seems to be a central theme of the paper but has no follow-up discussion on its meaning and implications." and "Suggestion: As mentioned in Weakness 1, the paper would benefit from a discussion on how the convergence and generalization properties of ultra-wide DNNs can depend only on the eigenspectra."
>
> **A**: We thank the reviewer for this helpful suggestion. It is known that the time evolutions (when trained with gradient descent using a sufficiently small step size) of both the residual error and the in-sample prediction of (sufficiently wide) neural networks can be expressed as *explicit* functions of the NTK eigenvalues and eigenvectors. More specifically, with the notations in the paper and consider $K_{\rm NTK} = \sum_{i=1}^n \lambda_i v_i v_i^T$ the spectral decomposition of the final-layer NTK matrix $K_{\rm NTK}$ with eigenvalue-eigenvector pair $(\lambda_i, v_i)$, one has, for square loss $L(W_1, \ldots, W_L, w) = \frac12 \parallel y - f(X) \parallel^2$ on the training set $(X, y) \in \mathbb{R}^{ p \times n} \times \mathbb{R}^n$ that
> $$ v_i^T \frac{d}{dt} ( y - f_t(X) ) = - \lambda_i r_i(t), \quad \frac{d}{dt} f_t(X) = \sum_{i=1}^n \lambda_i r_i(t) v_i,$$
> for $f_t(X)$ the network output at time step $t$ and residual error component $r_i(t) = v_i^T ( y - f_t(X) )$, see more details in, e.g., [1-3].
> In a sense, the NTK eigenvalue distribution entails the "train-ability" of the NN model and the eigenvectors of the largest eigenvalues indicate the direction in which the loss decays the most rapidly.
> In the revised version of the paper, we have added a sentence in lines 51-53 to make this clearer.
>
>
> * "Theoretical claims are presented in the asymptotic regime of infinite n and p (Assumption 1)."
>
> **A**: Despite the asymptotic nature of the claims in the paper, empirical results show a close match between theory and practice even for $n,p$ in hundreds, see Figure 2. We conjecture the asymptotic results can be extended to a non-asymptotic setting with some additional efforts and under the additional assumption that weights $W$ are sub-gaussian. That is, however, out of the scope of this paper.
>
> * "A particular GMM distribution is chosen for the input data of the studied model without justification for why it is the relevant distribution to be analyzing."
>
> **A**: GMM is among the most widely known and used distributions in the machine learning literature, see for example [Chapter 9, 4]. In the study of large (and deep) neural network models using high dimensional probability and random matrix theory, GMM is particularly appealing in that (i) it allows one to apply many convenient and/or advanced techniques such as the Stein's lemma and Hermite polynomials, and (ii) in the large $n,p$ regime in Assumption 1, many machine learning methods tend to treat real data *as if they were* mere simple GMM, and the eigenspectra of some core random matrices of interest in these methods only depend on the first- and second-order statistics of the data, so GMM is a first simple yet representative and effective model to study, see [5] and [Chapter 8, 6] for more discussions on this point.
> We have discussed more on this point in lines 242-256 as well as in Remark 4 in Appendix A of the revised version.
>
>
> [1]Fan Z, Wang Z. Spectra of the conjugate kernel and neural tangent kernel for linear-width neural networks[J]. Advances in neural information processing systems, 2020, 33: 7710-7721.
>
> [2] Arthur Jacot, Franck Gabriel, and Clément Hongler. Neural Tangent Kernel: Convergence and Generalization in Neural Networks. Advances in Neural Information Processing Systems, 2018, 33: 8571–8580.
>
> [3] Ben Adlam and Jeffrey Pennington. The Neural Tangent Kernel in High Dimensions: Triple Descent and a Multi-Scale Theory of Generalization. In Proceedings of the 37th International Conference on Machine Learning, volume 119 of Proceedings of Machine Learning Research, pages 74–84. PMLR, 2020.
>
> [4] C. M. Bishop, Pattern Recognition and Machine Learning, 1st ed. Springer-Verlag New York, 2006.
>
> [5] M. E. A. Seddik, C. Louart, M. Tamaazousti, and R. Couillet, "Random Matrix Theory Proves that Deep Learning Representations of GAN-data Behave as Gaussian Mixtures," in Proceedings of the 37th International Conference on Machine Learning, 2020, pp. 8573–8582.
>
> [6] R. Couillet and Z. Liao, Random Matrix Methods for Machine Learning. Cambridge University Press, 2022.

---

### Meta-Review · Area_Chair_mjT7 · 2022-08-25

**Recommendation:** Accept
**Confidence:** Certain

**Metareview:**

In the paper, the authors provide theorems that establish that for GMM input data, the NTK matrices of dense and quantized DNNs have the same eigenspectra in the asymptotic limit of high input data dimension and sample size.  These results motivate network compression algorithms which demonstrate good empirical performance even outside the regime for which the proofs are established.  The theorems provide a novel extension that contains previous studies as special cases.  The baseline comparisons included in the paper are somewhat limited in nature, and the authors should re-evaluate their choice to use the word "lossless" with quotes, and instead use a more accurate term that does not require quotes.

**Award:**

No

---

### Decision · Program_Chairs · 2022-09-14

Accept